# OVOL2 sustains postnatal thymic epithelial cell identity

Xue Zhong[1,6], Nagesh Peddada[1,6], Jianhui Wang [1], James J. Moresco[1], Xiaowei Zhan [1,2], John M. Shelton [3], Jeffrey A. SoRelle [4,5], Katie Keller[1], Danielle Renee Lazaro[1], Eva Marie Y. Moresco [1], Jin Huk Choi [1]✉ & Bruce Beutler [1]✉

Distinct pathways and molecules may support embryonic versus postnatal thymic epithelial cell (TEC) development and maintenance. Here, we identify a mechanism by which TEC numbers and function are maintained postnatally. A viable missense allele (C120Y) of *Ovol2*, expressed ubiquitously or specifically in TECs, results in lymphopenia, in which T cell development is compromised by loss of medullary TECs and dysfunction of cortical TECs. We show that the epithelial identity of TECs is aberrantly subverted towards a mesenchymal state in OVOL2-deficient mice. We demonstrate that OVOL2 inhibits the epigenetic regulatory BRAF-HDAC complex, specifically disrupting RCOR1-LSD1 interaction. This causes inhibition of LSD1-mediated H3K4me2 demethylation, resulting in chromatin accessibility and transcriptional activation of epithelial genes. Thus, OVOL2 controls the epigenetic landscape of TECs to enforce TEC identity. The identification of a non-redundant postnatal mechanism for TEC maintenance offers an entry point to understanding thymic involution, which normally begins in early adulthood.

Thymic epithelial cells (TECs) are key thymic stromal cells essential for production of functionally competent and self-tolerant T lymphocytes[1–6]. TECs are subdivided into two populations anatomically and functionally: the cortical TECs (cTECs) and medullary TECs (mTECs). Thymocyte progenitors from the bone marrow migrate into the thymic cortex and interact with cTECs, which are necessary for thymocyte differentiation to CD4 single-positive (SP) or CD8SP stages, proliferation, and positive selection. CD4SP T cells and CD8SP T cells then migrate into the thymic medulla and interact with mTECs, which are necessary for negative selection of self-reactive T cells[7,8]. Dysfunction of TECs causes several immunodeficiencies and autoimmune diseases[9].

In mice and humans, TECs differentiate from TEC progenitors in the thymic primordium beginning during embryogenesis. Expression of the transcription factor Foxn1 is an established feature of TEC progenitors. Foxn1 orchestrates the expression of hundreds of genes necessary for TEC expansion and differentiation[10]. Thus, mice and humans with mutations in Foxn1 are athymic and immunodeficient due to a block in T cell development at an early double-negative (DN) stage. TEC numbers in mice increase during fetal stages and continue to expand until they peak around puberty at 4 weeks of age; human TECs follow a similar developmental stage-specific pattern. From 4 weeks to 12 weeks of age in mice, when TEC numbers are relatively stable, TECs turn over every 10–14 days[11]. Previous reports have proposed that cTECs and mTECs are maintained by bipotent[12–15] and/or lineage-specific TEC progenitors[16,17]. However, the mechanisms necessary to maintain and replenish TECs in mature mice and humans are still largely unknown.

[1]Center for the Genetics of Host Defense, University of Texas Southwestern Medical Center, Dallas, TX 75390-8505, USA. [2]Department of Population and Data Sciences, Quantitative Biomedical Research Center, University of Texas Southwestern Medical Center, Dallas, TX 75390-8821, USA. [3]Internal Medicine-Histopathology Core, University of Texas Southwestern Medical Center, Dallas, TX 75390-8573, USA. [4]Department of Pathology, University of Texas Southwestern Medical Center, Dallas, TX 75390-9072, USA. [5]Department of Pediatrics, University of Texas Southwestern Medical Center, Dallas, TX 75390-9063, USA. [6]These authors contributed equally: Xue Zhong, Nagesh Peddada. ✉e-mail: Jin.Choi@UTSouthwestern.edu; Bruce.Beutler@UTSouthwestern.edu

The *Ovo* gene family encodes an evolutionarily conserved family of zinc-finger transcription factors. In mammals, three Ovo-like homologues contain a single zinc finger domain composed of a tetrad of C2H2 zinc fingers, and N- and C-terminal extensions that include disordered domains. Ovo Like Zinc Finger 2 (OVOL2) represses mesenchymal genes, including those encoding the SNAI1, TWIST, and ZEB family proteins, and thereby inhibits the epithelial to mesenchymal transition (EMT)[18]. OVOL2 is necessary for embryogenesis and organogenesis[19–24], and has been implicated in several types of cancer[25,26]. *Ovol2* knockout in mice results in embryonic lethality at E9.5-E10.5 with phenotypes that include neurectoderm expansion, impaired vascularization, and heart anomalies[19–24]. A role for OVOL2 in TECs, and consequently in thymocyte differentiation, has not previously been described.

Here, we describe a mutation in *Ovol2* that resulted in a failure of cTEC function and loss of mTECs in adult mice. We demonstrate that OVOL2 activates transcription by inhibiting lysine-specific histone demethylase 1 A (LSD1)-mediated H3K4me2 demethylation and remodels chromatin accessibility, particularly affecting genes involved in epithelial and mesenchymal cell proliferation and differentiation. We show that deficiency of OVOL2 induces a transition from an epithelial towards a mesenchymal state in TECs, resulting in progressive thymic hypoplasia due to loss of TECs and thymocytes, although initial thymic development appeared normal. OVOL2 thus supports T cell development in the thymus during postnatal life.

## Results

### Discovery of an *Ovol2* mutation responsible for a severe impairment of lymphopoiesis

We performed a genetic screen to identify mutations that cause abnormal immune cell frequencies in the blood of mice carrying ENU-induced mutations. To date, we have achieved 56.5% saturation of autosomal mouse protein-encoding genes with damaging mutations[27] tested in at least three homozygous mice. The screen has revealed numerous genes with important roles in immune cell development and function[28–33]. Here, we describe the '*boh*' immune phenotype, first identified in a single pedigree in which several animals showed ~72% reductions in B cell proportions in the blood (Fig. 1A, B). The *boh* phenotype also includes a metabolic component resulting in a ~30% increase in body weight (Supplementary Fig. 1A) and inadequate brown fat formation, reported elsewhere[34]. Both the immune (Fig. 1A, B) and metabolic phenotypes mapped to a forward strand C to T base transition at chromosome 2:144317860 (GRCm38) in the coding region of *Ovol2* using a recessive model of inheritance. The *boh* phenotype is the result of a cysteine to tyrosine substitution at position 120 (C120Y) in the OVOL2 protein, within the first zinc finger domain (Fig. 1C). A homozygous null allele of *Ovol2* is embryonic lethal. We validated the *boh* immune phenotype in compound heterozygous mice carrying a null allele and an independently generated recreation of the *boh* allele (*Ovol2*[C120Y/−]) on a clean C57BL/6 J background[34]. *Ovol2*[C120Y/−] mice showed a decreased proportion of B cells in the blood identical to that observed in the mice harboring the original ENU-induced allele (Fig. 1D). Moreover, we discovered that *Ovol2*[C120Y/−] mice had hypoplastic spleens (Fig. 1E and Supplementary Fig. 1B) and severe lymphopenia, monocytopenia, and anemia (Supplementary Fig. 1C). These data support the conclusion that C120Y is a viable hypomorphic substitution, but retains some functionality.

### Thymocyte developmental defects

We used flow cytometry to characterize defects in T cell development and function resulting from mutations in *Ovol2*. Compared to WT mice, *Ovol2*[C120Y/−] mice had reduced absolute numbers of developing T cells in the thymus (Fig. 1F). There was an increased percentage of DN T cells and a severe reduction in double-positive (DP) T cells, suggesting a partial block in the DN to DP transition (Fig. 1G).

The proportion of CD8 + TCR-β+ cells among CD8 + SP thymocytes was elevated in the *Ovol2*[C120Y/−] mice (Supplementary Fig. 1D). Within the DN stage, *Ovol2*[C120Y/−] mice had reduced frequencies of DN3 thymocytes and increased frequencies of DN4 cells (Fig. 1G). Surprisingly, *Ovol2*[C120Y/−] thymocytes undergoing or finished with positive selection (DP, SP4, and SP8) were in an activated state (CD44[high]) (Fig. 1H). Also, *Ovol2*[C120Y/−] mice had decreased fractions of T cells, CD4 T cells, and CD8 T cells in the peripheral blood (Fig. 1I–K), with elevated expression of CD44 compared to WT T cells (Fig. 1H). CD4 T cell deficiency in *Ovol2*[C120Y/−] mice correlated with a defect in T-dependent IgG responses after immunization with ovalbumin precipitated on alum adjuvant (alum-ova) (Fig. 1L). Consistent with the reduction in CD8 T cells, *Ovol2*[C120Y/−] mice showed severe cytotoxic dysfunction after challenge with recombinant Semliki Forest virus-encoded β-galactosidase (rSFV-βgal) (Fig. 1M). Overall, these data indicate that CD4 and CD8 T cell development and function are impaired by OVOL2 deficiency.

### B cell developmental and NK cell defects

*Ovol2*[C120Y/−] mice had reduced numbers of B cells in the bone marrow and spleen (Supplementary Fig. 2A). Frequencies of pre-B, immature, and transitional B cells were decreased, while mature circulating B cells were increased in *Ovol2*[C120Y/−] bone marrow (Supplementary Fig. 2B, C). *Ovol2*[C120Y/−] mice displayed impaired B cell development in the spleen, with an elevated percentage of marginal zone B (MZB) cells and a diminished percentage of follicular B (FO) cells (Supplementary Fig. 2D–G), although the T cell-independent antibody response in *Ovol2*[C120Y/−] mice was normal (Fig. 1N). These data suggested that B cells in *Ovol2*[C120Y/−] mice are maintained by homeostatic proliferation. Although the frequency of NK cells in the blood was preserved (Supplementary Fig. 2H), *Ovol2*[C120Y/−] mice also showed a severe defect in NK cell cytotoxic function in vivo based on their inability to effectively eliminate MHC-class I-deficient target cells (Fig. 1O).

### Hematopoietic extrinsic effects of OVOL2 on lymphocyte development

We examined the bone marrow of *Ovol2*[C120Y/−] mice histologically. Compared to WT mice, *Ovol2*[C120Y/−] mice had thinner cortical bone with larger and more numerous capillaries, and the bone marrow cavity was filled with adipocytes as detected by Oil Red O staining (Supplementary Fig. 2I). The *Ovol2*[C120Y/−] bone marrow also contained increased LepR+ cells (Supplementary Fig. 3).

Flow cytometry analyses revealed alterations in the frequencies of hematopoietic stem cells (HSC) and progenitor cells in *Ovol2*[C120Y/−] bone marrow (Supplementary Fig. 2J–L). Of these, proportions of multipotent progenitors (MPP), lymphoid primed multipotent progenitors (LMPP), common lymphoid progenitors (CLP), and granulocyte-macrophage progenitors (GMP) were decreased in *Ovol2*[C120Y/−] bone marrow (Supplementary Fig. 2J–L).

To distinguish whether the reductions in lymphocytes in *Ovol2*[C120Y/−] mice stemmed from defects intrinsic or extrinsic to the hematopoietic compartment, we performed bone marrow transplantations (BMT) from WT or *Ovol2*[C120Y/−] mice to lymphocyte-deficient (*Rag2*[−/−]) mice (Fig. 2A, B), as well as BMT from WT mice to either WT (*Ovol2*[+/+]) or *Ovol2*[C120Y/−] mice (Fig. 2E, F). Bone marrow cells from *Ovol2*[C120Y/−] mice repopulated CD4 + , CD8 + , and NK cell populations in the blood of irradiated *Ovol2*[+/+];*Rag2*[−/−] recipients as efficiently as cells derived from WT donors (Fig. 2A, B). In contrast, B cell reconstitution was impaired in *Ovol2*[+/+];*Rag2*[−/−] recipients that received *Ovol2*[C120Y/−] bone marrow (Fig. 2A, B). Compared to *Ovol2*[+/+] recipients of WT bone marrow cells, *Ovol2*[C120Y/−] mice that received WT bone marrow showed severe defects in reconstituting all lymphoid lineages in the blood (Fig. 2E, F).

We examined T cell development in the thymus and B cell development in the bone marrow and spleen of the same recipient

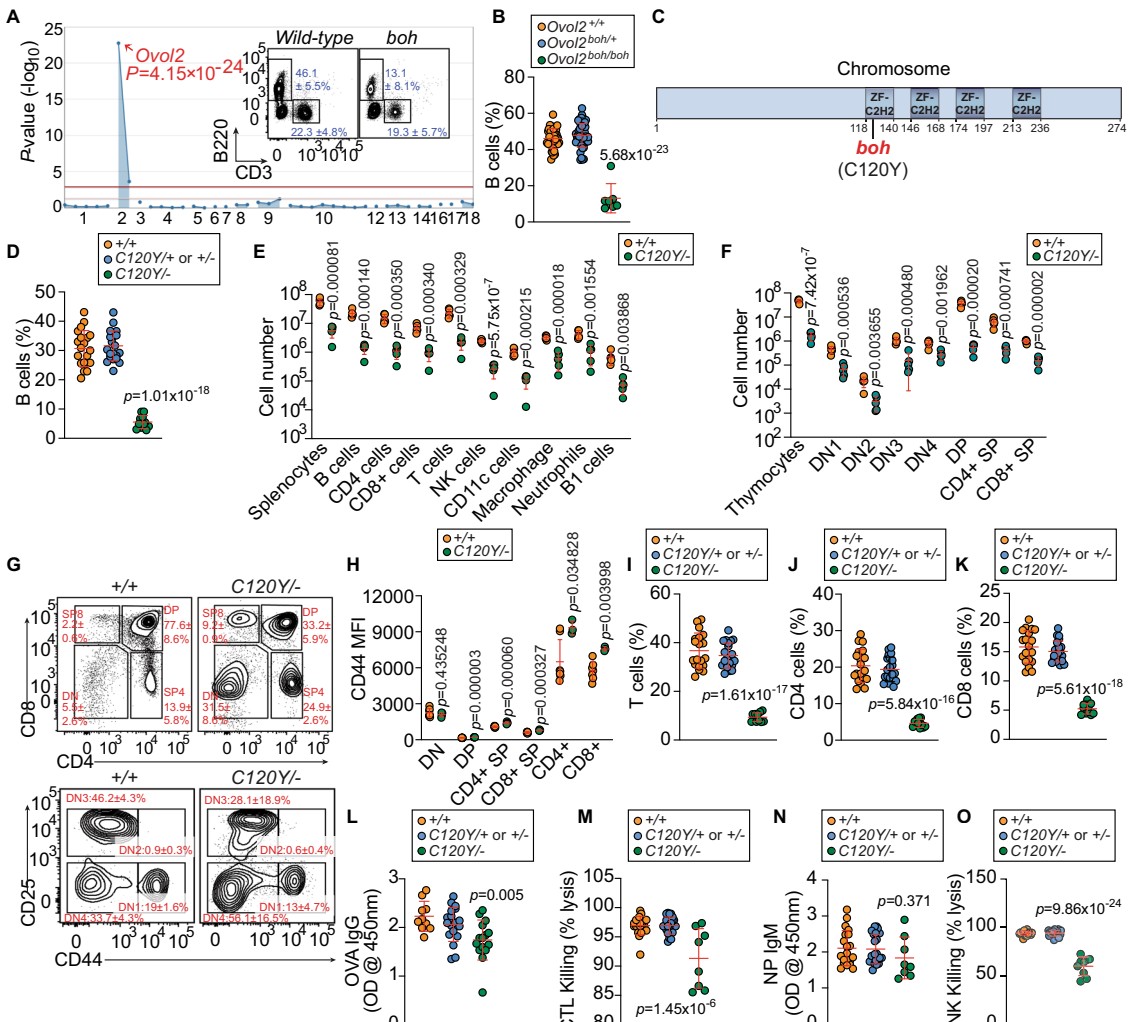

**Fig. 1 | Discovery of an *Ovol2* mutation responsible for a severe impairment of lymphopoiesis. A** Manhattan plot. -Log₁₀ *P*-values plotted *vs.* the chromosomal positions of mutations identified in the G1 founder of the affected pedigree. (Insets) Representative flow cytometry plots of B220+ and CD3+ peripheral blood lymphocytes in wild-type (WT) mice and mice with the *boh* phenotype. Numbers adjacent to outlined regions indicate percent cells in each (mean ± SD) (*n* = 7 *boh/boh*, 36 WT littermates). **B** The frequency of peripheral blood B cells from third generation (G3) descendants of a single ENU-mutagenized male mouse, with REF ( + / + ), HET (+/*boh*), or VAR (*boh/boh*) genotypes for *Ovol2* (*n* = 7 *boh/boh*, 49 *boh/* +, 36 WT littermates). **C** OVOL2 topology. Schematic of OVOL2^boh point mutation which results in substitution of cysteine to tyrosine at position 120 of total 274 amino acids in OVOL2 protein. Numbers are amino acid positions. **D** The frequency of peripheral blood B cells from 12-wk-old *Ovol2^C120Y/–* mice generated by the CRISPR/Cas9 system (*n* = 11 *C120Y/–*, 21 *C120Y/*+ or +/–, 20 WT littermates). **E** Numbers of splenocytes and the indicated cell populations in the spleens of 12-wk-old *Ovol2^C120Y/–* and WT littermates (*n* = 5 mice/genotype). **F–M** Thymocyte developmental defects and T cell functional defects in *Ovol2^C120Y/–* mice. **F, G** Numbers (**F**) and representative flow cytometry plots (**G**) of thymocyte populations in 12-wk-old *Ovol2^C120Y/–* mice and WT littermates. Numbers adjacent to outlined regions indicate percent cells in each (mean ± SD) (*n* = 5 mice/genotype).

(**H**) CD44 expression on thymic and peripheral blood T cells in 12-wk-old *Ovol2^C120Y/–* mice and WT littermates (*n* = 4 (thymus) or 3 (peripheral blood) *C120Y/–* mice, 7 WT littermates). **I–K** Frequency of CD3+ (**I**), CD4+ (**J**), and CD8+ (**K**) T cells in the peripheral blood (*n* = 11 *C120Y/–*, 21 *C120Y/*+ or +/–, 19 WT littermates). **L** Serum ova-specific IgG in mice immunized with T cell-dependent antigen alum-ova (*n* = 15 *C120Y/–*, 19 *C120Y/*+ or +/–, 10 WT littermates). Data presented as absorbance at 450 nm. **M** Quantitative analysis of the βgal-specific cytotoxic T cell killing response in mice immunized with rSFV-βgal (*n* = 8 *C120Y/–*, 22 *C120Y/*+ or +/–, 20 WT littermates). **N** Serum NP-specific IgM in mice immunized with T cell-independent antigen NP-Ficoll (*n* = 8 *C120Y/–*, 22 *C120Y/*+ or +/–, 20 WT littermates). Data presented as absorbance at 450 nm. **O** NK cell cytotoxicity against MHC class I-deficient (*β2m⁻/⁻*) target cells (*n* = 9 *C120Y/–*, 21 *C120Y/*+ or +/–, 20 WT littermates). Data are representative of one experiment (**A, B, M, O**) or two independent experiments (**D, E–L, N**). Data points represent individual mice and means ± SD are indicated. *P*-values were determined by one-way analysis of variance (ANOVA) with Dunnett's multiple comparisons (**B, D, I–O**) or two-tailed Student's *t*-test (**E, F, H**). +/+, *C120Y/*+, and *C120Y/–* indicate WT, *Ovol2^C120Y/*+, and *Ovol2^C120Y/–* genotypes, respectively. MFI Mean fluorescence intensity, DN Double-negative, DP Double-positive, SP Single-positive, CTL Cytotoxic T lymphocyte.

mice analyzed in Fig. 2A, E. We found that *Ovol2^C120Y/–* T cell progenitors in the thymus of irradiated WT recipients differentiated normally through the double-negative (DN) > double-positive (DP) > naïve single-positive (SP) stages (Fig. 2C, D). However, WT T cell progenitors in *Ovol2^C120Y/–* recipients had a severe block in the DN to DP transition (Fig. 2G, H). In the bone marrow, both *Ovol2^C120Y/–* recipients engrafted with WT bone marrow and *Ovol2^+/+;Rag2^–/–* recipients engrafted with *Ovol2^C120Y/–* bone marrow displayed impairment in B cell development

at several stages (Supplementary Fig. 4A, B). Splenic B cell populations were also deficient in *Ovol2^+/+;Rag2^–/–* or *Ovol2^C120Y/–* recipient mice (Supplementary Fig. 4C–F). Furthermore, compared to WT recipients, *Ovol2^C120Y/–* recipient mice had diminished frequencies of CLP (Supplementary Fig. 4G). Together, the BMT experiments strongly suggest that defects extrinsic to the hematopoietic compartment were responsible for the impaired development and reduced frequencies of thymocytes in *Ovol2^C120Y/–* mice, and that both hematopoietic-extrinsic

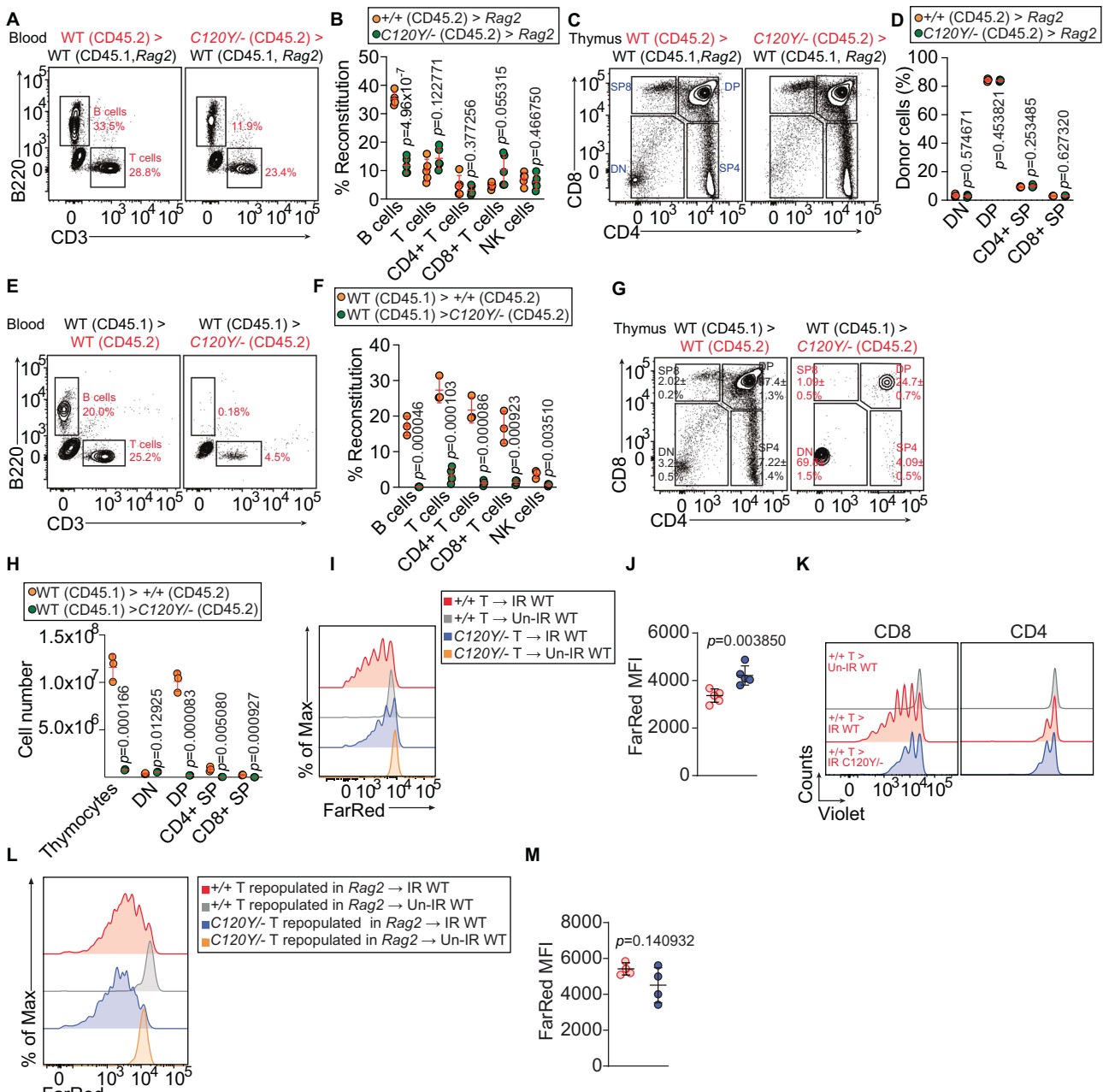

**Fig. 2 | Hematopoietic extrinsic effects of OVOL2 on lymphocyte development.**
**A**–**D** Representative flow cytometry plots (**A**, **C**) and quantitative analysis (**B**, **D**) showing repopulation of lymphocytes in the blood (**A**, **B**) or thymus (**C**, **D**) 12 wk after bone marrow transplantation (BMT) from WT (CD45.2) or *Ovol2^{C120Y}* (CD45.2) mice to lethally irradiated *CD45.1*; *Rag2^{–/–}* recipients. **E**–**H** Representative flow cytometry plots (**E**, **G**) and quantitative analysis (**F**, **H**) showing repopulation of lymphocytes in the blood (**E**, **F**) or thymus (**G**, **H**) 12 wk after BMT from congenic WT (C57BL/6 J; CD45.1) mice to lethally irradiated WT (CD45.2) or *Ovol2^{C120Y}* (CD45.2) recipients. Numbers adjacent to outlined regions indicate representative percentage in each population (**A**, **C**, **E**) or means ± SD (**G**) (*n* = 5 recipients per group in (**A**, **B**), 4 recipients per group in (**C**, **D**), 4 *C120Y/–* recipients or 3 WT recipients in (**E**, **F**), 3 recipients per group in (**G**, **H**)). **I**–**M** Homeostatic expansion of T cells. Representative flow cytometric histograms and frequency of proliferated dye-positive T cells in the spleen of irradiated recipients, 7 d after transfer (*n* = 5

recipients per group in (**I**, **J**), 4 recipients per group in (**L**, **M**)). **I**, **J** CellTrace Far Red-stained pan T cells isolated from the spleen of *Ovol2^{C120Y}* or WT littermates were adoptively transferred into sublethally irradiated (IR, 8.5 Gy) or unirradiated (Un-IR) WT hosts (CD45.1). **J** Mean fluorescence intensity (MFI) of CellTrace Far Red positive-cells. **K** CellTrace Violet-stained pan T cells isolated from the spleen of WT (CD45.1) mice were transferred into sublethally irradiated (8.5 Gy) *Ovol2^{C120Y}* or WT littermates. **L**, **M** WT pan T cells isolated from WT→*Ovol2^{+/+}*;*Rag2^{–/–}* bone marrow chimeras or *Ovol2^{C120Y}* pan T cells isolated from *Ovol2^{C120Y}*→*Ovol2^{+/+}*;*Rag2^{–/–}* bone marrow chimeras were stained by CellTrace Far Red, and were transferred into sublethally irradiated (8.5 Gy) WT recipients (CD45.1). **M** MFI of CellTrace Far Red positive-cells. Data are representative of two independent experiments. Data points represent individual mice; means ± SD are indicated. *P*-values were determined by two-tailed Student's *t* test. +/+ indicates WT and *C120Y/–* indicates the *Ovol2^{C120Y}* genotype.

and -intrinsic defects resulted in impaired development and diminished frequencies of B cells in *Ovol2^{C120Y}* mice. Below, we focus on understanding the extra-hematopoietic function of OVOL2 and how it affects T cell development in thymus.

## Defect in thymic stroma caused by OVOL2 mutation
Based on the BMT data, we suspected that a cell-extrinsic effect of OVOL2 deficiency may impair thymocyte development. To test this, we adoptively transferred mature *Ovol2^{C120Y}* splenic T cells to sublethally

irradiated WT recipient mice, and vice versa; WT splenic T cells transferred to sublethally irradiated WT recipients served as a positive control. Compared to WT T cells in WT recipients, reduced proliferation of *Ovol2*$^{C120Y/-}$ donor T cells was observed in the spleens of WT recipients based on cell tracker dye dilution (Fig. 2I, J), indicating a T cell-intrinsic proliferation defect. Moreover, WT T cells transferred into *Ovol2*$^{C120Y/-}$ hosts also showed significant proliferation defects (Fig. 2K), indicating that T cell-extrinsic proliferation signals are also impaired in *Ovol2*$^{C120Y/-}$ mice. Notably, in secondary transfer experiments, *Ovol2*$^{C120Y/-}$ T cells isolated from bone marrow chimeric mice (generated by BMT from *Ovol2*$^{C120Y/-}$ mice to irradiated *Ovol2*$^{+/+}$;*Rag2*$^{-/-}$ recipients) and transferred to WT (CD45.1) mice, proliferated to a similar extent as WT donor cells (sourced from WT to *Ovol2*$^{+/+}$;*Rag2*$^{-/-}$ bone marrow chimeras) in host spleens (Fig. 2L, M). These data changed our interpretation of the results in Fig. 2I, J. They suggest that *Ovol2*$^{C120Y/-}$ T cells that develop in a WT thymic environment are intrinsically capable of proliferating normally, but that development in the *Ovol2*$^{C120Y/-}$ thymus results in defective mature T cells with impaired responses to proliferation signals. These findings suggest that abnormalities in the *Ovol2*$^{C120Y/-}$ thymic stroma may lead to T cell developmental and functional defects.

## OVOL2 function in TECs is necessary for thymic T cell development

Examination of whole thymi showed that by 12 weeks of age *Ovol2*$^{C120Y/-}$ thymi were consistently much smaller than WT thymi (Fig. 3A). *Ovol2*$^{C120Y/-}$ thymi were surrounded by adipose tissue not observed in WT thymi and had a greatly reduced medullary region (Fig. 3B and Supplementary Fig. 5A, B). These changes progressed with age after 4 weeks when overall thymus size was similar between WT and *Ovol2*$^{C120Y/-}$ mice (Fig. 3A).

Thymic epithelial cells (TECs) are the most abundant cell population within the thymic stroma, and provide developing T cells with cues essential for T cell navigation, proliferation, differentiation and survival[35]. Since our findings suggested thymic stromal defects might exist in *Ovol2*$^{C120Y/-}$ mice (Figs. 2L, M, 3B), we measured the proportions and absolute numbers of cTECs and mTECs in *Ovol2*$^{C120Y/-}$ mice (Supplementary Fig. 6A). At 12 weeks of age, *Ovol2*$^{C120Y/-}$ thymi had increased frequencies of TECs compared to WT thymi, likely a reflection of the reduction in thymocytes in *Ovol2*$^{C120Y/-}$ mice (Fig. 3C). However, mTEC frequencies were drastically reduced and cTEC frequencies were elevated in *Ovol2*$^{C120Y/-}$ thymi relative to those in WT thymi at 12 weeks of age (Fig. 3D). Among those mTECs, the proportions of mTEC subsets (MHC-II$^{high}$ and MHC-II$^{low}$) were normal (Fig. 3E). Enumeration of cells in the thymi of 12-week-old *Ovol2*$^{C120Y/-}$ mice demonstrated significant elevation in non-TEC stromal cells and a reduction in TECs (Fig. 3F) due to a shortage of mTECs, but no change in cTEC numbers (Fig. 3G).

At 4 weeks of age, the ratios of cTECs and mTECs were normal in *Ovol2*$^{C120Y/-}$ mice (Fig. 3H), as were the numbers of TECs and non-TEC stromal cells (Fig. 3I).

To investigate the role of OVOL2 in TECs in vivo, *Ovol2*$^{fl/fl}$ mice that carry a *lox*P-flanked *Ovol2* exon 3 were crossed with *Foxn1-cre* mice to generate *Foxn1-cre;Ovol2*$^{fl/fl}$ mice, in which *Ovol2* is deleted in TECs. Deletion of *Ovol2* exon 3 in TECs from *Foxn1-cre;Ovol2*$^{fl/fl}$ mice, but not *Ovol2*$^{fl/fl}$ littermates was confirmed by PCR (Supplementary Fig. 6B). Compared with *Foxn1-cre;Ovol2*$^{+/+}$ and *Ovol2*$^{fl/fl}$ littermates, thymi from *Foxn1-cre;Ovol2*$^{fl/fl}$ mice exhibited shrinkage and were surrounded by adipose tissue, which progressed with age (Fig. 3J, K, and Supplementary Figs. 5A, B and 6C). All T cell developmental subsets were reduced in numbers in *Foxn1-cre;Ovol2*$^{fl/fl}$ thymi at 20 weeks of age (Fig. 3L). In addition, 20-week-old *Foxn1-cre;Ovol2*$^{fl/fl}$ mice phenocopied *Ovol2*$^{C120Y/-}$ mice in exhibiting increased percentages of DN T cells, decreased percentages of DP T cells (Fig. 3M), and an activated state (CD44$^{high}$) in DP, SP4, and SP8 T cells (Fig. 3N). *Foxn1-cre;Ovol2*$^{fl/fl}$ mice

had a defect in CD8 + T cell cytotoxic function after challenge with alum-ova (Fig. 3O). Moreover, compared to *Ovol2*$^{fl/fl}$ littermates, mTEC frequencies were drastically reduced and cTEC frequencies were elevated in *Foxn1-cre;Ovol2*$^{fl/fl}$ thymi at 20 weeks of age (Fig. 3P). Overall, except for later development of thymus hypoplasia (Fig. 3J), in all other aspects of the thymus phenotype *Foxn1-cre;Ovol2*$^{fl/fl}$ mice phenocopied *Ovol2*$^{C120Y/-}$ mice. These data demonstrate that OVOL2 deficiency impacts T cell development in the thymus through an effect on TECs.

To understand whether developmentally regulated *Ovol2* expression might contribute to the age-dependency of thymus hypoplasia and reduced frequency and number of mTEC, we tracked *Ovol2* gene expression in TEC subsets at several ages during embryogenesis and postnatally (Supplementary Fig. 6D). In cTECs, the relative mRNA level of *Ovol2* was lowest embryonically during thymic organogenesis and increased with age until it peaked at 12 weeks of age (Fig. 4A). In mTECs, relative *Ovol2* transcript expression was low at E13.5 and P0, peaked at 4 weeks of age, and returned to E13.5/P0 levels when tested at 12, 20, and 30 weeks of age (Fig. 4A). When we compared *Ovol2* mRNA levels in cTECs vs. mTECs, we observed that mTECs and cTECs had comparable *Ovol2* expression at E13.5, but cTECs displayed higher *Ovol2* transcript levels than mTECs postnatally (Fig. 4B). The progressive increase in *Ovol2* expression between E13.5 and 4 weeks (in mTECs) or E13.5 and 12 weeks (in cTECs) is consistent with the age dependent phenotypes of TECs in OVOL2 mutant mice. These data also imply an important role for OVOL2 in the maintenance of cTEC function despite a lack of effect on cTEC numbers.

In addition to TECs, non-TEC stromal cells, such as mesenchymal and endothelial cells, participate in thymus organogenesis and contribute to T cell development[36,37]. We therefore investigated the frequencies and numbers of thymic endothelial cells and mesenchymal cells in *Ovol2*$^{C120Y/-}$ mice (at 4 weeks and 12 weeks of age) and in *Foxn1-cre;Ovol2*$^{fl/fl}$ mice (at 20 weeks of age). Four-week-old *Ovol2*$^{C120Y/-}$ mice had normal frequencies (Supplementary Fig. 7E–H) and numbers (Fig. 4C) of endothelial and mesenchymal cell populations in the thymus, when TECs were also normal in number. At 12 weeks of age, *Ovol2*$^{C102Y/-}$ thymi had drastically increased frequencies (Supplementary Fig. 7A–D) and numbers (Fig. 4H) of endothelial cells and mesenchymal cells. Importantly, 20-week-old *Foxn1-cre;Ovol2*$^{fl/fl}$ mice also exhibited increased proportions (Supplementary Fig. 7I–L) and numbers (Fig. 4M) of mesenchymal cells, but endothelial cells were normal. These data strongly suggest that TEC-specific OVOL2 deficiency results in both fractional and numerical increases in thymic mesenchymal cells at the expense of epithelial cells. In contrast, the increase in endothelial cells in *Ovol2*$^{C120Y/-}$ thymi is a consequence of OVOL2 deficiency in cell type(s) other than TECs.

## Functionality of the thymic stroma: defective recruitment of T cell progenitors and impaired positive and negative T cell selection, but no autoimmunity

We observed numerical reductions of early T cell progenitors (ETP) in thymi from 12-week-old *Ovol2*$^{C120Y/-}$ mice and 20-week-old *Foxn1-cre;Ovol2*$^{fl/fl}$ mice relative to WT mice (Fig. 4I, N). The frequencies of these ETP were elevated, reflecting their impaired development to the DN stage (Supplementary Fig. 8A–D). Consistent with these findings, we found diminished transcript levels of *Ccl19* and *Ccl21*, mTEC-expressed chemokines that recruit CLP to the thymus[38], in mTEC from 12-week-old *Ovol2*$^{C120Y/-}$ mice (Fig. 4J) and 20-week-old *Foxn1-cre;Ovol2*$^{fl/fl}$ mice (Fig. 4O). An endothelial cell deficiency in the transcript expression levels of *P-selectin* and *Ccl25*, molecules expressed by the thymic endothelium that promote ETP settling[39], was observed in thymic endothelial cells only from 12-week-old *Ovol2*$^{C120Y/-}$ mice (Fig. 4K), but not 20-week-old *Foxn1-cre;Ovol2*$^{fl/fl}$ mice (Fig. 4P). These data suggest a mechanistic basis for the reduction of ETP in thymi from OVOL2 mutant mice, namely the failure of thymic stromal cells to express critical recruiting molecules. In addition, TEC-specific OVOL2

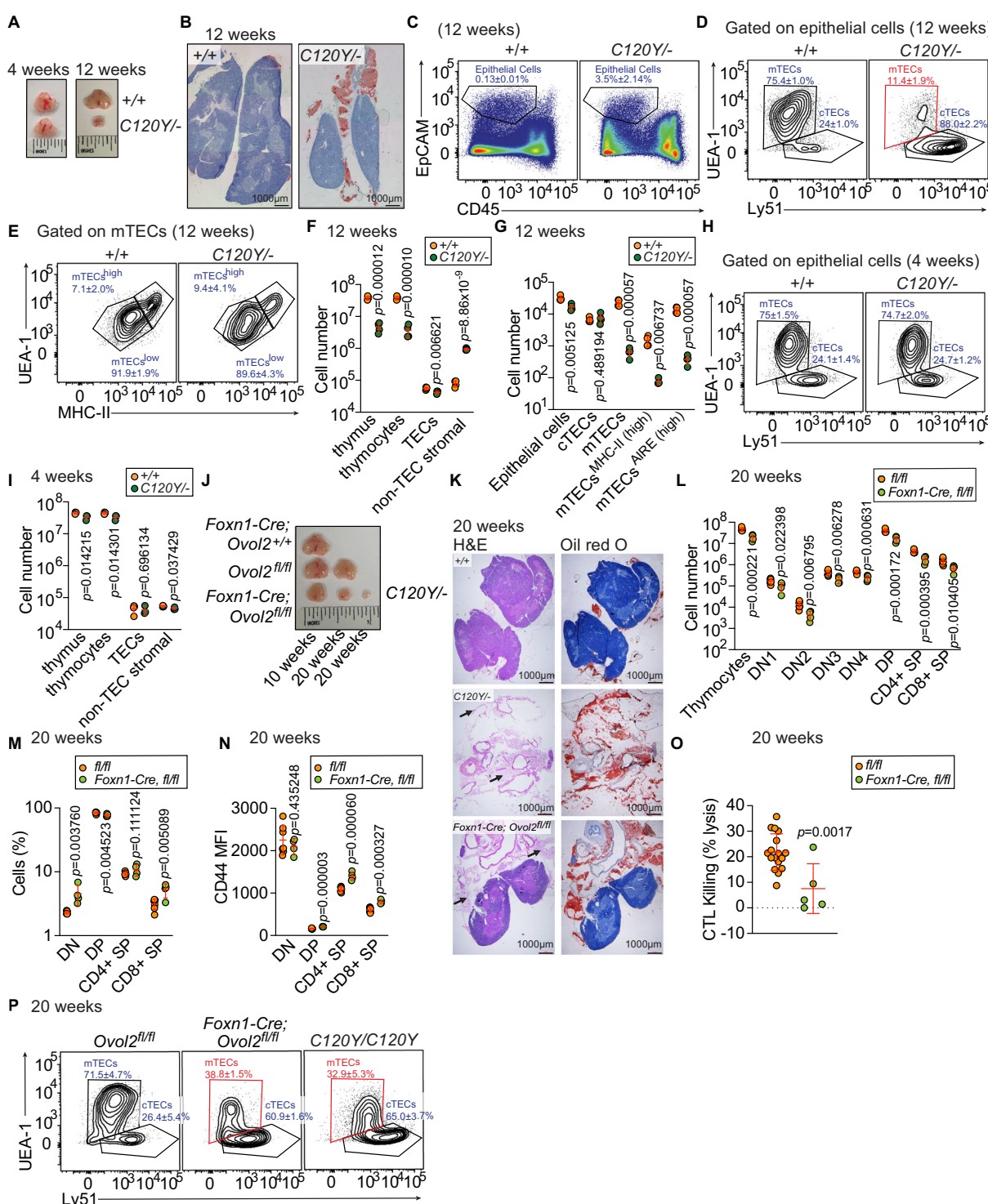

deletion altered recruiting molecule expression in TECs but not in endothelial cells.

By comparison, 4-week-old $Ovol2^{C120Y/-}$ mice had only slightly reduced numbers of ETP (Fig. 4D), and their frequencies were similar to those of WT mice at this age (Supplementary Fig. 8E, F). *Ccl19* displayed decreased expression in 4-week-old $Ovol2^{C120Y/-}$ mTECs (Fig. 4E), although *Ccl21*, *P-selectin*, and *Ccl25* expression were normal (Fig. 4E, F). These findings suggest that deficient expression of CCL19 may be one of the first mTEC malfunctions encountered by ETP, preceding deficiencies of endothelial cell P-selectin and CCL25 expression in $Ovol2^{C120Y/-}$ mice.

Consistent with TEC hypoplasia and functional defects, we found decreased percentages of T cells undergoing positive selection in 12-week-old $Ovol2^{C120Y/-}$ mice, with milder impairments observed in 4-week-old $Ovol2^{C120Y/-}$ mice (Supplementary Fig. 9A–F). 20-week-old $Foxn1-cre;Ovol2^{fl/fl}$ mice phenocopied $Ovol2^{C120Y/-}$ mice (Supplementary Fig. 9G–I). These impairments were consistent with the diminished expression of genes necessary for positive selection including *Psmb11*, *Cathepsin L*, and *Prss16*, and genes essential for thymocyte progenitor development toward mature T cells including *Dll4*, *Il-7*, *Scf*, and *Cxcl12*, in cTECs from 12-week-old and 4-week-old $Ovol2^{C120Y/-}$ mice (with the exception of *Psmb11* in 4-week-olds) (Fig. 4G, L)[40]. MHC class II protein

**Fig. 3 | OVOL2 function in TECs is necessary to support thymic T cell development.** **A** Representative photographs of thymi from 4- or 12-wk-old *Ovol2^C120Y/−^* and WT littermates. **B** Representative Oil red O staining of *Ovol2^C120Y/−^* and WT littermate thymi at 12 wk of age. **C–I** Representative flow cytometry plots of TECs (**C–E, H**) and quantitative analysis (**F, G, I**) of indicated subsets in the thymus (**F, I**) or TECs (**G**) from 12-wk-old (**C–G**) or 4-wk-old (**H, I**) *Ovol2^C120Y/−^* mice and WT littermates. Numbers adjacent to outlined regions indicate percent cells in each (mean ± SD) (*n* = 3 *C120Y/−*, 4 WT littermates in (**C–E**); 4 mice/genotype in (**F**); 4 (epithelial, cTEC, mTEC, mTEC^AIRE(high)^) or 3 (mTEC^MHC-II(high)^) *C120Y/−* mice, 5 (cTEC, mTEC, mTEC^AIRE(high)^) or 4 (epithelial, mTEC^MHC-II(high)^) WT littermates in (**G**); 3 mice/genotype in H; 3 *C120Y/−*, 4 WT littermates in (**I**)). **J** Representative photographs of thymi from 10- or 20-wk-old *Foxn1-Cre;Ovol2^fl/fl^* mice and their *Ovol2^fl/fl^* or *Foxn1-Cre;Ovol2^+/+^* littermates, and age matched *Ovol2^C120Y/−^* mice. (**K**) Representative hematoxylin and eosin (H&E) (left) and Oil red O (right) staining of *Ovol2^C120Y/−^*, *Foxn1-Cre;Ovol2^fl/fl^*, or age matched WT thymi at 20 wk of age. Black arrows indicate adipose tissue. **L, M** Numbers (**L**) and frequencies (**M**) of thymocytes in *Foxn1-Cre,Ovol2^fl/fl^* and *Ovol2^fl/fl^* littermates at 20 weeks of age (*n* = 4 *Foxn1-Cre,Ovol2^fl/fl^*, 7 *Ovol2^fl/fl^* littermates). **N** Surface marker CD44 expression by thymocytes in 20-wk-old *Foxn1-Cre;Ovol2^fl/fl^* and *Ovol2^fl/fl^* littermates (*n* = 4 *Foxn1-Cre,Ovol2^fl/fl^*, 7 *Ovol2^fl/fl^* littermates). **O** Quantitative analysis of the ova-specific cytotoxic T cell killing response in 20-wk-old *Foxn1-Cre;Ovol2^fl/fl^* mice and WT littermates immunized with alum-ova (*n* = 5 *Foxn1-Cre,Ovol2^fl/fl^*, 17 *Ovol2^fl/fl^* littermates). **P** Representative flow cytometry plots of TECs from 20-wk-old *Foxn1-Cre;Ovol2^fl/fl^* mice and their *Ovol2^fl/fl^* littermates, and age matched *Ovol2^C120Y/C120Y^* mice. Numbers adjacent to outlined regions indicate percent cells in each (mean ± SD) (*n* = 5 *C120Y/C120Y* mice, 4 *Foxn1-Cre,Ovol2^fl/fl^* mice, 6 age matched WT mice). Data are representative of two independent experiments (**F, G, I, L–O**). Data points represent individual mice (**F, G, I, L–O**) and means ± SD are indicated. *P*-values were determined by two-tailed Student's *t*-test (**F, G, I, L–O**). MFI Mean fluorescence intensity. +/+, WT; *C120Y/−*, *Ovol2^C120Y/−^*; *C120Y/C102Y, Ovol2^C102Y/C120Y^*; and *fl/fl, Ovol2^fl/fl^*.

expression on the surface of *Ovol2^C120Y/−^* cTECs was reduced, although MHC class I protein expression was normal (Fig. 4Q, R). These data suggest that cTECs in *Ovol2^C120Y/−^* mice are functionally impaired in their ability to promote T cell specification and positive selection resulting in defective development of ETP to DP T cells.

T cell negative selection (stages defined as in ref. 41) was also altered in OVOL2 mutant mice, in that greater frequencies of T cells were present at each stage of negative selection in 12-week-old *Ovol2^C120Y/−^* thymi compared to WT thymi (Supplementary Fig. 10A). Smaller increases in the frequencies of T cells at various stages of negative selection were detected in 4-week-old *Ovol2^C120Y/−^* mice and in 20-week-old *Foxn1-cre;Ovol2^fl/fl^* mice (Supplementary Fig. 10B, C). The mRNA levels of gene accessibility regulator *Aire* and zinc-finger transcription repressor *Fezf2*, necessary for expression of tissue-restricted antigens (TRA)[7], were reduced in *Ovol2^C120Y/C120Y^* mTECs (Fig. 4S). Protein levels of AIRE (intracellular), CD80, and CD86 (on the cell surface) were also reduced in mTECs from *Ovol2^C120Y/C120Y^* mice (Fig. 4T–V). However, AIRE protein levels were normal in mature mTECs (UEA1 + , MHC-II^high^) from *Ovol2^C120Y/C120Y^* mice (Fig. 4T), which was consistent with the preserved expression of several AIRE or FEZF2 dependent tissue-restricted self-antigens (TRAs), including *Tff3, Ins2, Mup1, Amy2a*, and *Afp* in mature mTECs from 12-week-old *Ovol2^C120Y/−^* mice (with the exception of *Ttr*) (Supplementary Fig. 11A, B).

Since our findings demonstrated altered negative selection, we analyzed if *Ovol2^C120Y/C120Y^* mice displayed signs of autoimmunity. *Ovol2^C120Y/C120Y^* mice displayed normal antibody titers (Supplementary Fig. 11C). *Ovol2^C120Y/C120Y^* mice had severe lymph node hypoplasia (Supplementary Fig. 11D). *Ovol2^C120Y/C120Y^* mice had normal percentages of CD4+FoxP3+ Treg cells in the lymph node and spleen, although they had decreased numbers of these cells because of lymph node and spleen hypoplasia (Supplementary Fig. 11E, F). Also, there were no signs of lymphocytic infiltration into the kidney, lung, liver, intestine, and salivary gland (Supplementary Fig. 11G–K). These data all support the absence of autoimmunity in *Ovol2^C120Y/C120Y^* mice. Normal AIRE protein levels in mature mTECs may support efficient T cell negative selection.

**Aberrant TEC transition toward mesenchymal state**
We found that thymic mesenchymal cells were augmented at the cost of epithelial cells in mice with TEC-specific OVOL2 deficiency (Fig. 4M and Supplementary Fig. 7I–L). OVOL2 is a repressor of the epithelial to mesenchymal transition (EMT)[42]. To investigate whether the *C120Y* allele affects EMT status, expression of four epithelial genes (*Cdh4, Krt18, Cldn12*, and *Cxadr*) and a mesenchymal gene (*Snai1*) was examined in *Ovol2^C120Y/C120Y^* cTECs and mTECs from mice at 4.5 weeks of age. The mRNA level of *Cdh4* was reduced in *Ovol2^C120Y/C120Y^* cTECs to ~50% of the level in WT cells, while in mTECs it was reduced to ~15% of WT levels (Fig. 5A, B). The other three epithelial genes in *Ovol2^C120Y/C120Y^*

cTECs and mTECs were reduced to ~50% of levels in the corresponding WT cells (Fig. 5A, B). In contrast to epithelial genes, *Ovol2^C120Y/C120Y^* cTECs and mTECs showed ~7-fold and 15-fold more *Snai1* expression, respectively, compared to WT cells (Fig. 5A, B). The aberrant expression of these genes suggests a transition by both mTECs and cTECs from epithelial to mesenchymal status.

To establish that mesenchymal cells originated from TECs in *Ovol2^C120Y/C120Y^* thymi, we generated mice in which developing TECs were marked with the fluorescent reporter tdTomato. These mice (*Ovol2^C120Y/C120Y^;Foxn1-Cre;Ai9*) express Ai9, a Cre reporter knocked into the *Rosa26* locus, which contains the *tdTomato* gene separated from a CAG promoter by a loxP-flanked STOP cassette. CAG promoter-driven expression of tdTomato is activated by Cre-mediated deletion of the STOP cassette. Driven by the *Foxn1* promoter, Cre expression activates tdTomato fluorescence in TEC progenitors. We analyzed the *tdtomato^high^* TEC population (see Supplementary Fig. 12 for gating strategy) and observed that ~80% of these cells in WT thymi were epithelial cells, whereas only ~26% were epithelial cells in *Ovol2^C120Y/C120Y^* thymi from 20-week-old mice (Fig. 5C–F). Moreover, ~50% of *Ovol2^C120Y/C120Y^* *tdtomato^high^* cells were CD31-EpCAM-PDGFR-α/β+ cells (i.e., mesenchymal cells), in comparison with only ~4% of WT *tdtomato^high^* cells (Fig. 5C–F). Endothelial cell frequencies among *tdtomato^high^* cells were slightly elevated in *Ovol2^C120Y/C120Y^* thymi compared to WT thymi (Fig. 5C–F). These data demonstrate that in *Ovol2^C120Y/C120Y^* thymi, TEC were converted to mesenchymal cells, which was also corroborated by analysis of epithelial and mesenchymal gene expression (Fig. 5G).

**OVOL2 interacts with and inhibits BRAF-HDAC complex activity**
Although its role as a negative regulator of EMT has been well established through the identification of several mesenchymal genes repressed by OVOL2, how OVOL2 mediates EMT repression is not known. To gain insight into the mechanism, we identified OVOL2-interacting proteins using co-immunoprecipitation (co-IP) combined with mass spectrometry (MS) analysis. Three independent MS analyses using immunoprecipitates prepared from EL4 T cells stably expressing FLAG-tagged OVOL2 with or without DNase I identified 351, 168, and 289 candidate proteins as putative OVOL2 interactors (Fig. 6A, B, and Supplementary Dataset 1). Among these candidates, 24 proteins were precipitated with OVOL2 in all three experiments, but not with the empty vector control (Fig. 6B and Supplementary Dataset 1).

The first four priority candidates were essential components of the BRAF-HDAC (BHC) complex, including lysine-specific demethylase 1 (LSD1/KDM1A), repressor element-1 silencing transcription factor (REST) corepressor 1 (RCOR1/CoREST), high mobility group 20 A (HMG20A), and genetic suppressor element 1 (GSE1) (Fig. 6B and Supplementary Dataset 1). We confirmed that BHC complex components including LSD1, RCOR1, HDAC1, HDAC2, HMG20A, HMG20B, and PHF21A interact with OVOL2 by co-IP from human embryonic

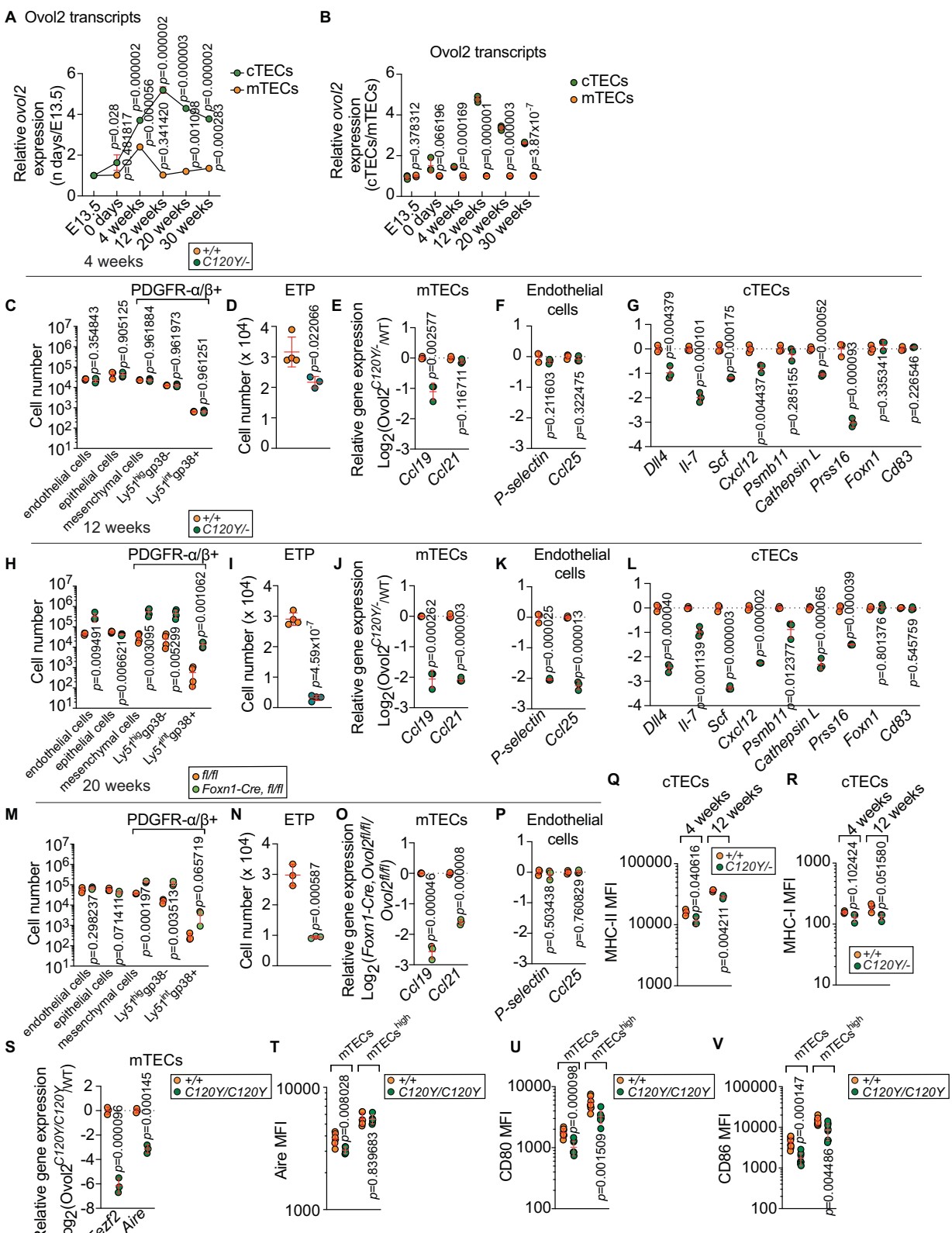

kidney 293 T (HEK293T) cells after transient transfections (Fig. 6C). We noted OVOL2 strongly interacted with LSD1, and weakly interacted with RCOR1 and PHF21A (Fig. 6C). Surprisingly, OVOL2$^{C120Y}$ had an interaction profile comparable to OVOL2$^{WT}$ (Fig. 6C). Based on the elution volume in size exclusion chromatography experiments, OVOL2$^{WT}$ forms a dimer; in contrast, OVOL2$^{C120Y}$ forms aggregates of soluble oligomers (Supplementary Fig. 13A) and the T$_m$ of OVOL2$^{C120Y}$

was 7 degrees lower than that of OVOL2$^{WT}$, indicating that the 1-aa replacement caused by the *C120Y* mutation reduces the stability of OVOL2 (Supplementary Fig. 13B).

We generated a CRISPR-based knock-in of a triple FLAG-tag appended to the C-terminus of endogenous OVOL2 in mice. OVOL2 was expressed in pancreas, testis, cortex region of brain, and TECs (Supplementary Fig. 13C–E, red arrow). A shorter splice isoform of

**Fig. 4 | Impaired functionality of the thymic stroma in OVOL2 deficient thymi.**
**A, B** Real time-PCR analysis of *Ovol2* transcripts from FACS sorted *Ovol2*⁺/⁺ cTECs and mTECs at E13.5 days, 0 days, 4 weeks, 12 weeks, 20 weeks, and 30 weeks of age (*n* = 3 per group). Mean and SD are shown in (**A**). The data are normalized to mean expression at E13.5 days (**A**) or to mean expression in mTECs (**B**). **C, H, M** Numbers of indicated subpopulations of thymic stromal cells in 4-wk-old (**C**), or 12-wk-old *Ovol2*^*C120Y/–* mice (**H**), or 20-wk-old *Foxn1-Cre;Ovol2*^*fl/fl* mice (**M**) and WT or *Ovol2*^*fl/fl* littermates (*n* = 4 *C120Y/−*, 3 WT littermates in (**C**); 4 mice/genotype in (**H**); 3 mice/genotype in (**M**). Gating strategy is shown in Supplementary Fig. 7. **D, I, N** Quantitative analysis of early T-cell progenitors (ETP) in 4-wk-old (**D**), or 12-wk-old *Ovol2*^*C120Y/–* thymi (**I**), or 20-wk-old *Foxn1-Cre;Ovol2*^*fl/fl* thymi (**N**) and WT thymi (*n* = 3 *C120Y/−*, 4 WT littermates in (**D**); 4 mice/genotype in (**I**); 3 mice/genotype in (**N**).
**E, F, J, K, O, P** Real time-PCR analysis of the indicated genes in FACS sorted mTECs (**E, J, O**) and endothelial cells (**F, K, P**) from 4-wk-old (**E, F**), or 12-wk-old *Ovol2*^*C120Y/–* thymi (**J, K**), or 20-wk-old *Foxn1-Cre;Ovol2*^*fl/fl* thymi (**O, P**) and WT thymi (*n* = 3 per

group). **G, L** Real time-PCR analysis of the indicated genes in FACS sorted cTECs from 4-wk-old (**G**), or 12-wk-old *Ovol2*^*C120Y/–* thymi (**L**) and WT littermates (*n* = 3 per group). **Q, R** Surface marker MHC-II (**Q**) and MHC-I (**R**) expression by cTECs in 4- or 12-wk-old *Ovol2*^*C120Y/–* mice and WT littermates (*n* = 3 mice/genotype). **S** Real time-PCR analysis of *Aire* and *Fezf2* from FACS sorted *Ovol2*⁺/⁺ and *Ovol2*^*C120Y/C120Y* mTECs at 4.5 weeks of age (*n* = 3 per group). **T** Intracellular AIRE intensity in total or MHC-II^high mTECs of 4.5-wk-old *Ovol2*^*C120Y/C120Y* and WT littermates (*n* = 5 *C120Y/C120Y*, 6 WT littermates). **U, V** Surface marker CD80 (**U**) and CD86 (**V**) expression by total or MHC-II^high mTECs in 4.5-wk-old *Ovol2*^*C120Y/C120Y* mice and WT littermates (*n* = 8 mice/genotype). Data are representative of two (**A–R, T–V**) or four independent experiments (**S**). Data points represent individual mice (**C, D, H, I, M, N, Q, R, T–V**) or individual sorting pools from 5 mice per pool (**A, B, E–G, J–L, O, P, S**). Means ± SD are indicated and *P* values were determined by two-tailed Student's *t* test. MFI, mean fluorescence intensity. +/+, WT; *C120Y/−, Ovol2*^*C120Y/–*; *C120Y/C102Y, Ovol2*^*C102Y/C120Y*; and *fl/fl, Ovol2*^*fl/fl*.

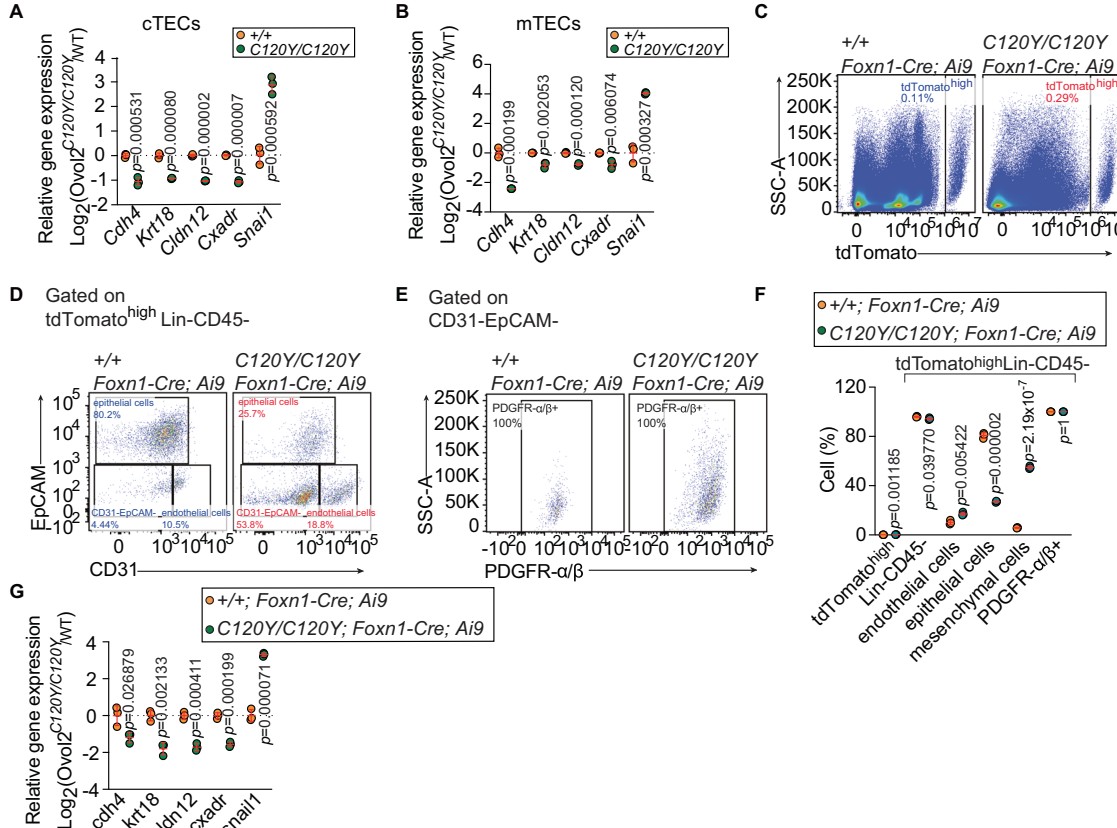

**Fig. 5 | *Ovol2*^*C120Y/C120Y* TECs displayed transition toward mesenchymal state.**
**A, B** Real time-PCR analysis of the indicated genes in FACS sorted *Ovol2*⁺/⁺ and *Ovol2*^*C120Y/C120Y* cTECs (**A**) and mTECs (**B**) at 4.5 weeks of age (*n* = 3 per group). **C–F** Lineage tracing using tdTomato expression in Foxn1+ cells in 20-wk-old *Ovol2*^*C102Y/C120Y*;*Foxn1-Cre;Ai9* and *Ovol2*⁺/⁺;*Foxn1-Cre;Ai9* littermates (*n* = 3 mice/genotype). **C** Representative flow cytometry plots showing side scatter profile and tdTomato expression by cells from thymi of the indicated mice. **D** Representative flow cytometry plots of tdTomato^high cells, gated on Lin-CD45- cells, from (**C**). **E** Representative flow cytometry plots of tdTomato^highCD31-EpCAM- cells from (**D**). Each subset was gated as follows: endothelial cells: EpCAM⁻CD31⁺, epithelial cells:

EpCAM⁺CD31⁻, mesenchymal cells: EpCAM⁻CD31⁻PDGFR-α/β⁺ (see also Supplementary Fig. 12). Numbers adjacent or inside outlined regions indicate representative percentage in each population (*n* = 3 mice/genotype). **F** Quantitative analysis of the indicated tdTomato^high cell populations. **G** Real time-PCR analysis of the indicated genes in FACS sorted *Ovol2*⁺/⁺ and *Ovol2*^*C120Y/C120Y* tdTomato^high cells in thymi at 20 weeks of age (*n* = 3 per group). Data are representative of two (**C–G**) or four independent experiments (**A, B**). Data points represent individual mice (**F**) or individual sorting pools from 5 mice per pool (**A, B, G**). Means ± SD are indicated and *P* values were determined by two-tailed Student's *t*-test. +/+, WT; and *C120Y/C102Y, Ovol2*^*C102Y/C120Y*.

OVOL2 (isoform B) is the major form in the testis (Supplementary Fig. 13C, red arrow). Immunoblot analysis of immunoprecipitates from primary TECs confirmed specific recruitment of endogenous LSD1, RCOR1, HDAC1, and HDAC2 by FLAG-tagged OVOL2, but not by the B6 control (Fig. 6D). Further, mixing purified OVOL2 and LSD1 at a 1:1 ratio (7 μM) resulted in a shift of the peak profile indicative of a larger complex compared to the individual proteins in size

exclusion chromatography, suggesting OVOL2 directly interacts with LSD1 (Fig. 6E and Supplementary Fig. 13F). In contrast, OVOL2 did not directly interact with RCOR1 (Fig. 6F and Supplementary Fig. 13F). These data indicate that OVOL2 interacts with the BHC complex.

LSD1, a core component of the BHC complex, is a flavin-dependent monoamine oxidase, which can demethylate mono- and

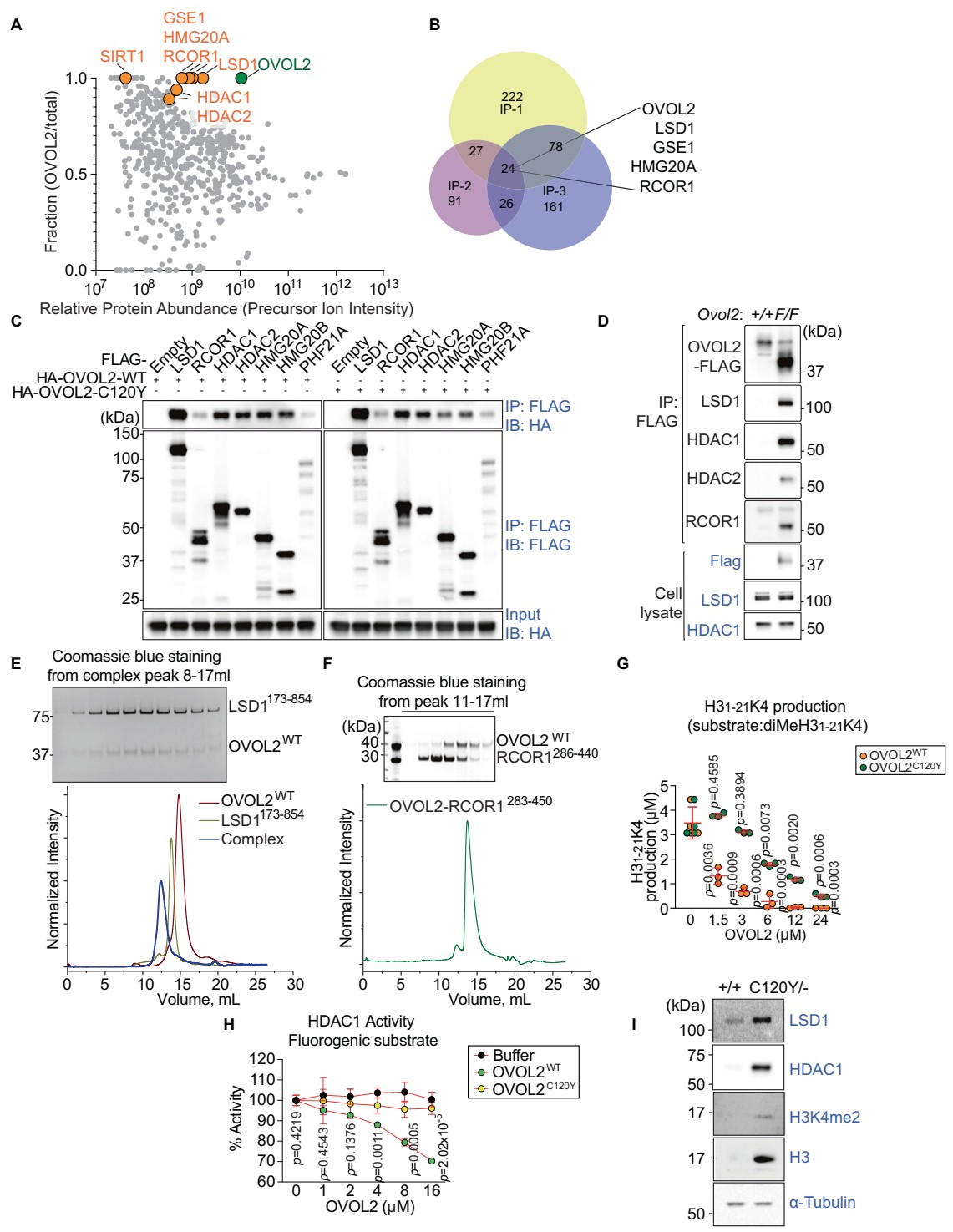

di-methylated lysines, specifically histone 3, lysines 4 and 9 (H3K4 and H3K9)[43,44]. LSD1 is an epigenetic regulator that represses epithelial genes by removing methyl groups from H3K4[45]. LSD1 regulates cell fate determination during embryonic development in mice and controls stem cell identity and cell differentiation of multiple cell types[46–48]. An in vitro LSD1 demethylation activity assay demonstrated that 1.5 μM OVOL2$^{WT}$ reduced LSD1 activity to -38% of the activity of LSD1 (300 nM) alone (Fig. 6G). 12 μM OVOL2$^{C120Y}$ showed comparable inhibition to 1.5 μM OVOL2$^{WT}$, indicating that the *C120Y* mutation reduced OVOL2 inhibitory activity towards LSD1 by 87.5% (Fig. 6G). We also investigated the effect of OVOL2 on demethylation of diMeH3$_{1-21}$K9, however, no H3K9 production was detected

(Supplementary Fig. 14A), consistent with a previous report[49]. We found that high concentrations of OVOL2$^{WT}$, but not OVOL2$^{C120Y}$, reduced the deacetylation activity of HDAC1 by 30% (Fig. 6H). In addition to the impaired ability of OVOL2$^{C120Y}$ to inhibit LSD1 demethylation and HDAC1 deacetylase activities, we observed that LSD1, HDAC1, and H3 expression were significantly higher in *Ovol2*$^{C120Y/-}$ TECs than in WT TECs (Fig. 6I; most TECs are mTECs at this age, Supplementary Fig. 6D). We also noted a mildly increased level of H3K4me2 in *Ovol2*$^{C120Y/-}$ TECs (Fig. 6I). The ratio of H3K4me2 and H3 implies hyperactive BHC mediated histone demethylation in *Ovol2*$^{C120Y/-}$ TECs. These data indicate that OVOL2 negatively regulates BHC complex activity in vitro.

**Fig. 6 | OVOL2 interacts with and negatively regulates the BRAF-HDAC (BHC) complex. A–F** OVOL2 directly interacts with LSD1, but not RCOR1. **A** Interactors of OVOL2 identified by co-immunoprecipitation (co-IP) combined with LC-MS/MS. Data shown are combined from three independent MS analyses of IPs prepared from EL4 T cells stably expressing FLAG-tagged OVOL2. Multiple components of the BHC complex, including LSD1, GSE1, HMG20A, RCOR1, HDAC1, HDAC2, and SIRT1 were identified. Relative protein abundance calculated using precursor ion intensities (abundance in OVOL2 IP ÷ sum of abundances in OVOL2 IP plus control IP) is plotted on the Y-axis. Y = 0.5 indicates equivalent abundance in the OVOL2 IP and control IP. Y > 0.5 indicates enrichment in the OVOL2 IP sample with Y = 1 indicating the protein was exclusively detected in the OVOL2 IP. **B** Venn diagram showing numbers of OVOL2 interactors identified by MS in the three co-IP experiments, of which 24 proteins including OVOL2, LSD1, GSE1, HMG20A, and RCOR1 were identified in all three experiments. **C** HEK293T cells were transfected with either FLAG-tagged LSD1, RCOR1, HDAC1, HDAC2, HMG20A, HMG20B, PHF21A, or empty vector and HA-tagged OVOL2. Lysates were subsequently immunoprecipitated using anti-FLAG M2 agarose and immunoblotted with antibodies against HA or FLAG. **D** Endogenous FLAG-tagged OVOL2 was immunoprecipitated from lysates of primary TECs of *Ovol2*-Flag knock-in mice and

immunoblotted using Flag, LSD1, HDAC1, RCOR1, or HDAC2 antibodies. Immunoblot analysis of cell lysates (lower panels). **E, F** Purified OVOL2 mixed with either LSD1 (**E**) or RCOR1 (**F**) was analyzed by size exclusion chromatography to assess complex formation. **G–I** OVOL2 inhibits LSD1 and HDAC1 activity. **G** LSD1 demethylation activity assay. Demethylation activity of purified LSD1 (300 nM) towards 60 μM dimethylated H3$_{1-21}$K4 (diMeH3$_{1-21}$K4) substrate was tested in the presence or absence of OVOL2$^{WT}$ or OVOL2$^{C120Y}$ at the indicated concentrations ($n = 4$ wells without OVOL2, 3 wells per diluted OVOL2 condition). **H** HDAC1 activity assay. HDAC1 deacetylation activity was determined in the presence or absence of OVOL2$^{WT}$ or OVOL2$^{C120Y}$ at the indicated concentrations. Means ± SD are shown ($n = 3$ wells per condition). **I** Immunoblot analysis of LSD1, HDAC1, dimethylated H3K4 (H3K4me2), H3, and α-tubulin in total cell lysates of FACS sorted TECs from 4.5-wk-old *Ovol2$^{C120Y/-}$* or WT littermates. Data are representative of three (**A–F, I**) or five (**G, H**) independent experiments. Data points represent individual wells and means ± SD are indicated (**G**). *P*-values were determined by two-tailed Student's *t*-test (**G**) or one-way analysis of variance (ANOVA) with Dunnett's multiple comparisons (**H**). +/+, *F/F*, and *C120Y/–* indicate WT, *Ovol2$^{3xFlag/3xFlag}$*, and *Ovol2$^{C120Y/-}$* genotypes, respectively.

## OVOL2 competes with RCOR1 for binding to LSD1

The essential BHC complex component RCOR1 promotes the demethylase activity of LSD1 towards core histones and nucleosomal substrates, protects LSD1 from proteasomal degradation in vitro, enhances the interaction of LSD1 with SNAI1, and promotes the stability of the SNAI1-LSD1-RCOR1 ternary complex resulting in enhanced EMT[50,51]. Among the BHC complex components, we found that OVOL2 interacted most strongly with LSD1. Therefore, to understand the effect of OVOL2 on LSD1-RCOR1 interaction, we incubated LSD1-Flag/RCOR1-HA complexes purified by gel filtration (to exclude unbound proteins) with increasing concentrations of purified OVOL2$^{WT}$, followed by precipitation using anti-FLAG M2 agarose. OVOL2$^{WT}$ effectively dissociated the LSD1-Flag/RCOR1-HA complex and recruited LSD1 (Fig. 7A). Size exclusion chromatography showed that OVOL2$^{WT}$ (7 μM) did not interact with pre-formed LSD1-RCOR1 complex (7 μM) (Supplementary Fig. 14B) and dissociated a portion of RCOR1 from LSD1 (Fig. 7B). Furthermore, whereas 6 μM OVOL2$^{WT}$ reduced LSD1 demethylase activity to ~10% of that of LSD1 alone (300 nM) (Fig. 6G), we found that addition of RCOR1 to the same assay at ≥ 6 μM (1:1 or greater RCOR1:OVOL2 molar ratio) abrogated the inhibitory effect of 6 μM OVOL2$^{WT}$ towards LSD1 (Fig. 7C). In contrast, with increasing concentrations of OVOL2 in the presence of RCOR1 (1:2 up to 1:9 RCOR1:OVOL2 molar ratio), LSD1 demethylation activity was progressively reduced (Fig. 7C), suggesting that OVOL2 competes with RCOR1 for binding to LSD1.

OVOL2 reportedly binds chromatin primarily at the consensus sequences CCGTTA or CCGCTA[23]. OVOL2 may also bind to GAAACC or GGTTTC within chromatin[52]. Using EMSA, we tested the binding of OVOL2$^{WT}$ and OVOL2$^{C120Y}$ to the DNA motif AACCGTTACC. The OVOL2$^{C120Y}$ protein displayed severely impaired DNA motif binding, with a ratio of OVOL2$^{WT}$: OVOL2$^{C120Y}$ DNA binding ability of 128:1 (Fig. 7D and Supplementary Fig. 14C). Addition of 2 μM LSD1 to a sample containing 2 μM OVOL2$^{WT}$ and the DNA motif AACCGTTACC resulted in formation of a supershifted LSD1-OVOL2-DNA complex in EMSA, but 2 μM or 4 μM RCOR1 dissociated LSD1 from the complex leaving only the OVOL2-DNA complex detectable by EMSA (Fig. 7E), further supporting the idea that RCOR1 competes with OVOL2 for binding to LSD1. We noted that OVOL2 inhibition of LSD1 demethylation activity was independent of DNA binding (Fig. 7F).

## OVOL2 enforces TEC fate by regulating the epigenetic profile to suppress EMT in postnatal TECs

The chromatin modifier BHC complex is recruited to repressor element 1/neuron-restrictive silencer element (RE1/NRSE) sites by repressor element-1 silencing transcription factor (REST) and

deacetylates and demethylates histones at particular sites. Therefore, we tracked genome-wide chromatin accessibility by ATACseq in 4.5-week-old *Ovol2$^{+/+}$* and *Ovol2$^{C120Y/C120Y}$* TECs (Fig. 8A, B and Supplementary Datasets 2 and 3), when approximately 75–85% and 10–24% of TECs are medullary or cortical, respectively (Fig. 3H and Supplementary Fig. 6D). We identified 230 genes only accessible in *Ovol2$^{+/+}$* TEC chromatin while 470 distinct genes were only accessible in *Ovol2$^{C120Y/C120Y}$* TEC chromatin (Fig. 8A and Supplementary Datasets 2 and 3). A group of 18 genes was accessible in both *Ovol2$^{+/+}$* and *Ovol2$^{C120Y/C120Y}$* TEC chromatin, but in different regulating regions (Fig. 8A and Supplementary Datasets 2 and 3).

Functional categorization by Gene Ontology (GO) annotations revealed that open chromatin from *Ovol2$^{+/+}$* and *Ovol2$^{C120Y/C120Y}$* TECs was enriched for genes involved in 25 and 98 signaling pathways, respectively (Fig. 8B)[53]. The four pathways most significantly enriched among open chromatin regions in *Ovol2$^{C120Y/C120Y}$* TECs were involved in epithelial and mesenchymal cell proliferation, specifically negative regulation of epithelial cell proliferation and positive regulation of mesenchymal cell proliferation (40 genes) (Fig. 8B and Supplementary Dataset 4). In contrast, relative to *Ovol2$^{C120Y/C120Y}$* TECs, WT TECs had open chromatin regions enriched for genes involved in negative regulation of cell differentiation (44 genes) (Fig. 8B and Supplementary Dataset 5). Two genes, *Tgfbr1* and *Trp63*, were accessible in both *Ovol2$^{+/+}$* and *Ovol2$^{C120Y/C120Y}$* TECs, but the detected open regions were different (Fig. 8B and Supplementary Fig. 14D). These data indicate that the epigenetic profile of *Ovol2$^{C120Y/C120Y}$* TECs is characterized by increased chromatin accessibility overall, with specific enrichment of accessible genes involved in negative regulation of epithelial cell proliferation and positive regulation of mesenchymal cell proliferation. At the same time, aberrantly inaccessible genes in *Ovol2$^{C120Y/C120Y}$* TECs were genes that negatively regulate cell differentiation. We conclude that OVOL2$^{C120Y}$ alters TEC fate.

We used real-time PCR to measure the mRNA level of 42 genes in the GO categories "negative regulation of cell differentiation" and "negative regulation of developmental process" that were accessible in *Ovol2$^{+/+}$* TECs but not in *Ovol2$^{C120Y/C120Y}$* TECs. 29 out of 42 genes were differentially expressed in *Ovol2$^{C120Y/C120Y}$* vs. *Ovol2$^{+/+}$* TECs, with 8 upregulated genes and 21 downregulated genes in *Ovol2$^{C120Y/C120Y}$* TECs, compared to *Ovol2$^{+/+}$* TECs (Fig. 8C). The expression of 12 genes was unchanged, and 1 gene was not detectable (Fig. 8C). Of note, expression of the EMT suppressor *Grhl2*[54] was reduced in *Ovol2$^{C120Y/C120Y}$* TECs to ~5% of the level in TECs from WT littermates (Fig. 8C). Moreover, a recessive hypomorphic *Grhl2* mutation resulted in reduced frequencies of peripheral B cells and T cells, impaired T-dependent antibody responses to immunization, and defective CD8 T cytolytic

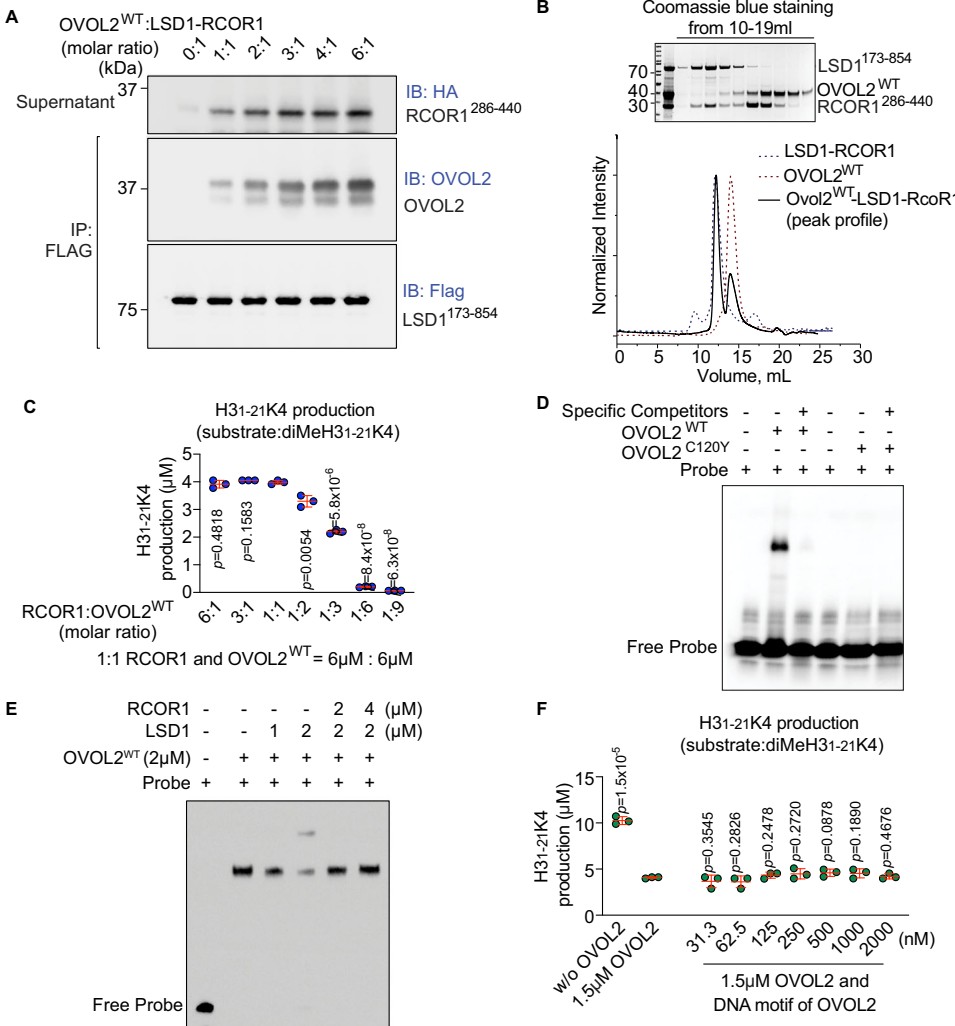

**Fig. 7 | OVOL2 competes with RCOR1 for binding to LSD1 and inhibits LSD1 demethylase activity in vitro. A** OVOL2 competes with RCOR1-HA for binding to LSD1-Flag. Increasing concentrations of purified OVOL2 were incubated overnight with pre-formed LSD1-Flag-RCOR1-HA complexes. Protein mixtures were immunoprecipitated using anti-FLAG M2 agarose and supernatants were immunoblotted with antibodies against HA (RCOR1). Immunoprecipitates were immunoblotted with OVOL2 antibody or FLAG antibody. **B** Size exclusion chromatography of pre-formed 7 μM LSD1-RCOR1 complex mixed with 7 μM OVOL2. (Inset) Coomassie blue staining of the peak fractions from 10 to 19 mL elution volume. **C** LSD1 demethylation activity assay. Demethylation activity of purified LSD1 (300 nM) towards 60 μM diMeH3$_{1-21}$K4 substrate was tested in the presence of OVOL2$^{WT}$ and RCOR1 added at the indicated concentration ratios [RCOR1: OVOL2$^{WT}$ = 6:1, 3:1, 1(6 μM):1(6 μM), 1:2, 1:3, 1:6, 1:9] ($n = 3$ wells per condition). **D** EMSA analysis of the effect of the *C120Y* mutation on OVOL2-DNA binding. 53 ng of purified OVOL2$^{WT}$ or

OVOL2$^{C120Y}$ was incubated with OVOL2 probe (AACCGTTACC) (1 nM) with or without specific competitor (4.5 μM) for 30 min before being harvested for EMSA analysis. **E** EMSA analysis of the effect of RCOR1 on LSD1-OVOL2-DNA complex formation. 2 μM OVOL2$^{WT}$ and 1 or 2 μM LSD1 were incubated overnight to form a complex. Pre-formed LSD1-OVOL2 complexes were incubated with 2 μM (1:1:1 OVOL2:LSD1:RCOR1) or 4 μM (1:1:2) RCOR1 and OVOL2 probe (1 nM) for 1 h before being harvested for EMSA analysis. **F** LSD1 demethylation activity assay. Demethylation activity of purified LSD1 (300 nM) towards 60 μM diMeH3$_{1-21}$K4 substrate was tested in the presence or absence of OVOL2$^{WT}$ (1.5 μM) and increasing concentrations of the DNA motif bound by OVOL2 ($n = 3$ wells per condition). Data are representative of two (**A**, **B**, **E**, **F**) or four independent experiments (**C**, **D**). Data points represent individual wells and means ± SD are indicated (**C**, **F**). *P*-values were determined by two-tailed Student's *t*-test (**C**, **F**).

function (Supplementary Fig. 15), similar to the hematopoietic phenotypes of *Ovol2*$^{C120Y/−}$ mice. The BMP4 regulator DLX1, which is important for T/B cell differentiation[55,56], was also greatly reduced in *Ovol2*$^{C120Y/C120Y}$ TECs.

Real-time PCR analysis demonstrated that 25 out of 38 epithelial and mesenchymal dynamics components accessible only in *Ovol2*$^{C120Y/C120Y}$ TECs displayed altered mRNA expression, with 21 genes upregulated and 4 genes downregulated (Fig. 8D). The expression of 12 genes was unchanged, and 1 gene was not detectable (Fig. 8D). Importantly, the 21 upregulated genes in *Ovol2*$^{C120Y/C120Y}$ TECs are EMT enhancers (Fig. 8D). One of them, *Tbx18*, was increased ~10-fold in *Ovol2*$^{C120Y/C120Y}$ TECs relative to WT TECs (Fig. 8D), consistent with a report that overexpression of TBX18 resulted in smaller thymi via regulating TEC differentiation and

proliferation[57]. Four genes downregulated in *Ovol2*$^{C120Y/C120Y}$ TECs (*Gas1*, *Gata3*, *Ovol2*, and *Zfp36l1*) are EMT suppressors (Fig. 8D).

*Tgfbr1* and *Trp63* were accessible in both *Ovol2*$^{+/+}$ and *Ovol2*$^{C120Y/C120Y}$ TEC chromatin, but the accessible regions were different (Fig. 8B and Supplementary Dataset 4 and 5). Real-time PCR analysis demonstrated that the mRNA level of *Tgfbr1* was unchanged, while that of *Trp63* was significantly decreased in *Ovol2*$^{C120Y/C120Y}$ TECs, compared to *Ovol2*$^{+/+}$ TECs (Supplementary Fig. 14D). Notably, in a recent study TEC-specific TRP63 deletion resulted in catastrophic thymic hypoplasia due to decreased and immature TECs[58]. Collectively, the gene expression data strongly suggest that OVOL2 inhibits EMT in TECs.

To directly link chromatin accessibility changes to epigenetic marks, we used a CUT&RUN assay to perform chromatin

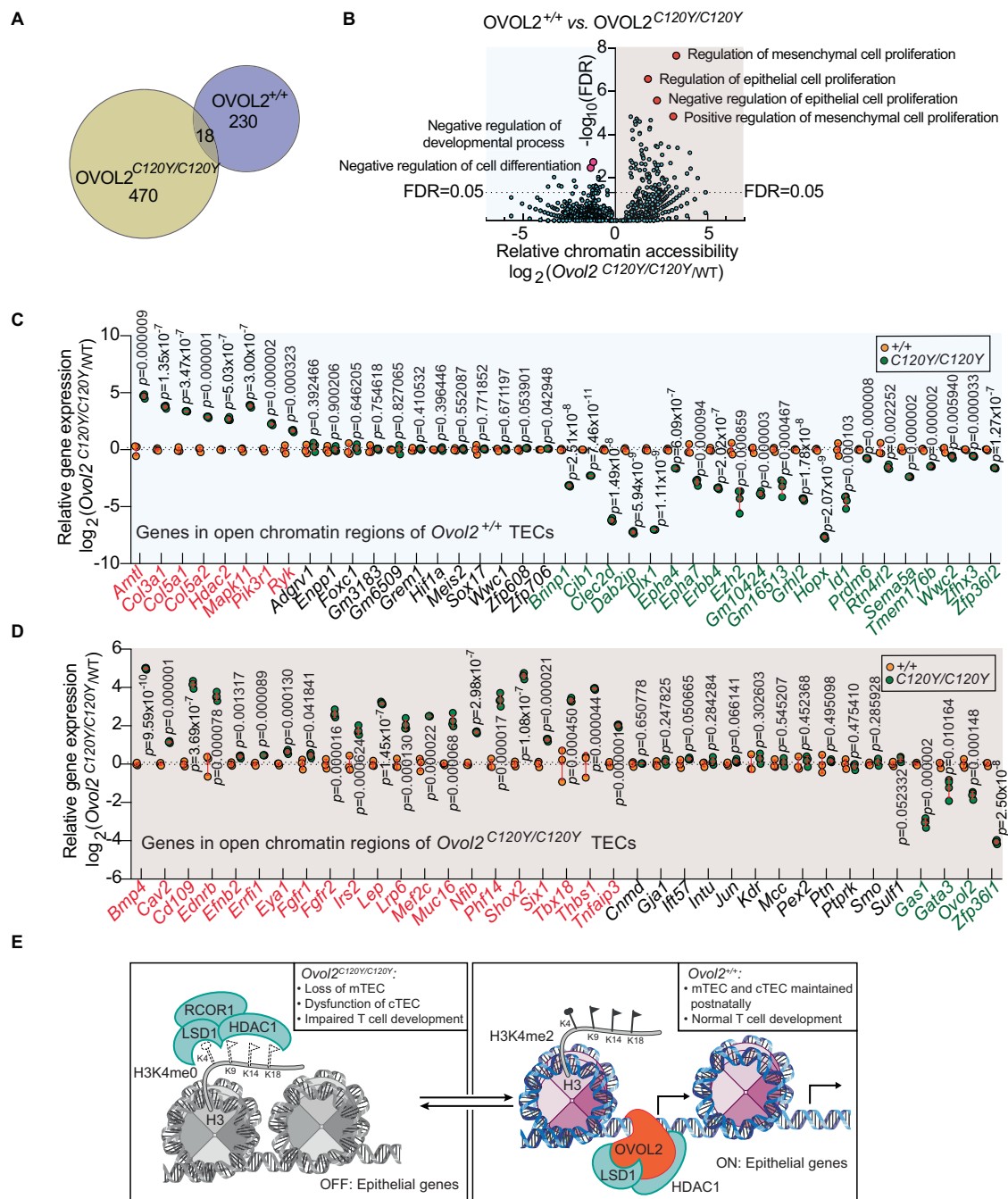

**Fig. 8 | OVOL2 regulates the epigenetic profile to suppress EMT in postnatal TECs. A** Summary of genome-wide chromatin accessibility by ATACseq in FACS sorted TECs from 4.5-wk-old *Ovol2^{C120Y/C120Y}* and *Ovol2^{+/+}* littermates. **B** Gene enrichments analyzed by GREAT based on Gene Ontology (GO) annotations in ATACseq peaks of *Ovol2^{+/+}* and *Ovol2^{C120Y/C120Y}* TECs. False discovery rate (FDR, -log$_{10}$) is plotted vs. relative chromatin accessibility for genes in a GO category (X-axis; left reduced, right increased accessibility relative to *Ovol2^{+/+}*). Each data point represents a group of genes in a GO biological process category. The six selected GO biological processes are highlighted by red dots. FDR = 0.05 criteria line is marked. **C** Real time-PCR analysis of 42 components in "negative regulation of cell differentiation" and "negative regulation of developmental process" GO categories accessible in *Ovol2^{+/+}* TECs (*n* = 4 *C120Y/C120Y* wells, 3 WT wells). **D** Real-time PCR analysis of 38 epithelial and mesenchymal dynamics components accessible in *Ovol2^{C120Y/C120Y}* TECs (*n* = 4 *C120Y/C120Y* wells, 3 WT wells). Data are representative of one experiment (**A**, **B**) or two independent experiments (**C**, **D**). Data points represent individual sorting pools from 5 mice per pool and means ± SD are indicated (**C**, **D**). *P*-values were determined by two-tailed Student's *t*-test (**C**, **D**). *FDR* value (**B**)

was determined by GREAT. +/+ indicates WT and *C120Y/C120Y* indicates the *Ovol2^{C120Y/C120Y}* genotype. **E** Model for epigenetic regulation of epithelial genes by OVOL2. Left: The LSD1-RCOR1-HDAC1 complex inhibits chromatin accessibility and gene expression by demethylating H3K4me2 and deacetylating selected lysine sites on histone 3. RCOR1 is required for LSD1 activity toward H3K4 within its physiological substrate, the nucleosome complex. Without OVOL2, the LSD1-RCOR1 interaction is intact, promoting LSD1 demethylase activity. Absence of methyl groups on H3K4 (dashed circles, H3K4me0) is associated with gene repression. Dashed flags indicate absence of acetyl groups. Right: OVOL2 displaces RCOR1 from binding to LSD1 and thereby inhibits LSD1 demethylase activity, resulting in retention of methylation on H3K4 (filled circles) and gene activation. OVOL2 forms a complex with LSD1 and its DNA target, which may provide DNA sequence specificity for transcriptional activation. Our data suggest that an effect of OVOL2 interaction with LSD1 is activation of epithelial genes, which prevents EMT in TECs. OVOL2 may also inhibit H3K9-directed LSD1 demethylase activity, resulting in repression of mesenchymal genes (not shown; see Discussion). Filled flags indicate the presence of acetyl groups.

immunoprecipitation (ChIP) using an H3K4me2 antibody and *Ovol2*[+/+] and *Ovo2*[C120Y/C120Y] TECs sorted from 4.5-week-old mice. By quantitative PCR, we measured precipitated DNA representing 45 loci with altered chromatin configuration (detected by ATACseq, Fig. 8A, B and Supplementary Datasets S2 and S3) and gene expression (detected by RT-PCR, Fig. 8C, D) in *Ovol2*[C120Y/C120Y] TECs. Among these 45 loci, 25 were accessible in *Ovol2*[+/+] TECs but not in *Ovol2*[C120Y/C120Y] TECs, and 23 out of the 25 were more abundant in H3K4me2 immunoprecipitates from *Ovol2*[+/+] TECs than from *Ovol2*[C120Y/C120Y] TECs (Supplementary Fig. 16A). Among the 45 loci, 24 loci were accessible only in *Ovol2*[C120Y/C120Y] TECs, and 17 out of the 20 were more abundant in H3K4me2 immunoprecipitates from *Ovol2*[C120Y/C120Y] TECs than from *Ovol2*[+/+] TECs (Supplementary Fig. 16B). These data suggest that altered chromatin accessibility in *Ovol2*[C120Y/C120Y] TECs is linked to the change of epigenetic marks on H3K4.

Together, our data show that OVOL2 supports T cell development in the thymus by maintaining TEC fate and function. OVOL2 counters RCOR1 binding to LSD1, thereby inhibiting LSD1 demethylation activity and suppressing EMT in TECs.

## Discussion

In this study, we analyzed the physiological effects of OVOL2 loss-of-function in the hematopoietic system. *Ovol2*[C120Y/-] mice displayed reduced numbers of hematopoietic progenitor cells (CLP, MEP, and GMP), and were anemic, monocytopenic, and lymphopenic. They had progressively hypoplastic spleens and thymi. Both hematopoietic-extrinsic and -intrinsic defects resulted in impaired development and diminished frequencies of B cells in *Ovol2*[C120Y/-] mice. However, the deficits of *Ovol2*[C120Y/-] thymocytes were restricted by their developmental environments in bone marrow transplantation experiments, which indicated they are hematopoietic cell extrinsic. By conditionally deleting OVOL2 in TECs, we demonstrated that defective TEC function and reductions in TEC numbers underlie the impaired thymocyte development observed in OVOL2-deficient mice. However, the TEC and T cell phenotypes of *Foxn1-cre;Ovol2*[fl/fl] mice were delayed and less severe compared with those in *Ovol2*[C120Y/-] mice, indicating that OVOL2 function in cell types that do not express Foxn1 also contributes to these phenotypes. Our findings suggest that thymic endothelial cells and bone marrow stromal cells are two such cell types.

Thymic endothelial cells play a very important role in recruiting ETPs into the thymus and in promoting egress of mature thymocytes via blood vessel formation and chemokine secretion[39,59–61]. Deficiency of OVOL2 in non-TEC cells resulted in the aberrant accumulation of ~10-fold more thymic endothelial cells than in WT thymi. These endothelial cells are likely to be dysfunctional based on their diminished expression of the chemokine genes *P-selectin* and *Ccl25*. Thus, impaired thymic endothelial cell function may contribute to defective T cell development in *Ovol2*[C120Y/-] mice.

Bone marrow stromal cells are predominantly mesenchymal and are necessary for hematopoiesis[62–66]. *Ovol2*[C120Y/-] bone marrow was filled with fat, had reduced proportions of CLPs, and increased LepR+ cells, indicating an altered microenvironment in *Ovol2*[C120Y/-] bone marrow might be another factor affecting T cell development.

Via lineage tracing using tdTomato expression in Foxn1+ cells (i.e., in TEC progenitors), we showed that the majority of TECs in *Ovol2*[C120Y/C120Y] mice were converted to mesenchymal cells. We also saw minimal conversion to endothelial cells, but these did not appear to contribute to excessive endothelial cells in 20-week-old *Foxn1-cre;Ovol2*[fl/fl] thymi. Although thymic mesenchymal cells contribute to thymus organogenesis by producing factors such as bone morphogenic protein-4 (BMP4)[67], and support thymus maintenance in adult animals[68], the excessive mesenchymal cells in OVOL2 deficient thymi are likely to be functionally impaired. This is supported by our observations that unlike WT thymi, *Foxn1-cre;Ovol2*[fl/fl] thymi and *Ovol2*[C120Y/-] thymi were surrounded by adipocytes, which are derived from mesenchymal cells. We conclude that deficiency of OVOL2 in TECs beginning from the TEC progenitor stage causes not only EMT but also the further conversion of thymic mesenchymal cells to adipocytes.

Initial thymus development in OVOL2-deficient mice is characterized by normal mTEC and cTEC frequencies and thymus size through 4 weeks of age, suggesting that OVOL2 is not required for the initial specification of epithelial identity. Consistent with this timing, we found that *Ovol2* mRNA was relatively low at E13.5 during thymus organogenesis, and increased postnatally. We also found that altered gene expression in *Ovol2*[C120Y/C120Y] TECs is evident by 4.5 weeks of age. Thus, our data suggest that OVOL2 becomes important around this time to enforce epithelial identity long term. Substantial heterogeneity in TEC subsets is known to exist[41,69–71], and we do not know whether OVOL2 is important in some or all of them; for example, whether OVOL2 maintains mature TECs vs. TEC progenitors postnatally. Notably, whereas OVOL2 deficiency resulted in loss of mTEC by 12 weeks of age in *Ovol2*[C120Y/-] mice, it impaired the function of cTECs (resulting in reduced ETP recruitment and positive selection) without reducing cTEC numbers.

Previous work has demonstrated that OVOL2 is an important negative regulator of EMT[23,25,26,42,72,73]. OVOL2 has been shown to both repress mesenchymal genes and activate epithelial genes. In the case of repression, OVOL2 was found to directly bind the promoter regions of target genes resulting in their reduced expression, establishing the dogma that OVOL2 is a transcriptional repressor[23,25,42,72]. Mesenchymal genes repressed by OVOL2 include those encoding the Snail, Twist, and Zeb family proteins[25,26,42]. Despite knowledge of OVOL2 target genes, the mechanism of gene repression by OVOL2 has not been reported. Moreover, overexpression of OVOL2 can induce epithelial cell fate and activate epithelial gene transcription[25,42,73], suggesting the possibility that OVOL2 can also transactivate gene expression. This has not been investigated, and earlier reports attributed upregulation of epithelial genes to repression of *Zeb1* since knockdown of *Zeb1* alone had a similar epithelial-inducing effect[25,42,73]. Here, we show that OVOL2 physically interacts with BHC complex components, most prominently with LSD1, and inhibits its chromatin modifier function. Since LSD1 represses transcription by demethylating H3K4me2[74], our data indicate that one effect of OVOL2, by inhibiting LSD1, is to derepress/activate gene expression (Fig. 8E). Thus, OVOL2 may have direct roles in both gene activation and repression.

LSD1 can reportedly also activate gene expression by demethylating H3K9me2[74,75], resulting in transcriptional activation of genes involved in migration and invasiveness in cancer, i.e., mesenchymal genes[75,76]. A failure of OVOL2 to inhibit H3K9-directed demethylase activity could explain the derepression of genes we observed in *Ovol2*[C120Y/C120Y] TECs. However, we were unable to determine the effect of OVOL2 on H3K9 demethylation because we failed to detect LSD1 demethylase activity towards $diMeH3_{1-21}K9$ in vitro, similar to another report[49].

We demonstrate that OVOL2 competes with RCOR1, a positive regulator of LSD1, for binding to LSD1 and inhibits LSD1 demethylation activity. Interestingly, Zhu *et al* reported that LSD1 binds to the promoter and directly represses expression of OVOL2[77], and Egolf et al. showed that *Ovol2* is one of several genes highly upregulated after treatment of epidermal progenitors with an LSD1 inhibitor[78]. *Ovol2* lost LSD1 binding sites and gained H3K4me1 and/or H3K4me2[78]. These reports suggested that *Ovol2* is subject to epigenetic regulation by LSD1. Consistent with that idea, we found that in *Ovol2*[C120Y/C120Y] TECs, *Ovol2* mRNA level was reduced by 67% relative to the level in WT TECs, despite the finding that the *Ovol2* locus was within accessible chromatin. These findings suggest that hyperactive LSD1 might repress *Ovol2* transcription in OVOL2 deficient TECs. Considering these data, we propose that the mutual inhibition between OVOL2 and LSD1 serves to tightly control the balance between these two factors. OVOL2

physically interacts with and inhibits LSD1, preventing overwhelming LSD1 activity that we show alters TEC fate. Conversely, LSD1 prevents unchecked OVOL2 transcription.

We used ATACseq to investigate global chromatin accessibility in 4.5-week-old *Ovol2*[C120Y/C120Y] or WT mouse TECs. Functional categorization of the unique peaks, the four pathways most significantly enriched in open chromatin in *Ovol2*[C120Y/C120Y] TECs were involved in orchestration of EMT. Real-time PCR analysis corroborated the elevated expression of 21/38 EMT regulators identified in accessible chromatin of *Ovol2*[C120Y/C120Y] TECs. In contrast, compared to *Ovol2*[C120Y/C120Y] TECs, WT TECs had an enrichment of accessible genes (correlated with elevated expression) involved in negative regulation of cell differentiation. These data imply that WT TECs actively maintain their population by suppressing in an OVOL2-dependent manner genes that promote mesenchymal cell proliferation or inhibit epithelial cell proliferation, and by expressing genes that inhibit further differentiation.

We know of no other single gene mutation reported to cause the same TEC and T cell phenotypes observed in OVOL2-deficient mice. We favor the hypothesis that the multiplicity of epigenetic changes resulting from OVOL2 mutation leads to the complex and progressive TEC/thymocyte phenotype, although certain OVOL2 targets may play larger roles, e.g. *Grhl2*. A 2019 report showed that knockdown of GRHL2 causing CpG methylation gain and nucleosomal remodeling, including reductions in H3K4me3 and H3K27ac and elevations in H3K27me3, promoted EMT[54]. Finally, we point out the possibility that some phenotypes of *Ovol2*[C120Y/−] thymi and aged, involuted thymi might be mechanistically linked; in particular, EMT has been proposed as a driver of adipogenesis that is associated with declining TECs in the thymus[79].

Overall, OVOL2 controls an extensive epigenetic program through inhibition of LSD1 demethylation activity. In addition, the DNA binding ability of OVOL2 and its impairment by the C120Y mutation suggest direct regulation of its gene targets. In postnatal TECs, OVOL2 prevents aberrant EMT to maintain the numbers and functions of TECs in support of T cell development in the thymus. *OVOL2* expression changes should be considered in cases of unexplained thymic hypoplasia and lymphopenia.

## Methods

### Mice

Eight- to ten-week-old male and female C57BL/6 J mice were purchased from The Jackson Laboratory. N-ethyl-N-nitrosourea (ENU) mutagenesis was performed as described previously[80]. Subsequent breeding to produce G3 mice for phenotypic screening was performed as described previously[27]; only G0 to C57BL/6 J breeding was used. Whole-exome sequencing and mapping were performed as described[27].

To generate mice carrying a targeted re-creation of the *Ovol2*[boh] allele, female C57BL/6 J mice were superovulated and mated with C57BL/6 J male mice as described before[28–30]. Fertilized eggs were collected and in vitro transcribed Cas9 mRNA (50 ng/μL), *Ovol2* small base-pairing guide RNA (50 ng/μL; 5′-AGAGTTGTCGCATGTGCCGG-3′) and Ultramar template DNA (ggttgtggcacttaaggtgacggttgag-catgcgctgcaggcggaagctcttgccacaaaggtcaTagttgtgaatcacagagttgtcgca tgtgccggtggtaaactgggagcagagagagaaagagagc) were injected into the cytoplasm or pronucleus of the embryos. The injected embryos were cultured in M16 medium (Sigma-Aldrich) and 2-cell stage embryos were transferred into the ampulla of the oviduct (10-20 embryos per oviduct) of pseudopregnant Hsd:ICR (CD-1) (Harlan Laboratories) females[28–30].

The founder *C120Y* mice had a 1-bp G to A transition [TCTGTGATTCACAACT(G)[A]TGACCTTTGTGGCAAG with 1-bp replacement denoted in parenthesis] of the 120th residue in the mRNA sequence NM_026924.3 within the third exon, or the 87th residue in the mRNA sequence NM_152947.2 within the third exon, resulting in a change of cysteine to tyrosine at position 120 of 274 amino acids in

OVOL2 isoformA (NP_081200.2) or at position 87 of 241 amino acids in OVOL2 isoformB (NP_694455.2). For genotyping of *C120Y* alleles, the PCR and sequencing primers are listed in Table S1. Compound heterozygotes were produced by crossing *Ovol2*[C120Y/+] and *Ovol2*[+/−] mice, which also resulted from the C120Y targeting strategy. The null allele is a homozygous lethal 8-bp deletion in exon3 [TTGTCGCA TG(TGCCGGTG)GTAAACTGGGA] predicted to result in a frame shifted protein product beginning after amino acid 108 of the protein, which is normally 274 amino acids in length, and terminating after the inclusion of 9 aberrant amino acids. Except for initial phenotypic screening of mice carrying ENU-induced mutations (Fig. 1A, B, and Supplementary Fig. 1A), all experiments were performed using mice with the CRISPR-Cas9 targeted *Ovol2*[C120Y] allele.

To generate mice carrying 3xFlag at the C-terminus of OVOL2[WT] or OVOL2[C120Y] proteins, in vitro fertilization was performed using sperm from *Ovol2*[C120Y/C120Y] male mice and eggs from *Ovol2*[+/+] female mice. Then in vitro transcribed Cas9 mRNA (50 ng/μL), *Ovol2* small base-pairing guide RNA (50 ng/μL; 5′- ttccaccctgagtgaagagg-3′) and Ultramar template DNA (AGACTTCCAAAAAGTTGGCGGCCCTTATGCAG AACAAGCTGACGTCCCCGCTGCAGGAGAATTCCACCCTGAGCGAGGA GGAGGAAAAAAAGGACTACAAAGACCATGACGGTGATTATAAAGATC ATGATATCGATTACAAGGATGACGATGACAAGTGAAGAGCGAGCAAG AAAAAGCAGAGGAGAGACGCCGGCGCC) were injected into the cytoplasm or pronucleus of the embryos. For genotyping of 3xFlag insertion, the PCR and sequencing primers are listed in Table S1. *Ovol2*[C120Y/C120Y] and OVOL2-3xFlag-expressing *Ovol2*[F/F] mice were produced by intercrossing *Ovol2*[C120Y/+] or *Ovol2*[F/+] mice, respectively.

*Ovol2*[fl/fl] mice carrying a *lox*P-flanked *Ovol2* exon 3 cassette (containing 190 bp coding sequence) were obtained from Dr. Zhao Zhang (UT Southwestern Medical Center).

Mice carrying the *Grhl2*[clayton] allele were generated in the same way as mice with a re-creation of the *Ovol2*[boh] allele, except that *Grhl2* small base-pairing guide RNA (50 ng/μL; 5′-TTGCCTCTCCCTCTAGGCGT-3′) and Ultramar template DNA (TACTGGCACTCCCGGCAGCACAC TGCCAAGCAGAGGGTCCTTGACATTGGTGGGTTGCCTCTCCCTCTAG GCGTTGGCAACTGGCTGGGTTGACCACTACAGGGGCTGGGTCGCAC TGTGCTATGCCAGGCACTGAAT) were used. The founder *intron + 4 A > G* mice have a 1-bp A to G transition of the fourth nucleotide of intron 7 (ENSMUST00000022895.15) [ccttgacattggtg(a)[G]gttgcc tctccctct with 1-bp replacement denoted in parenthesis] genotyped by PCR and sequencing using primers listed in Table S1.

C57BL/6 J (#:000664), C57BL/6.SJL (CD45.1) (#:002014), *Rag2*[−/−] (#:008449), B6(Cg)-*Foxn1*[tm3(cre)Nrm]/J (*Foxn1-Cre*) (#:018448), and B6.Cg-Gt(ROSA)26Sor[tm9(CAG-tdTomato)Hze]/J (*Ai9*) (#:007909) mice were purchased from The Jackson Laboratory. *Foxn1-Cre;Ovol2*[fl/fl], *Ovol2*[C120Y/C102Y];*Foxn1-cre;Ai9*, and CD45.1;*Rag2*[−/−] strains were generated by intercrossing mouse strains. Conventionally reared mice were housed in specific pathogen–free conditions, 12 h light/12 h dark cycle, 20–26 °C ambient temperature, and 30-70% humidity at the University of Texas Southwestern Medical Center. Experimental and control animals were littermates and were co-housed. Mice were euthanized by $CO_2$ inhalation followed by cervical dislocation. All experimental procedures were performed in accordance with protocols approved by the Institutional Animal Care and Use Committee.

### Flow cytometry analysis of B cell, T cell, and HSC development

Peripheral blood was collected and treated as described before[28–30]. Briefly, red blood cells (RBCs) were lysed with hypotonic buffer (eBioscience) and RBC-depleted samples were stained for 1 hour at 4 °C, in 100 μL of 1:200 cocktail of fluorescence-conjugated antibodies to 15 cell surface markers encompassing the major immune lineages: B220 (BD, clone RA3-6B2), CD19 (BD, clone 1D3), IgM (BD, clone R6-60.2), IgD (Biolegend, clone 11-26 c.2a), CD3ε (BD, clone 145-2C11), CD4 (BD, clone RM4-5), CD8α (Biolegend, clone 53-6.7), CD11b (Biolegend, clone M1/70), CD11c (BD, clone HL3), F4/80 (Tonbo, clone BM8.1),

CD44 (BD, clone 1M7), CD62L (Tonbo, clone MEL-14), CD5 (BD, clone 53-7.3), CD43 (BD, clone S7), NK 1.1 (Biolegend, clone OK136) and 1:200 Fc block (Tonbo, clone 2.4G2). For the lymphocytic infiltration analysis, kidney, lung, liver, intestine, and salivary gland were digested in PBS containing 0.5 U/mL Liberase TM (Sigma-Aldrich) and 0.02% (w/v) DNase I (Roche) to produce single cell suspensions before antibody staining. Flow cytometry data was collected on a BD LSR Fortessa and the proportions of immune cell populations in each G3 mouse were analyzed with FlowJo 10 software. The resulting screening data were uploaded to Mutagenetix for automated mapping of causative alleles.

To stain the B cell and T cell subsets, bone marrow cells, splenocytes, or thymocytes were isolated and RBC lysis buffer was added to remove RBCs. Cells were stained at 1:200 antibody dilution in the presence of anti-mouse CD16/32 antibody (Tonbo, clone 2.4G2) and different antibody cocktails: B cells in bone marrow [IgM (BD, clone R6-60.2), IgD (Biolegend, clone 11-26 c.2a), B220 (BD, clone RA3-6B2), CD43 (BD, clone S7), CD24 (Biolegend, clone M1/69), Ly-51 (BD, clone BP-1), CD19 (BD, clone 1D3)]; splenic B cells [CD4 (BD, clone RM4-5), IgM (BD, clone R6-60.2), IgD (Biolegend, clone 11-26 c.2a), B220 (BD, clone RA3-6B2), CD23 (BD, clone B3B4), CD21/CD35 (Biolegend, clone 7E9), CD93 (BD, clone AA4.1)]; thymocytes [CD25 (Biolegend, clone 3C7), CD44 (BD, clone 1M7), CD4 (BD, clone RM4-5), CD8α (Biolegend, clone 53-6.7), CD69 (eBioscience, clone H1.2F3), CD5 (BD, clone 53-7.3), CD117 (c-kit) (Biolegend, clone 2B8), TCR-β (BD, clone H57-597), PD-1 (eBioscience, clone 29 F.1A12), CD197 (CCR7) (Biolegend, clone 4B12), Helios (Biolegend, clone 22F6)][28–30,41,81].

HSC and progenitor cells were stained with Alexa Fluor 700 anti-mouse lineage cocktail (Biolegend, clones 17A2/RB6-8C5/RA3-6B2/Ter-119/M1/70), CD34 (eBioscience, clone RAM34), CD135 (Biolegend, clone A2F10), CD16/CD32 (eBioscience, clone 93), CD127 (BD, clone SB/199), Ly-6A/E (Sca-1) (Biolegend, clone D7), CD117 (c-kit) (Biolegend, clone 2B8) at 1:200 antibody dilution[28–30].

TECs, including mTECs, mTEC^high, mTEC^low, and cTECs were detected by flow cytometry using the following antibodies and the gating strategy shown in Supplementary Fig. 6A: EpCAM (eBioscience, clone G8.8), CD45 (eBioscience, clone 30-F11), Ly51 (BD Pharmingen, clone 6C3), UEA-1 (Vector Laboratories, clone FL-1061), and I-A/I-E (Biolegend, clone M5/114.15.2).

Flow cytometry gating strategies to detect lymphocytes and myeloid cells in secondary lymphoid tissues and other tissues, LepR+ cells in bone marrow, endothelial and mesenchymal cells in thymi, T-cell progenitors (ETP) in thymi, thymocytes undergoing positive selection, and regulatory T cells (Treg cells) are shown in Supplementary Fig. 17A–F. Flow cytometry gating strategies to detect B, T, and HSC development were previously published in Supplementary Figs. 1–3 of PMID: 32071239[31]. The gating strategy to detect thymocytes in stages of negative selection was previously published in Supplementary Fig. 2a of PMID: 32840480[41].

For the intracytoplasmic AIRE staining in mTECs and FoxP3+ regulatory T cell analysis in mesenteric lymph node and spleen, after a first staining with antibodies at 1:200 antibody dilution for surface markers, cells were collected, fixed, permeabilized with Cytofix/Cytoperm (BD Bioscience), and stained for 30 min with Aire (eBioscience, clone 5H12) or CD25 (Biolegend, clone 3C7) and FoxP3 (BD, clone MF23) antibodies at 1:200 antibody dilution in the presence of Perm/Wash solution (BD Bioscience). Flow cytometry data were collected on a BD LSR Fortessa and analyzed with FlowJo 10 software.

### Flow cytometry analysis and sorting of thymic stromal cells and tdTomato+ cells

Preparation of thymic stroma was performed as previously described[82]. Briefly, thymic tissues were cut into small pieces, enzymatically digested, and mesh-filtered. Single cell suspensions were stained with antibodies at 1:200 antibody dilution against CD45 (eBioscience, clone 30-F11), Alexa Fluor 700 anti-mouse lineage

cocktail (Biolegend, clones 17A2/RB6-8C5/RA3-6B2/Ter-119/M1/70), EpCAM (eBioscience, clone G8.8), CD31 (Biolegend, clone 390), CD140a (PDGFR-α) (Biolegend, clone APA5), CD140b (PDGFR-β) (Biolegend, clone APB5), CD80 (Biolegend, clone 16-10A1), CD86 (Biolegend, clone GL-1), podoplanin/gp38 (Biolegend, clone 8.1.1), H-2Kb (eBioscience, clone AF6-88.5.5.3), Ly51 (BD Pharmingen, clone 6C3), UEA-1 (Vector Laboratories, clone FL-1061), and I-A/I-E (Biolegend, clone M5/114.15.2) and sorted using a FACS Aria II (Becton Dickinson)[83,84]. The flow cytometry gating strategy to detect epithelial, endothelial, and mesenchymal cells in the thymus of *Foxn1-Cre;Ai9* mice is shown in Supplementary Fig. 17G.

### Bone marrow chimeras

Bone marrow transplantation was performed as previously described[28–30]. Recipient mice were lethally irradiated by giving a 7 Gy (1 Gy = 100 rads) exposure by X-ray irradiator twice at a 5 h interval. Femurs derived from donor C57BL/6.SJL (CD45.1), *Ovol2^{C120Y/−}* (CD45.2) compound heterozygotes, or their WT littermates were flushed with PBS using a 25 G needle. To remove bits of bone, the marrow was homogenized, and the solution was filtered through a sterile 40 μm nylon cell strainer (BD Biosciences). The RBC-depleted cells were pelleted and resuspended in 1 mL PBS and kept on ice.

The following transfers were performed: (1) Bone marrow cells from *Ovol2^{C120Y/−}* (CD45.2) compound heterozygotes, or their WT littermates were transferred into irradiated CD45.1, *Rag2^{−/−}* recipients. (2) Bone marrow cells from C57BL/6.SJL (CD45.1) mice were injected into irradiated *Ovol2^{C120Y/−}* (CD45.2) compound heterozygotes, or their WT littermates through retro-orbital injection. For 4 weeks post engraftment, mice were maintained on an antibiotic. Twelve weeks after bone marrow transplantation, peripheral blood, T cells in the thymus, and B cells in the bone marrow and spleen were sampled and assessed with flow cytometry using fluorescence-conjugated antibodies against the CD45 congenic markers [CD45.1 (Biolegend, cloneA10), CD45.2 (Biolegend, clone 104)].

### Immunization and enzyme-linked immunosorbent assay (ELISA) analysis

Immunization and ELISA were performed as previously described[28–30]. Briefly, 12–20-week-old mice were intramuscularly immunized with ovalbumin (200 μg; Invivogen) + alum (20 μg, Invivogen) as adjuvant on day 0 and intraperitoneally immunized with NP$_{50}$-AECM-Ficoll (50 μg, Biosearch Technologies) on day 8. Blood was collected on day 6 post NP-Ficoll immunization in minicollect tubes (Mercedes Scientific) and centrifuged at 1500 x g to separate serum for analysis of antigen-specific IgG or IgM concentration by ELISA.

ELISA analysis of antigen-specific IgG and IgM responses was performed as previously described[28–30].

Total IgM, IgA, or IgG2b concentration in *Ovol2^{C120Y/C120Y}* and *Ovol2^{+/+}* littermates was analyzed by ELISA with IgM, IgA, or IgG2b mouse Uncoated ELISA Kit (Thermo Fisher Scientific).

### In vivo NK Cell and CD8 + T cell cytotoxicity assay

Cytolytic CD8 + T cell effector function was determined by a standard in vivo cytotoxic T lymphocyte (CTL) assay described before[28–30]: Briefly, splenocytes were isolated from naïve B6 mice and divided in half. According to established methods, half were stained with 5 μM CFSE (CFSE^hi), and half were labeled with 0.5 μM CFSE (CFSE^lo). CFSE^hi cells were loaded with 5 μM ICPMYARV peptide, an MHC I-restricted peptide epitope of the *E. coli* β-galactosidase protein for mice with the H-2^b haplotype[85] or 5 μM SIINFEKL peptide, an MHC I-restricted peptide epitope of the chicken ovalbumin protein for mice with the H-2^b haplotype[28]. CFSE^lo cells were left untreated. CFSE^hi and CFSE^lo cells were mixed (1:1) and 2 × 10^6 cells were administered through retro-orbital injection to naïve mice and mice immunized with rSFV-βgal or alum-ova. Blood was collected 48 h after adoptive transfer,

and CFSE intensities from each population were assessed by flow cytometry. Lysis of target (CFSE$^{hi}$) cells was calculated as: % lysis = [1 − (ratio $_{control\ mice}$/ratio $_{vaccinated\ mice}$)] × 100; ratio = percent CFSE$^{lo}$/percent CFSE$^{hi}$.

Measurement of NK cell-mediated killing was described previously[28–30]: Splenocytes from control C57BL/6 J (0.5 μM Violet; Violet$^{lo}$) and $\beta2m^{-/-}$ mice (5 μM Violet; Violet$^{hi}$) were stained with Cell-Trace Violet. Equal numbers of Violet$^{hi}$ and Violet$^{lo}$ cells were transferred to recipient mice by retro-orbital injection. Twenty-four hours after transfer, blood was collected and Violet intensity from each population was assessed by flow cytometry. % lysis = [1 − (target cells/control cells) / (target cells/control cells in $\beta2m^{-/-}$)] × 100.

## In vivo homeostatic T cell proliferation

Splenic pan-T cells were isolated using the EasySep mouse pan-T cell isolation kit (Stemcell Technologies). Pan-T cells were labeled for 20 min at 37 °C with 5 μM CellTrace Far Red (Life Technologies) and then intravitreally injected into recipient mice. The following transfers were performed: (1) A 1:1 mixture of T cells from CD45.1 (WT) and CD45.2 ($Ovol2^{C120Y/-}$) mice was transferred to irradiated CD45.1 recipients. (2) T cells from CD45.1 (WT) mice were transferred to irradiated CD45.2 ($Ovol2^{C120Y/-}$) or littermate WT recipients. (3) $Ovol2^{C120Y/-}$ T cells from bone marrow chimeric mice (generated by BMT from $Ovol2^{C120Y/-}$ mice to irradiated $Ovol2^{+/+};Rag2^{-/-}$ recipients) were transferred to irradiated CD45.1 recipients. (4) T cells from bone marrow chimeric mice (generated by BMT from WT mice to irradiated CD45.1;$Rag2^{-/-}$ recipients) were transferred to irradiated CD45.1 recipients. The marked cells were analyzed at day 7 post-injection using an LSR flow cytometer (Becton Dickinson) and CD4 (BD, clone RM4-5), CD8α (Biolegend, clone 53-6.7), and fluorescence-conjugated antibodies against the CD45 congenic markers [CD45.1 (Biolgend, cloneA10), CD45.2 (Biolgend, clone 104)].

## Plasmids

Mouse OVOL2$^{WT}$ (NM_026924.3) and OVOL2$^{C120Y}$ were tagged N-terminally with HA epitope in pCMV6 vector. Full-length mouse LSD1 (NM_133872.2), RCOR1 (NM_19023), HDAC1 (NM_008228), HDAC2 (NM_008229), HMG20A (NM_025812), HMG20B (NM_001163165), or PHF21A (NM_138755) with FLAG epitope were cloned into pCDNA6 vector.

For protein expression, full-length mouse OVOL2$^{WT}$ and OVOL2$^{C120Y}$ were cloned into a custom pET-derived vector with an N-terminal 6x histidine (His)-tag and maltose binding protein (MBP)-tag followed by the Tobacco Etch Virus (TEV) protease cleavage site. Mouse LSD1$^{173-854}$ (NM_133872.2) and RCOR1$^{286-440}$ (NM_19023) were cloned into modified pET-derived vector encoding an N-terminal 14x histidine followed by bdNEDD8 protease cleavage site. The C120Y point mutation of OVOL2 was generated by standard site-directed mutagenesis using the QuikChange II site-directed mutagenesis kit (Agilent Technologies).

All plasmids were sequenced to confirm the absence of undesirable mutations.

## Cell culture, transfection, co-immunoprecipitation (co-IP), and immunoblotting

Cell culture, co-IP, and immunoblotting was described previously[32]: HEK293T cells were grown at 37 °C in DMEM (Life Technologies)/10% (v/v) FBS (Gibco)/1% antibiotics (Life Technologies) in 5% CO$_2$. Transfection of plasmids was carried out using Lipofectamine 2000 (Life Technologies) according to the manufacturer's instructions. 36 to 48 h after transfection, cells were harvested in NP-40 lysis buffer (20 mM Tris−Cl, pH 7.5, 150 mM NaCl, 1 mM EDTA, 1 mM EGTA, 1% (v/v) Nonidet P-40, 2.5 mM Na$_4$P$_2$O$_7$, 1 mM C$_3$H$_9$O$_6$P, 1 mM Na$_3$VO$_4$, and protease inhibitors) for 45 min at 4 °C. Co-IP assays were performed using cell extracts from HEK293T cells overexpressing OVOL2$^{WT}$, OVOL2$^{C120Y}$,

LSD1, RCRO1, HDAC1, HDAC2, HMG20A, HMG20B, or PHF21A proteins in separate cultures. Extracts of cells expressing the proteins of interest were mixed. Proteins were immunoprecipitated by anti-Flag M2 affinity gel (Sigma Aldrich). Captured proteins were eluted by PBS with 30 μL 200 μg/mL 3xFlag peptides (Sigma Aldrich), and 8 μL of sample buffer mixed with 0.1% bromphenol blue, heated to 95 °C for 3 min, centrifuged at 12,000 × g for 1 min, and the supernatants were loaded onto SDS-PAGE for immunoblotting analysis by anti-Flag (Sigma Aldrich, clone M2) or anti-HA (Cell Signaling Technology, clone C29F4), as described below.

For direct immunoblot analysis, spleen, thymus, bone marrow, liver, pancreas, ovary, cerebellum, brainstem, pituitary gland, hypothalamus, olfactory bulb, and cortex were homogenized, and the solution was filtered through a sterile 40 μm nylon cell strainer (BD Biosciences). Single cell suspensions from tissues and FACS sorted TECs were lysed in buffer (1% SDS (Thermo Fisher), 1:10,000 Benzonase (Sigma), 1:100 Protease Inhibitor Cocktail (Cell Signaling Technology), in buffer A (50 mM HEPES, 2 mM MgCl$_2$, 10 mM KCl). Protein concentration was measured using the BCA assay (Pierce). Equal amounts (~20 μg) of protein extracts were separated by electrophoresis on 4–12% Bis-Tris mini gels (Life Technologies) and transferred to nitrocellulose membranes (Bio-Rad). After blocking in Tris-buffered saline containing 0.05% (v/v) Tween-20 (TBS-T) with 5% (w/v) BSA at room temperature for 2 h, the membrane was incubated overnight with primary antibody at 1:1000 dilution [anti-Flag (Sigma Aldrich, clone M2), anti-LSD1 (CST, clone C69G12), anti-HDAC1 (CST, clone D5C6U), anti-HDAC2 (CST, clone D6S5P), anti-RCOR1 (Thermo Fisher Scientific, AF6047-SP), anti-α-Tubulin (CST, clone DM1A), anti-H3 (Abcam, ab1791), anti-H3K4me2 (Thermo Fisher Scientific, 39141), anti-GAPDH (CST, clone D16H11), anti-OVOL2 (Santa Cruz, clone E-9), and anti-β-Actin (CST, clone 8H10D10)] at 4 °C in 5% (w/v) BSA in TBS-T with gentle rocking. The membrane was then incubated with secondary antibody at 1:4000 dilution [goat anti-mouse IgG-HRP (Southern Biotech), goat anti-rabbit IgG-HRP (Thermo Fisher Scientific), or donkey anti-sheep IgG-HRP (Santa Cruz)] for 1 h at room temperature. The chemiluminescence signal was developed by using SuperSignal West Dura Extended Duration Substrate kit (Thermo Fisher Scientific) and detected by a G:Box Chemi XX6 system (Syngene).

## Generation of EL4 cell line stably expressing OVOL2-Flag protein for LC-MS/MS Analysis

Retrovirus expressing OVOL2 was produced and used for EL4 cell transduction as described previously[29]: PT67 cells (ATCC) were transfected with FLAG-tagged OVOL2 (NM_026924.3)-pMSCV-IRES-GFP or empty vector (pMSCV-IRES-GFP) using Lipofectamine 2000 (Life Technologies). Green fluorescent protein (GFP)-positive PT67 cells were sorted twice using an Aria II cell sorter (BD). Retroviruses recovered from the media of PT67 cells were concentrated by Retro-X concentrator kit (Takara) according to the manufacturer's instructions and used to infect WT EL4 cells (ATCC) with polybrene (4 μg/mL). Forty-eight hours after retrovirus infection, GFP+ cells were sorted by flow cytometry and single colonies selected by limiting dilution assay. The single colonies were screened and confirmed by immunoblotting with anti-Flag (Sigma Aldrich, clone M2). Immunoprecipitation was performed using anti-FLAG M2 agarose beads (Sigma Aldrich) for 4 h at 4 °C with (20 or 100 unit) or without DNase I, and beads were washed five times in Nonidet P-40 lysis buffer (Table S2). The proteins were eluted with 100 μg/mL 3xFlag at 4 °C for 30 min.

## Protein digestion and mass spectrometry

Protein digestion, mass spectrometry, and protein/peptide identification were performed as described previously[86]: Protein was solubilized in 8 M urea 100 mM triethylammonium bicarbonate pH 8.5 and reduced with 5 mM Tris (2-carboxyethyl) phosphine hydrochloride (Sigma-Aldrich, product C4706) and alkylated with 55 mM

2-Chloroacetamide (Sigma-Aldrich, product 22790). Proteins were digested for 18 h at 37 °C in 2 M urea 100 mM triethylammonium bicarbonate pH 8.5, with 0.5 ug trypsin (Promega, Madison, WI, product V5111). Single phase analysis was performed using am Orbitrap Fusion Lumos Tribrid Mass Spectrometer (Experiments 1 and 3) (Thermo Scientific) or a Q Exactive HF Orbitrap Mass Spectrometer (Experiment 2) (Thermo Scientific).

Protein and peptide identification were done with MSFragger (version 3.4)[87] (https://fragpipe.nesvilab.org/) using a mouse protein database downloaded from UniProt (uniprot.org) (11/15/2022, 21986 entries), common contaminants and reversed sequences added. The search space included all fully tryptic peptide candidates with a fixed modification of 57.021464 on C, variable modification of 15.9979 on M and 42.0106 on the N-terminus. MS1 quantification was done with Total Intensity and no match between runs. Protein intensity values were combined for replicates. OVOL2 interactors identified by MS were summarized by Venn diagram generated using http://www.biovenn.nl/index.php[88].

### Electrophoretic mobility shift assays (EMSA)

OVOL2$^{WT}$, OVOL2$^{C120Y}$, LSD1, or RCOR1 protein at the concentrations indicated in the figures and 1 nM 5′-end biotin labeled DNA probe were incubated with or without specific competitor (4.5 μM) at room temperature for 30 min. The proteins and DNA interactions were analyzed by LightShift Chemiluminescent EMSA Kit (Thermo Fisher).

### Thermal stability assay

The thermal stability assay was performed as described previously[29]: OVOL2$^{WT}$ and OVOL2$^{C120Y}$ were analyzed using a SYPRO orange thermal shift assay following the manufacturer's protocol (Life Technologies). Briefly, 5 μL of 100× SYPRO orange was added into a 45 μL protein sample containing OVOL2$^{WT}$ or OVOL2$^{C120Y}$. The final concentration of each component was as follows: 50 μM protein and 10× SYPRO orange. The melting profile of each protein sample was monitored using a qPCR instrument reading the fluorescence of SYPRO orange. The temperature was increased from 25 °C to 95 °C. The derivatives of each melting cure were plotted against temperature.

### Immunohistochemistry (IHC), Hematoxylin and eosin (H&E) and Oil red O staining

Bones were decalcificated before staining. IHC, H&E and Oil red O staining were performed using standard procedures by UT Southwestern Histology Core. The primary antibodies to distinguish cortical and medullary regions in the thymus were cytokeratin 8 (CK8) (ABclonal, clone A1024) and cytokeratin 5 (CK5) (ABclonal, clone A2662) at 1:50 dilution. The thymi were incubated with secondary goat anti-rabbit IgG-HRP antibody. The chemiluminescence signal was developed by using SignalStain DAB Substrate Kit (Cell Signaling, #8059).

### Expression and purification of recombinant OVOL2, LSD1 and RCOR1

A customized pET-derived vector (containing an N-terminal, TEV cleavable 6x histidine tag fused to MBP tag) encoding full-length wild-type OVOL2 or OVOL2$^{C120Y}$ were transformed into E. coli Rosetta (DE3) cells. Cells were grown at 37 °C to an OD$_{600}$ of 0.8 and induced with a final concentration of 0.25 mM IPTG (isopropyl-β-D-thiogalactoside) at 18 °C overnight. After harvesting cells by centrifugation for 20 min at 4000 × g, cell pellets were resuspended in B1 buffer (50 mM HEPES pH 8.0, 800 mM NaCl, 40 mM imidazole, 10% glycerol) with complete protease inhibitors (Thermo Scientific Pierce Protease Inhibitor Tablets) and lysed via sonication. The lysate was cleared by centrifuging for 40 min at 18,000 × g. The cleared lysate was loaded onto a 5 mL Hi-Trap Ni-NTA column (Cytiva), washed with 5 column volumes of buffer B1 and subsequently eluted with 2 column volumes of buffer

B2 (50 mM HEPES pH 8.0, 100 mM NaCl, 400 mM imidazole, 10% glycerol) and loaded directly onto a 5 mL Hi-Trap Heparin column (Cytiva). Protein was eluted using a linear gradient of buffer A1 (50 mM HEPES pH 8.0, 100 mM NaCl, 40 mM imidazole, 10% glycerol) and buffer B1 spanning 10 column volumes. The main peak eluted from ion exchange was pooled, next affinity/solubility tags were cleaved overnight with TEV protease. To remove cleaved MBP protein from the sample, we performed extra orthogonal purification with Hi-Trap Heparin column. Peak fractions from orthogonal ion-exchange column were further purified by size exclusion chromatography on an S200 16/60 Superdex column (Cytiva) equilibrated in gel filtration buffer (50 mM HEPES pH 8.0, 250 mM NaCl, 1 mM TCEP, 10% glycerol and 0.05 mM ZnCl$_2$). The final peak from size exclusion chromatography was concentrated using Amicon Ultra spin columns (Merck Millipore) with a 30 kDa cutoff.

Expression of (His)$_{14}$-bdNEDD8-LSD1$^{173-854}$ and (His)$_{14}$-bdNEDD8-RCOR1$^{286-440}$ was carried out in E. coli Rosetta (DE3) cells. (His)$_{14}$-bdNEDD8-LSD1$^{173-854}$ and (His)$_{14}$-bdNEDD8-RCOR1$^{286-440}$ were purified using a similar method as described above with two column purifications, first Ni-NTA followed by Heparin column (Cytiva) using the same buffer conditions (A1, B1and B2 buffers). The main peak eluted from ion exchange was pooled, next affinity/solubility tags were cleaved overnight with bdNEDD8p protease, and the proteins were purified by size exclusion chromatography on an S200 16/60 Superdex column equilibrated in gel filtration buffer (50 mM HEPES pH 7.8, 200 mM NaCl, 1 mM TCEP, 10% glycerol). Purity was assessed by Coomassie-stained SDS-PAGE, and aliquots were concentrated using Amicon Ultra spin columns (Merck Millipore) with 30 kDa cutoff and snap frozen in liquid nitrogen for later use.

### Size exclusion chromatography

**OVOL2-LSD1 complex.** The interactions between OVOL2 and LSD1 were probed by size exclusion chromatography (SEC). Gel filtration purified full-length Ovol2$^{WT}$ and LSD1$^{173-854}$ were mixed at equimolar concentrations and incubated on ice for 30 min before injection onto a Superdex 200 Increase 10/300 GL size exclusion column equilibrated with 30 mM HEPES pH 7.8, 150 mM NaCl, 1 mM TCEP, 5% glycerol and 0.05 mM ZnCl$_2$) at a flow rate of 0.75 mL/min. Samples were centrifuged at 22,000 × g for 30 min to remove any aggregates before injection onto a Superdex 200 column. Peak fractions from gel filtration were assessed by Coomassie stained SDS-PAGE to visualize co-migration of the two proteins in the peak fractions. For comparison, full-length Ovol2$^{WT}$ and LSD1$^{173-854}$ were run individually on the same column with identical running conditions as for the complex.

**LSD1-RCOR1 complex.** To purify the LSD1-ROCR1 complex, individual gel filtration purified LSD1$^{173-854}$ and RCOR1$^{286-440}$ were mixed at 1:1 molar ratio and incubated on ice for 30 min before injection onto a Superdex 200 Increase 10/300 GL size exclusion column equilibrated with 30 mM HEPES pH 7.8, 200 mM NaCl, 1 mM TCEP, 10% glycerol) at a flow rate of 0.75 mL/min. Peak fractions from gel filtration were assessed by Coomassie stained SDS-PAGE.

**OVOL2 and LSD1-RCOR1 complex.** Similar to analysis of the OVOL2-LSD1 complex, OVOL2 and the LSD1-RCOR1 complex mixed at 1:1 molar ratio, or OVOL2, LSD1, and RCOR1 mixed at 1:1:1 molar ratio was run on a Superdex 200 Increase 10/300 GL size exclusion column with identical conditions as for the OVOL2-LSD1 complex. Peak fractions from gel filtration were assessed by Coomassie stained SDS-PAGE.

**OVOL2 and RCOR1 complex.** OVOL2 and RCOR1$^{286-440}$ were mixed at 1:1 molar ratio and run on a Superdex 200 Increase 10/300 GL size exclusion column as above and peak fractions were analyzed by Coomassie stained SDS-PAGE.

## LSD1 activity assay

LSD1 activity was measured by peroxidase-coupled assay[89], which monitors hydrogen peroxide production. The 150 µL reactions were initiated by adding 50 µL of buffered substrate solution [diMeH3$_{1-21}$K4 (Enzo) or diMeH3$_{1-21}$K9 (Diagenode), final concentration 60 µM] to reaction mixtures (100 µL) consisting of 50 mM HEPES buffer (pH 7.5), 0.1 mM 4-aminoantipyrine, 1 mM 3,5-dichloro-2-hydroxybenzenesulfonic acid, 0.76 µM horseradish peroxidase (Worthington Biochemical Corporation), and 300 nM LSD1 with (1) OVOL2$^{WT}$ or OVOL2$^{C120Y}$ protein at the concentrations indicated in the figures; or (2) OVOL2$^{WT}$ and RCOR1 at concentration ratios of 9:1, 6:1, 3:1, 2:1, 1(6 µM):1(6 µM), 1:3, and 1:6. Reaction mixtures were equilibrated at 25 °C for 2 min prior to activity measurement. Absorbance changes were monitored at 515 nm, and an extinction coefficient of 26,000 M$^{-1}$ cm$^{-1}$ was used to calculate product formation.

## HDAC1 activity assay

OVOL2$^{WT}$ or OVOL2$^{C120Y}$ protein at the concentrations indicated in the figures was added to the HDAC1 Fluorogenic Assay Kit (BPS Bioscience) and deacetylation of the substrate was determined according to the manufacturer's instructions.

## ATACseq library preparation and data analysis

$5 \times 10^5$ FACS sorted TECs from Ovol2$^{C120Y/C120Y}$ or WT littermates were centrifuged at $500 \times g$ for 5 min at 4 °C. The supernatant was removed and discarded. Cells were washed once with 50 µL of cold PBS and then centrifuged at $500 \times g$ for 5 min at 4 °C. The supernatant was removed and discarded. The cell pellet was resuspended in 50 µL of cold lysis buffer (10 mM Tris·Cl, pH 7.4, 10 mM NaCl, 3 mM MgCl2, 0.1% (v/v) Igepal CA-630) and centrifuged immediately at $500 \times g$ for 10 min at 4 °C to obtain nuclei in the pellet. The supernatant was discarded, and the nuclei pellet was immediately used to generate libraries using the Illumina Tagment DNA TDE1 Enzyme and Buffer Kit (Illumina, cat. no. FC-121-1030). Briefly, the nuclei pellet was resuspended in the transposition reaction mix [25 µL TD (2× reaction buffer from kit), 2.5 µL TDE1 (Nextera Tn5 Transposase from kit), 22.5 µL nuclease-free H$_2$O] and incubated at 37 °C for 30 min. Immediately following the transposition reaction, DNA was purified using a Qiagen MinElute PCR purification Kit (Qiagen, 28004). Transposed DNA was eluted in 10 µL elution buffer (Buffer EB from the MinElute kit consisting of 10 mM Tris·Cl, pH 8). Paired-end 2 x 150 bp sequencing was performed using Illumina HiSeq X.

For ATACseq data analysis, we followed the ENCODE guidelines to process ATACseq sequence data. This includes quality control, alignment to the reference genome, peak calling, and functional analysis. 1. Quality control of raw data: We used FastQC version 0.11.8[90] to generate a report that includes various metrics such as read quality, GC content, and sequence duplication levels. Any reads that did not meet quality standards were removed at this stage. We also used cutadapt version 2.5 to remove adapter sequences. 2. Alignment to the reference genome: We aligned the reads to the mouse reference genome using BWA (Burrows-Wheeler Aligner) version 0.7.17-r1188 and adjusted the tag fragment positions[91]. 3. Peak calling: After alignment, we identified peaks where the regions of the genome are enriched for mapped reads. We used the ENCODE-recommended MACS2 (Model-based Analysis of ChIP-Seq) peak-calling algorithm[92–94]. For each experiment condition, we identified peaks using the IDR filtering criteria. Additionally, to detect the open chromatin regions specifically for the homogenous genotype group, we selected the wild-type mice as the control group and the mice with homogenous genotype as the treatment group in MACS2 version 2.2.4. 4. Peak annotation: We used TxDb.Mmusculus.UCSC.mm10.knownGene and the annotatePeak function in ChIPseeker version 1.36.0[95]. We obtained the gene list based on the annotation results. 5. Functional analysis: To understand the biological function of the genes in detected peaks, we performed functional analysis using GREAT version 2.2.0[53]. The annotation is based Gene Ontology (GO), which includes the genes and pathways that are associated with the peaks.

## RNA preparation, reverse transcription, and RT-PCR

RNA preparation, reverse transcription, and RT-PCR were performed as previously described[30]: Total RNA from FACS sorted TECs, endothelial cells, or tdTomato$^{high}$ cells from Ovol2$^{C120Y/C120Y}$, Ovol2$^{C120Y/-}$, or Ovol2$^{C102Y/C102Y}$;Foxn1-cre;Ai9, and WT littermates was prepared using TRIzol Reagent (Sigma–Aldrich), treated with RQ1 RNase-free DNase I (Promega) at 37 °C for 30 min to remove contaminating genomic DNA, and DNase I was inactivated by heating to 65 °C for 10 min. Equal amounts (200 ng) of RNA were used for reverse transcription using an oligo (dT) primer (Promega) and ImProm-II reverse transcriptase (Promega) following the manufacturer's instructions. RT-PCR was performed for relative quantification. The RT-PCR primer pairs used to detect splicing errors are listed in Table S1.

## CUT&RUN assay

Chromatin immunoprecipitation (ChIP) was performed using CUT&RUN assay Kit (Cell Signaling, #86652) following the manufacturer's instructions. CUT&RUN assay was done with 100,000 TECs sorted from 4.5-wk-old Ovol2$^{C120Y/C120Y}$ and WT littermates and Di-Methyl-Histone H3 (Lys4) antibody (Active Motif, AB_2614985) at 1:50 dilution. Precipitated DNAs were purified using phenol/chloroform extraction and ethanol precipitation. We quantified DNA by qPCR and normalized the data to the mouse RPL30 gene, for which primers were provided by the kit (Cell Signaling, #7015). Other RT-PCR primer pairs used to detect ChIP signal are listed in Table S1.

## Statistical analysis

Genetic linkage analysis was performed as described[27] and Bonferroni correction was applied to the alpha level. For all other experiments, the statistical significance of differences between groups was analyzed using GraphPad Prism 9 by performing the indicated statistical tests (Student's t-test for two experimental groups or ANOVA for three or more experimental groups followed by pairwise post-hoc comparisons). Differences in the raw values among groups were considered statistically significant when $P < 0.05$.

## Reporting summary

Further information on research design is available in the Nature Portfolio Reporting Summary linked to this article.

## Data availability

The raw ATACseq data generated in this study have been deposited in Sequence Read Archive with the BioProject accession no. PRJNA914728. The raw mass spectrometry data generated in this study have been deposited in ProteomeXchange with accession no. PXD039041. Source data are provided with this paper.

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

## Acknowledgements

The authors thank the UTSW Proteomics Core facility for assistance with proteomics experiments, and Diantha La Vine for illustration of the model in Fig. 8. This work was supported by National Institutes of Health grants AI125581 (B.B.).

## Author contributions

Conceptualization: X.Zhoung, N.P., J.H.C., B.B.; Methodology: X.Zhoung, N.P., J.H.C.; Formal analysis: J.J.M., X.Zhan Investigation: X.Zhong, N.P., J.W., J.M.S., J.A.S., K.K., D.R.L., J.H.C.; Visualization: X.Zhong, J.J.M., X.Zhan Funding acquisition: B.B.; Writing – original draft: X.Zhong, E.M.Y.M., B.B.; Writing – review & editing: X.Zhong, E.M.Y.M., B.B.

## Competing interests

The authors declare no competing interests.
