## [Peer review file · Nature Communications]

REVIEWER COMMENTS

Reviewer #1 (expert in TEC development and epigenetics):

The paper by X. Zhang and colleagues demonstrates a role for the *Ovol2* locus in the development and function of the haematopoietic system. In particular, the authors demonstrate with a series of elegant experiments that T cells are affected in mice with either a ubiquitous or a TEC-targeted compound heterozygosity or a C120Y missense and null allele of *Ovol2*. A major cause of this defect is secondary to a loss of medullary (m) and a dysfunction of cortical thymic epithelial cells (cTEC). The work continues to reveal the molecular role of *OVOL2* in cooperation with the BRAF-HDAC complex, especially the competitive interaction with the epigenetic regulator *LSD1*. In this context and via inhibition of *LSD1* function, *OVOL2* controls the epigenetic landscape, represses EMT and seemingly enforces an epithelial cell identity in the context of TEC differentiation and function.

This is a well-written, very data rich, and clearly presented set of experiments where the demonstrated results and their interpretation support the conclusions drawn. A set of observational and bone marrow transplantation studies nicely define TEC as the target cell population causing the observed T cell deficiency. The elucidation of the molecular mechanisms by which *OVOL2* interferes with the *LSD1:RCOR1* interaction is well demonstrated and provides a mechanistic explanation for the phenotype observed in these mutant mice. The inhibition of *LSD1* by *OVOL2* can therefore be further probed (see below) to explain the particular cellular and functional changes in mutant mice.

The data is novel and will significantly inform the field of immunology, haematology and developmental biology. Specifically, the work casts light on the complexities of TEC differentiation and function and provides yet another molecular mechanism that maintains TEC identity. Moreover, the presented study will further aid in addressing why differences of specific loss of function mutants differently affect cortical and medullary TEC.

There are, however, some central points of interest that would need to be addressed in more detail to fully understand the role of *OVOL2* in thymus biology (which constitutes the major focus of the submitted manuscript). The points are:

- The phenotype of thymus hypoplasia is shown to be only observed in mice older than 4 weeks of age. It would therefore be informative and necessary to demonstrate the expression of *OVOL2* in thymic stromal cells during the life course of mice ranging from before that point in time, e.g. a mid-gestation embryonic stage to adulthood beyond 4 weeks of age. This analysis should include an analysis of a possibly differential expression of *OVOL2* in cTEC and mTEC, best in different subpopulations within these lineages to better understand the distinct and in part separate impact of the mutation on cell

function and fate of each of these subpopulations. Whereas the discussion section provides some speculations why a phenotype is not observed prior to adulthood, this important point is not experimentally addressed nor underpinned by additional information despite its relevance to understand TEC fate and maintenance and thus the thymopoietic function of these cells.

- Given that mutant mice display a reduction (cTEC) or even absence (mTEC) of parts of the epithelial scaffold in 12-week old animals that is molecularly linked to a loss of EMT inhibition, it would be critical to show how the non-TEC stroma, in particular its mesenchymal components, are altered in gene targeted mice, especially those older than 4 weeks of age and with the *Foxn1-Cre::OVOL2^{fl/fl}* mutation. An analysis of this crucial cell compartment would be particularly helpful in employing markers that identify mesenchymal cells to have originated from TEC via EMT (which could be demonstrated, for example, with *Foxn1-cre; OVOL2^{fl/fl}; Rosa-stop-RFP* mice). An analysis of this kind would allow to detail which changes of the non-TEC stromal compartment are the direct consequence of altered EMT involving TEC.

- The authors write that *OVOL2* is seemingly not required for the initial specification of TEC identity (page 11) but then also remark that '*OVOL2* might also contribute to TEC development before the period of *Foxn1* expression'. This aspect could indeed be relevant to explain why the severity of the thymus phenotype differs between *OVOL2 C120Y/-* and TEC-restricted *OVOL2* null mutants. Further molecular evidence should be provided to support this contention, or, alternatively, the corresponding section in the Discussion would need to be reworded and another explanation should be given for a role of *OVOL2* in thymus development that is TEC independent.

- Defective early thymocyte development may be caused by factors/mechanisms other than the impaired cellularity, differentiation and function of the TEC scaffold. Changes in the bone marrow microenvironment are inferred by the authors to account for this observation. What is the direct molecular evidence that supports this assertion and explains the difference observed between *OVOL2 C120Y/-* and *Foxn1-Cre::OVOL2^{FL/FL}* mice. In this context, a comparative analysis of T cell functions between these two mouse models is also missing.

- The authors nicely demonstrate changes in *OVOL2*- controlled chromatin accessibility for hundreds of loci in mutant mice. To directly link these changes to differences in epigenetic marks as placed by *BRAF-HDAC*'s altered function, data would need to be demonstrated of changes in at least *H3K4me2* marks at the loci with altered chromatin configuration and gene expression.

In addition to the above points, the authors should address/ comment on the following specific points that concern individual datasets in the indicated figures of the manuscript:

Figure 1G: Which specific subpopulation of single CD8 positive thymocytes is increased in the mutant mice as the demonstrated phenotype could concern immature (i.e. CD3 negative, pre DP stage) and/or mature (CD3 positive) CD8+ thymocytes.

Figure 2G + H: Absolute cell numbers should be provided.

Figure 2I: Could the authors explain why in this dataset the frequency of mutant B cells is not different, whereas in experiments with mutant bone marrow cells transferred into Rag deficient recipients (panel B) the reconstitution of the B cell compartment is impaired.

Figure 3 A-H: Total cellularity of the thymus and the cellularities of thymocytes, TEC and non-TEC stromal cells should be displayed for both 4 and 12 week old, wild type and mutant animals.

Figure 3B: H+E stains represent cell density but do not necessarily reveal the presence and location of the individual subpopulations of TEC. Hence, staining thymus tissue sections for different cytokeratins (CK) distinguishing cortical (CK8) and medullary (CK5, CK14) TEC subsets would be necessary to detail the architectural organisation of the TEC scaffold and to exclude that mTEC are absent as their recovery is typically diminished when isolated from thymus tissue to obtain single cell suspensions. Moreover, is the increase in adipose tissue also observed in TEC targeted mutant mice (Foxn1-Cre::OVOL2FL/FL).

Figure 3F: Given the data regarding Fezf2 and Aire expression (figure 3P), individual mTEC subpopulations will need to be better detailed, using in addition to UEA1 reactivity and MHCII expression for example also antibodies specific for Sca1, CD80, CD86, and intracytoplasmic Aire. This information would reveal whether the cellularity of mature mTEC expressing Aire is reduced, the frequency decreased and/or the expression levels of Aire among mTEC diminished. This information has implications on the author's conclusions that mutant mice do not display signs of autoimmunity (which is unfortunately only documented by serum antibody concentrations but not tissue sections revealing a lack of lymphocytic infiltrates).

Figure 3I: A comparative analyses of non-epithelial stromal cells between wild type, OVOLC120Y/- and Foxn1-Cre::Ovol2fl/fl mice would explain whether the observed differences are caused by a loss of OVOL2 function in mesenchymal cells such as fibroblasts (see also remark above). Data from such an analysis will also support a specific point that the authors made in their discussion but for which no data is provided.

Figure 3L: Data is provided for an upregulated expression of CD44 in all thymocytes at and beyond the double positive developmental stage. The authors should discuss/explain this finding and relate it to extent of positive thymocyte selection as measured for example by the concurrent upregulation of CD3 and CD69 or CD5.

Figure 3P: It would be important to demonstrate whether the reduced expression of the transcriptional facilitators Aire and Fezf2 results in diminished transcripts of tissue restricted antigens in mature mTEC

and the reduced detection of Aire and Fezf2 transcripts will need to be related to the frequency of cells that physiologically express these two factors most prominently (i.e., CD80+, CD86+, MHCIIhigh, mTEC).

Reviewer #2 (expert in thymic function and TEC development):

This manuscript describes a novel function of the transcription factor OVOL2 in postnatal maintenance of the thymus function, based on immune cell screening of a large-scale ENU mutagenesis in mice. Results show that the ENU-induced C120Y mutation in OVOL2 causes a defect in immune cell development, including T cells and B cells. Interestingly, the severe defect in T cell development is ascribed to defective function of thymic stromal cells. Foxn1-Cre-mediated OVOL2 deletion phenocopies the ENU mutants in causing defective T cell development. Results are also provided to show the interaction of OVOL2 protein with BRAF-HDAC complex and OVOL2-mediated negative regulation of BRAF-HDAC activity. It is further suggested that OVOL2 regulates epithelial-mesenchymal transition (EMT) in postnatal thymic epithelial cells. Postnatal maintenance of the thymus function is an important topic in basic and clinical immunology. The discovery that OVOL2 is a key regulator of postnatal thymus involution is interesting. However, the manuscript suffers from the following uncertainties to draw the major conclusion that OVOL2 sustains TEC identity by preventing EMT.

The manuscript describes that the medullary region and mTECs are severely affected in the OVOL2 mutant mice and in Foxn1-Cre-mediated OVOL2-deleted mice. On the contrary, the effects to the thymic cortex or cTECs are not clearly presented. cTECs rather than mTECs play a major role in thymic T cell production by promoting T cell specification in lymphoid progenitors and inducing positive selection of newly generated T cells, so that the defective T cell production is most likely due to some problems in cTECs rather than mTECs. The manuscript describes that cTECs are not reduced in number but should provide more in details how OVOL2 deficiency may affect cTECs. Are cTECs reduced in T cell specifying molecules, such as DLL4, IL-7, SCF, and CXCL12, and/or positive selection inducing molecules, such as MHC-I/II, Psmb11, cathepsin L, Prss16, and CD83? Does OVOL2 deficiency postnatally reduce Foxn1 in cTECs?

The lack in the thymic medulla (Fig. 3B and 3C) is not convincing and should be supported by keratin 5 and/or keratin 14 immunohistological analysis. Despite the lack in the medulla and the reduction in Aire and Fezf2, the lack of autoimmunity described in the manuscript is a surprise, because T cell self-tolerance is the best-known function of the thymic medulla. The description of “no lymphocytic infiltration of tissues” should be supported by showing the results.

Foxn1-Cre is not a TEC-specific deleter but deletes floxed genes in TECs and skin cells, although a TEC-specific deleter is available (PMID: 35042581). The thymus involution late in the life could be secondary to systemic stress due to the problems in skin cells. Is the skin and hair cells normal without signs of lesions in those animals?

Purity and cTEC/mTEC profiles of isolated TECs for biochemical analysis should be shown (Fig 4, Fig. 6). Do those data from isolated TECs primarily reflect the biology of mTECs rather than cTECs?

Regulation of EMT is an important conclusion of the study and should be more directly evaluated by immunohistological visualization of EMT in the thymic sections (e.g. PMID: 19648267).

Reviewer #3 (expert in T cell development and transcriptional regulation of T cell fate):

The authors study the effects of loss of *Ovol2*, previously implicated in maintaining expression of epithelial lineage genes. Mice with germline defects in *Ovol2* were shown to have profound defects in T and B lymphopoiesis, as well as NK cell effector function, and reciprocal bone marrow chimeras indicated the requirement for *Ovol2* is extrinsic to hematopoietic lineage cells. The authors evaluate the requirement for *Ovol2* specifically in thymic epithelial cells (TEC) using a conditional allele crossed to *Foxn1-Cre*. Interestingly, these mice have normal thymus size initially, and normal frequencies of TEC populations, but the thymus progressively declines in size from 4 to 20 weeks, with an apparent loss of mTEC populations, and loss of *Aire* and *Fezf2* transcription factor expression (genes associated with establishment of self-tolerance) in mTECs. The authors subsequently perform biochemical analyses in EL4 and HeK293T cells, and their studies support a model where *Ovol2* is required to displace *Lsd1* from interactions with a complex containing *Rco1*, *Lsd1* and HDAC. In the absence of *Ovol2*, there is increased association of *Lsd1* in this complex, and increased histone H3K4 demethylation. This model is consistent with ATAC-seq data showing increased open chromatin in thymic epithelial cells of *Ovol2* deficient mice. Because genes altered in expression in *Ovol2* deficient thymus include genes implicated in epithelial-mesenchymal transition, the authors conclude that *Ovol2* is implicated in thymic epithelial cell identity.

The work that is performed here is certainly of interest. However, there are areas where significant improvement is needed. The analysis of thymic epithelial cell diversity and T cell development in *Foxn1-Cre*, *Ovol2^{f/f}* mice needs to be significantly enhanced. Autoimmunity is expected in mice with cortical TEC but lacking medullary TEC, and the authors pay almost no attention to this point except to remark that the mice show no signs of autoimmunity. The authors do not directly show that TEC lacking *Ovol2* undertake EMT, as opposed to dying of apoptosis.

In more detail - levels of Kit and CD25 on DN subsets should be assessed to determine whether reduced frequencies of ETP are evident, indicating defects in thymic settling contribute to the observed phenotype, perhaps due to reduced expression of chemokines by TEC (for example, see Zlotoff et al., *Blood*, 2010). Defects in negative selection should be assessed, minimally using markers for stages of T cell development (for example, see Baran-Gale, *eLife*, 2020). Defects in immune tolerance should be assessed (Korai et al., *JEM*, 2017), as expected in these mice; or an explanation why autoimmunity is not evident is needed. Finally, it is difficult to assess apoptosis in TEC, but Foxn1-Cre based lineage-tracing can establish whether TEC adopt other mesenchymal cell fates in the absence of *Ovol2*. This needs to be performed, and given the answers, the proposed mechanism should be discussed considering the possibilities. With these improvements, I think this will be a fine contribution.

REVIEWER COMMENTS

Reviewer #1 (expert in TEC development and epigenetics):

We appreciate the reviewer's comments. The following are our point-by-point responses:

The paper by X. Zhang and colleagues demonstrates a role for the *Ovol2* locus in the development and function of the haematopoietic system. In particular, the authors demonstrate with a series of elegant experiments that T cells are affected in mice with either a ubiquitous or a TEC-targeted compound heterozygosity or a C120Y missense and null allele of *Ovol2*. A major cause of this defect is secondary to a loss of medullary (m) and a dysfunction of cortical thymic epithelial cells (cTEC). The work continues to reveal the molecular role of *OVOL2* in cooperation with the BRAF-HDAC complex, especially the competitive interaction with the epigenetic regulator LSD1. In this context and via inhibition of LSD1 function, *OVOL2* controls the epigenetic landscape, represses EMT and seemingly enforces an epithelial cell identity in the context of TEC differentiation and function.

This is a well-written, very data rich, and clearly presented set of experiments where the demonstrated results and their interpretation support the conclusions drawn. A set of observational and bone marrow transplantation studies nicely define TEC as the target cell population causing the observed T cell deficiency. The elucidation of the molecular mechanisms by which *OVOL2* interferes with the LSD1:RCOR1 interaction is well demonstrated and provides a mechanistic explanation for the phenotype observed in these mutant mice. The inhibition of LSD1 by *OVOL2* can therefore be further probed (see below) to explain the particular cellular and functional changes in mutant mice.

The data is novel and will significantly inform the field of immunology, haematology and developmental biology. Specifically, the work casts light on the complexities of TEC differentiation and function and provides yet another molecular mechanism that maintains TEC identity. Moreover, the presented study will further aid in addressing why differences of specific loss of function mutants differently affect cortical and medullary TEC.

There are, however, some central points of interest that would need to be addressed in more detail to fully understand the role of *OVOL2* in thymus biology (which constitutes the major focus of the submitted manuscript). The points are:

1. The phenotype of thymus hypoplasia is shown to be only observed in mice older than 4 weeks of age. It would therefore be informative and necessary to demonstrate the expression of *OVOL2* in thymic stromal cells during the life course of mice ranging from before that point in time, e.g. a mid-gestation embryonic stage to adulthood beyond 4 weeks of age. This analysis should include an analysis of a possibly differential expression of *OVOL2* in cTEC and mTEC, best in different subpopulations within these lineages to better understand the distinct and in part separate impact of the mutation on cell function and fate of each of these subpopulations. Whereas the discussion section provides some speculations why a phenotype is not observed prior to adulthood, this important point is not experimentally addressed nor underpinned by additional information despite its relevance to understand TEC fate and maintenance and thus the thymopoietic function of these cells.

To understand whether developmentally regulated *Ovol2* expression might contribute to the age-dependency of thymus hypoplasia and the reduced frequency and number of mTEC, we used RT-PCR to measure *Ovol2* gene expression in TEC subsets at several ages during embryogenesis and postnatally. We added these data to Fig. 4A and 4B. In cTECs, the relative mRNA level of *Ovol2* was lowest embryonically during thymic organogenesis and increased with age until it peaked at 12 weeks of age (Fig. 4A). In mTECs,

relative *Ovo/2* transcript expression was low at E13.5 and P0, peaked at 4 weeks of age, and returned to E13.5/P0 levels when tested at 12, 20, and 30 weeks of age (Fig. 4A). When we compared *Ovo/2* mRNA levels in cTECs vs. mTECs, we observed that mTECs and cTECs had comparable *Ovo/2* expression at E13.5, but cTECs displayed higher *Ovo/2* transcript levels than mTECs postnatally (Fig. 4B). The progressive increase in *Ovo/2* expression between E13.5 and 4 weeks (in mTECs) or between E13.5 and 12 weeks (in cTECs) is consistent with the age dependent phenotypes of TECs in *OVOL2* mutant mice. These data also imply an important role for *OVOL2* in the maintenance of cTEC function despite a lack of effect on cTEC numbers.

- Given that mutant mice display a reduction (cTEC) or even absence (mTEC) of parts of the epithelial scaffold in 12-week old animals that is molecularly linked to a loss of EMT inhibition, it would be critical to show how the non-TEC stroma, in particular its mesenchymal components, are altered in gene targeted mice, especially those older than 4 weeks of age and with the *Foxn1-Cre::OVOL2^{fl/fl}* mutation. An analysis of this crucial cell compartment would be particularly helpful in employing markers that identify mesenchymal cells to have originated from TEC via EMT (which could be demonstrated, for example, with *Foxn1-cre; OVOL2^{fl/fl}; Rosa-stop-RFP* mice). An analysis of this kind would allow to detail which changes of the non-TEC stromal compartment are the direct consequence of altered EMT involving TEC.

We appreciate this valuable suggestion. To understand whether mesenchymal cells originated from TECs in *Ovo/2^{C120Y/C120Y}* thymi, we generated mice in which developing TECs were marked with the fluorescent reporter tdTomato. These mice (*Ovo/2^{C120Y/C120Y};Foxn1-Cre;Ai9*) express Ai9, a Cre reporter knocked into the *Rosa26* locus, which contains the *tdTomato* gene separated from a CAG promoter by a loxP-flanked STOP cassette. CAG promoter-driven expression of tdTomato is activated by Cre-mediated deletion of the STOP cassette. Driven by the *Foxn1* promoter, Cre expression activates tdTomato fluorescence in TEC progenitors and keratinocytes. Note the pink color of the skin in Ai9+ mice.

We analyzed the *tdtomato*^{high} TEC population and **present these new data below and in new Fig. 5C-G.** We observed that ~80% of *tdtomato*^{high} cells in WT thymi were epithelial cells, whereas only ~26% were epithelial cells in *Ovol2*^{C120Y/C120Y} thymi from 20-week-old mice (Fig. 5C-F). Moreover, ~50% of *Ovol2*^{C120Y/C120Y} *tdtomato*^{high} cells were CD31-EpCAM-PDGFR- α/β ⁺ cells; i.e. mesenchymal cells, in comparison to only ~4% of WT *tdtomato*^{high} cells (Fig. 5C-F). Endothelial cell frequencies among *tdtomato*^{high} cells were slightly elevated in *Ovol2*^{C120Y/C120Y} thymi compared to WT thymi (Fig. 5C-F). These data definitively demonstrate that in *Ovol2*^{C120Y/C120Y} thymi, TEC were converted to mesenchymal cells, which was also corroborated by analysis of epithelial and mesenchymal gene expression (Fig. 5G).

We note that *Foxn1-Cre;Ai9* mice on a WT background contain a population of cells with intermediate $tdTomato^{int}$ that is not detected in *Ai9* mice lacking Cre expression (figure below, and Supplementary Fig. 12). We found that the population of $tdTomato^{int}$ cells in *Foxn1-Cre;Ai9* thymi was >99.8% Lin+CD45+ (marking cells of hematopoietic origin), while the remaining 0.17% $tdTomato^{int}$ Lin-CD45- cell population was not thymic stromal cells, in that they did not express surface markers of TECs, endothelial, or mesenchymal cells. The specific expression of Foxn1-Cre in TECs and keratinocytes has been reported dozens of times. Our bone marrow transplantation and adoptive transfer experiments (Fig. 2), and the epithelial-specific phenotypes observed in *Foxn1-Cre;Ovol2^{fl/fl}* mice strongly support that a hematopoietic- and T cell-extrinsic defect impairs T cell development in *Ovol2^{C120Y/-}* mice. Thus, our analyses of marked TECs focused on the clearly separated population of $tdTomato^{high}$ cells that had appropriate epithelial characteristics.

Further support that mesenchymal cells originated from TEC via EMT in *Foxn1-cre;Ovol2^{fl/fl}* mice included the finding that thymi from *Foxn1-cre;Ovol2^{fl/fl}* mice were surrounded by adipose tissue (black arrows) similar to thymi in *Ovol2^{C120Y/-}* mice (figure below, and now in Fig. 3K and Supplementary Fig. 6C). In *Foxn1-cre;Ovol2^{fl/fl}* mice this adipose tissue was only present around the thymus, whereas in *Ovol2^{C120Y/-}* mice the adipose tissue was present throughout the body. This strongly suggests that deletion of OVOL2 in TECs results in the conversion of TECs to mesenchymal cells, specifically adipocytes, which accumulated around the thymus.

- The authors write that OVOL2 is seemingly not required for the initial specification of TEC identity (page 11) but then also remark that 'OVOL2 might also contribute to TEC development before the period of *Foxn1* expression'. This aspect could indeed be relevant to explain why the severity of the thymus phenotype differs between OVOL2 *C120Y/-* and TEC-restricted OVOL2 null mutants. Further molecular evidence should be provided to support this contention, or, alternatively, the corresponding section in the Discussion would need to be reworded and

another explanation should be given for a role of OVOL2 in thymus development that is TEC independent.

By analyzing the non-TEC stromal components in *Ovol2*^{C120Y/C120Y} and *Ovol2*^{+/+} mice, and in *Ovol2*^{fl/fl} and *Foxn1-Cre;Ovol2*^{fl/fl} mice (based on cell type-specific markers in PMID: 26947077 and PMID: 22504647), we obtained evidence that deficiency of OVOL2 in a non-TEC cell type impacts the cellular makeup of *Ovol2*^{C120Y/-} thymi. We found drastically increased frequencies (Supplementary Fig. 7A-D) and numbers (Fig. 4D) of endothelial cells and mesenchymal cells in *Ovol2*^{C120Y/-} thymi. In contrast, *Foxn1-cre;Ovol2*^{fl/fl} mice exhibited increased proportions (Supplementary Fig. 7I-L) and numbers (Fig. 4E) of mesenchymal cells, but endothelial cells were normal. These data strongly suggest that the increase in endothelial cells in *Ovol2*^{C120Y/-} thymi is a consequence of OVOL2 deficiency in cell type(s) other than TECs. Moreover, these endothelial cells were likely to be dysfunctional based on their diminished expression of the chemokine genes *P-selectin* and *Ccl25* (Fig. 4M). Thus, non-TEC cells lacking OVOL2 may alter thymic development by causing aberrant accumulation of endothelial cells. **We present these new data below and in Fig. 4C-E and Supplementary Fig. 7A-L.** *Foxn1* expression is initiated on E8, remains high during embryogenesis, and rapidly diminishes after birth. Since we found that *Ovol2* expression is instead lowest in the embryo (E13.5), increases postnatally, and is highest at 4 weeks (mTEC) or 12 weeks of age (cTEC), it is unlikely that OVOL2 contributes to TEC development before the period of *Foxn1* expression. We have revised the Discussion accordingly.

- Defective early thymocyte development may be caused by factors/mechanisms other than the impaired cellularity, differentiation and function of the TEC scaffold. Changes in the bone marrow microenvironment are inferred by the authors to account for this observation. What is the direct molecular evidence that supports this assertion and explains the difference observed between *OVOL2*^{C120Y/-} and *Foxn1-Cre::OVOL2FL/FL* mice. In this context, a comparative analysis of T cell functions between these two mouse models is also missing.

In support of the possibility that bone marrow stromal cells in *Ovo2*^{C120Y/-} mice might contribute to defects in thymocyte development is the finding that transfer of bone marrow HSCs from WT mice to *Ovo2*^{C120Y/-} recipient mice resulted in reduced frequencies of CLP (Supplementary Fig. 4G) and in B cell developmental defects (Supplementary Fig. 4A and 4B) in the bone marrow when compared to transfer of WT bone marrow to WT mice. These data suggest an aberrant bone marrow stroma in *Ovo2*^{C120Y/-} mice. **We present these new data below and in Supplementary Fig. 4G.** Moreover, gross defects in the cellular composition of the bone marrow of *Ovo2*^{C120Y/-} mice were observed by histology, including excessive adipocytes that filled the bone marrow cavity. These data suggest that bone marrow stromal defects exist in these mice.

We detected an impaired T cell-dependent IgG response to immunization with ovalbumin precipitated on alum adjuvant (alum-ova) in 20-week-old *Foxn1-Cre;Ovo2*^{fl/fl} mice (Fig. 3O), similar to the response in *Ovo2*^{C120Y/-} mice. This finding indicates that *OVOL2* deletion in TECs is sufficient to cause defective CD4 T cell function. **We present the data here and have added them in Fig. 3O:**

- The authors nicely demonstrate changes in *OVOL2*- controlled chromatin accessibility for hundreds of loci in mutant mice. To directly link these changes to differences in epigenetic marks as placed by *BRAF-HDAC*'s altered function, data would need to be demonstrated of changes in at least H3K4me2 marks at the loci with altered chromatin configuration and gene expression.

To directly link chromatin accessibility changes to epigenetic marks, we used a CUT&RUN assay to perform chromatin immunoprecipitation (ChIP) using an H3K4me2 antibody and *Ovo2*^{+/+} and *Ovo2*^{C120Y/C120Y} TECs sorted from 4.5-week-old mice. By quantitative PCR, we measured precipitated DNA representing 45 loci

with altered chromatin configuration (detected by ATACseq, Fig. 8A and 8B and Supplementary Datasets 2 and 3) and gene expression (detected by RT-PCR, Fig. 8C and 8D) in *Ovo12*^{C120Y/C120Y} TECs. Among these 45 loci, 25 were accessible in *Ovo12*^{+/+} TECs but not in *Ovo12*^{C120Y/C120Y} TECs, and 23 out of the 25 were more abundant in H3K4me2 immunoprecipitates from *Ovo12*^{+/+} TECs than from *Ovo12*^{C120Y/C120Y} TECs (Supplementary Fig. 16A). Among the 45 loci, 20 loci were accessible only in *Ovo12*^{C120Y/C120Y} TECs, and 18 out of the 20 were more abundant in H3K4me2 immunoprecipitates from *Ovo12*^{C120Y/C120Y} TECs than from *Ovo12*^{+/+} TECs (Supplementary Fig. 16B). These data suggest that altered chromatin accessibility in *Ovo12*^{C120Y/C120Y} TECs is linked to the change of epigenetic marks on H3K4. **We present the data below and in Supplementary Fig. 16.**

In addition to the above points, the authors should address/ comment on the following specific points that concern individual datasets in the indicated figures of the manuscript:

6. Figure 1G: Which specific subpopulation of single CD8 positive thymocytes is increased in the mutant mice as the demonstrated phenotype could concern immature (i.e. CD3 negative, pre DP stage) and/or mature (CD3 positive) CD8+ thymocytes.

We analyzed single CD8 positive thymocytes for cell surface expression of TCR-β.

We found that the proportion of mature T cells (CD8+TCR-β+ cells among CD8+ SP thymocytes) was elevated in the *Ovol2^{bah/-}* mice, reflecting the reduced frequency of newly produced T cells (above and Supplementary Fig. 1D). The data demonstrate that the phenotype is caused by the shortage of newly produced T cells. **We added the data in Supplementary Fig. 1D.**

7. Figure 2G + H: Absolute cell numbers should be provided.

We combined original Figs. 2G and 2H and **now provide the absolute cell numbers in new Fig. 2H:**

8. Figure 2I: Could the authors explain why in this dataset the frequency of mutant B cells is not different, whereas in experiments with mutant bone marrow cells transferred into Rag deficient recipients (panel B) the reconstitution of the B cell compartment is impaired.

We thank the reviewer for pointing out these contrasting findings. The data in Fig. 2A, 2B, and Supplementary Fig. 4A-C (left panel) showed that *OVOL2* deficiency had a hematopoietic intrinsic effect on B cell development, resulting in a decreased proportion of mature B cells in bone marrow, increased proportion of splenic marginal zone B cells, and decreased frequency of follicular B cells in the spleen. In those experiments (Fig. 2A, 2B, and Supplementary Fig. 4A-C left panel), we injected 100% WT or 100% *Ovol2^{C120Y/-}* bone marrow to individual recipients. However, in original Fig. 2I, we injected a mixture of 50% WT and 50% *Ovol2^{C120Y/-}* bone marrow into individual recipients. In the same environment, WT and *Ovol2^{C120Y/-}* bone marrow had similar abilities to differentiate to mature B cells in *Rag2^{-/-}* recipients. The difference in mature B cell reconstitution in Fig. 2A-B versus original Fig. 2I might be due to the effect of stromal cells from *Ovol2^{C120Y/-}* bone marrow on WT HSC in the 50:50 BMT experiment. We speculate that *Ovol2^{C120Y/-}* stromal cells in bone marrow might inhibit WT HSC differentiation into mature B cells. Thus, within an individual recipient, WT and *Ovol2^{C120Y/-}* HSC showed similar B cell differentiation.

We repeated the experiments of Fig. 2A-B and original Fig. 2I and checked the peripheral blood in the recipients 2 months after BMT. We obtained data similar to our original findings. Since the contrasting findings might be confusing, we decided to delete the original Fig. 2I.

9. Figure 3 A-H: Total cellularity of the thymus and the cellularities of thymocytes, TEC and non-TEC stromal cells should be displayed for both 4 and 12 week old, wild type and mutant animals.

We analyzed the cellularity of the thymus, thymocytes, TEC and non-TEC stromal cells (mesenchymal + endothelial components) in 4-week-old or 12-week-old *Ovo12^{C120Y/-}* and *Ovo12^{+/+}* mice (based on markers in PMID: 26947077 and PMID: 22504647). We present the data in Fig. 3F and 3I.

Based on the results, we discovered that non-TEC stromal cells were elevated and TECs were reduced in 12-week-old *Ovo12^{C120Y/-}* thymi (Fig. 3F). At 4 weeks of age, the numbers of TECs and non-TEC stromal cells were normal in *Ovo12^{C120Y/-}* mice (Fig. 3I).

10. Figure 3B: H+E stains represent cell density but do not necessarily reveal the presence and location of the individual subpopulations of TEC. Hence, staining thymus tissue sections for different cytokeratins (CK) distinguishing cortical (CK8) and medullary (CK5, CK14) TEC subsets would be necessary to detail the architectural organisation of the TEC scaffold and to exclude that mTEC are absent as their recovery is typically diminished when isolated from thymus tissue to obtain single cell suspensions.

We used immunohistochemistry to analyze cytokeratins in *Ovol2*^{C120Y/-} and *Foxn1-cre;Ovol2*^{fl/fl} thymi. **We present the data below and have added them in Supplementary Fig. 5A-C.** Cytokeratin (CK) 8 clearly labeled the cortical regions and was absent from the medullary regions of WT thymus, while the ubiquitous CK8 staining in the *Ovol2*^{C120Y/-} and *Foxn1-cre;Ovol2*^{fl/fl} thymi suggested that most of the medullary epithelium was lost (Supplementary Fig. 5A). CK5 and CK14 labeled the medullary regions and were absent from the cortical regions of WT thymus, while in *Ovol2*^{C120Y/-} and *Foxn1-cre;Ovol2*^{fl/fl} thymi the weak and diffuse ubiquitous staining suggested low levels of these epitopes (Supplementary Fig. 5B and 5C). Note that for WT samples, sections of a portion of the thymus are shown; for mutant samples, sections of the whole thymus are shown. These data support the conclusions in our original manuscript. However, we revised our statement in the Results to say, “*Ovol2*^{C120Y/-} thymi ... had a greatly reduced medullary region (Fig. 3B).”

11. Moreover, is the increase in adipose tissue also observed in TEC targeted mutant mice (Foxn1-Cre::OVOL2FL/FL).

Thymi from *Foxn1-cre;Ovol2^{fl/fl}* mice were surrounded by adipose tissue (black arrows) similar to thymi in *Ovol2^{C120Y/-}* mice (figure below, and now in Fig. 3K and Supplementary Fig. 6C). In *Foxn1-cre;Ovol2^{fl/fl}* mice this adipose tissue was only present around the thymus, whereas in *Ovol2^{C120Y/-}* mice the adipose tissue was present throughout the body. This strongly suggests that deletion of OVOL2 in TECs results in the conversion of TECs to mesenchymal cells, specifically adipocytes, which accumulated around the thymus.

12. Figure 3F: Given the data regarding Fezf2 and Aire expression (figure 3P), individual mTEC subpopulations will need to be better detailed, using in addition to UEA1 reactivity and MHCII expression for example also antibodies specific for Sca1, CD80, CD86, and intracytoplasmic Aire. This information would reveal whether the cellularity of mature mTEC expressing Aire is reduced, the frequency decreased and/or the expression levels of Aire among mTEC diminished.

As suggested by the reviewer, we used FACS to analyze the cellularity of mTECs expressing Aire and intracytoplasmic Aire expression in total mTECs (UEA1+) and mature mTECs (mTEC^{high}, UEA1+ MHC-II^{high}) from *Ovol2^{C120Y/C120Y}* mice. Compared to WT littermates, the cellularity of mTECs expressing Aire was decreased in *Ovol2^{C120Y/-}* mice (Fig. 3G). Protein levels of AIRE (intracellular) were reduced in mTECs and were normal in mature mTECs (UEA1+ MHC-II^{high}) from *Ovol2^{C120Y/C120Y}* mice (Fig. 4T). **We added these data in Fig. 3G and 4T.**

Also, we detected CD80 and CD86 expression on the surface of *Ovo12*^{C120Y/C120Y} mTECs or mature mTECs (UEA1+ MHC-II^{high}).

Cell surface levels of CD80 and CD86 were also reduced in mTECs and mature mTECs from *Ovo12*^{C120Y/C120Y} mice. **We added the data in Fig. 4U-V.**

13. This information has implications on the author’s conclusions that mutant mice do not display signs of autoimmunity (which is unfortunately only documented by serum antibody concentrations but not tissue sections revealing a lack of lymphocytic infiltrates).

We conducted further analyses for signs of autoimmunity in *Ovo12*^{C120Y/C120Y} mice. **The new data are presented below and in Supplementary Fig. 11.** *Ovo12*^{C120Y/C120Y} mice displayed normal antibody titers (Supplementary Fig. 11A). *Ovo12*^{C120Y/C120Y} mice had severe lymph node and spleen hypoplasia (Supplementary Fig. 11B and 1B). *Ovo12*^{C120Y/C120Y} mice had normal percentages of CD4+FoxP3+ Treg cells in the lymph node and spleen, although they had decreased numbers of these cells because of lymph node and spleen hypoplasia (Supplementary Fig. 11C-D). Also, there were no signs of lymphocytic infiltration into the kidney, lung, liver, intestine, and salivary gland (Supplementary Fig. 11E-I). In contrast, there were decreased proportions of lymphocytes (CD45+), especially CD19+ and CD3+ cells in several organs, and *Ovo12*^{C120Y/-} mice had reduced B cell and T cell numbers. Finally, as previously reported (PMID: 36228616), H&E staining showed no lymphocytic infiltration of the liver and skin in *Ovo12*^{C120Y/-} mice. Therefore, we concluded that *Ovo12*^{C120Y/-} mice do not have autoimmunity.

14. Figure 3: A comparative analyses of non-epithelial stromal cells between wild type, *OVOLC120Y*^{-/-} and *Foxn1-Cre::Ovol2*^{fl/fl} mice would explain whether the observed differences are caused by a loss of *OVOL2* function in mesenchymal cells such as fibroblasts (see also remark above). Data from such an analysis will also support a specific point that the authors made in their discussion but for which no data is provided.

As suggested by the reviewer here and in comment 3, we analyzed the non-TEC stromal components in *Ovol2*^{C120Y/C120Y} and *Ovol2*^{+/+} mice, and in *Ovol2*^{fl/fl} and *Foxn1-Cre;Ovol2*^{fl/fl} mice (based on markers in PMID: 26947077 and PMID: 22504647). We present these new data again below and in Fig. 4C-E and Supplementary Fig. 7A-L.

We discovered that 4-week-old *Ovol2*^{C120Y/-} mice had normal frequencies (Supplementary Fig. 7E-H) and numbers (Fig. 4C) of endothelial and mesenchymal cell populations in the thymus, when TECs were also normal in number. At 12 weeks of age, *Ovol2*^{C120Y/-} thymi had drastically increased frequencies (Supplementary Fig. 7A-D) and numbers (Fig. 4D) of endothelial cells and mesenchymal cells. Importantly, 20-week-old *Foxn1-cre;Ovol2*^{fl/fl} mice also exhibited increased proportions (Supplementary Fig. 7I-L) and numbers (Fig. 4E) of mesenchymal cells, but endothelial cells were normal. These data strongly suggest that TEC-specific *OVOL2* deficiency results in both fractional and numerical increases in thymic mesenchymal cells at the expense of epithelial cells. In contrast, the increase in endothelial cells in *Ovol2*^{C120Y/-} thymi is a consequence of *OVOL2* deficiency in cell type(s) other than TECs.

15. Figure 3L: Data is provided for an upregulated expression of CD44 in all thymocytes at and beyond the double positive developmental stage. The authors should discuss/explain this finding and relate it to extent of positive thymocyte selection as measured for example by the concurrent upregulation of CD3 and CD69 or CD5.

We analyzed thymocyte positive selection defined by CD5 or CD69 and TCR-β cell surface expression in 12-week-old or 4-week old *Ovol2*^{C120Y/-} mice and their WT littermates, and in 20-week-old *Foxn1-Cre;Ovol2*^{fl/fl} mice and *Ovol2*^{fl/fl} littermates. We present the data here and in Supplementary Fig. 9.

We discovered decreased percentages of T cells undergoing positive selection in 12-week-old *Ovo12^{C120Y/-}* mice, with milder impairments observed in 4-week-old *Ovo12^{C120Y/-}* mice (Supplementary Fig. 9A-F). 20-week-old *Foxn1-cre;Ovo12^{fl/fl}* mice phenocopied *Ovo12^{C120Y/-}* mice (Supplementary Fig. 9G-I). The data demonstrate defective positive selection in *Ovo12^{C120Y/-}* mice and in mice with TEC-targeted OVOL2 deletion.

16. Figure 3P: It would be important to demonstrate whether the reduced expression of the transcriptional facilitators Aire and Fezf2 results in diminished transcripts of tissue restricted antigens in mature mTEC and the reduced detection of Aire and Fezf2 transcripts will need to be related to the frequency of cells that physiologically express these two factors most prominently (i.e., CD80+, CD86+, MHCIIhigh, mTEC).

We detected normal frequencies of mTEC^{high} (UEA-1+ MHC-II^{high}) in *Ovo12^{C120Y/-}* thymi (Fig. 3E), although numbers of these cells were reduced (Fig. 3G). The mTEC^{high} cells in *Ovo12^{C120Y/-}* thymi displayed normal expression of intracellular AIRE (Fig. 4T), but reduced surface expression of CD80 and CD86, as shown below and in new Fig. 4U-V:

Reviewer #2 (expert in thymic function and TEC development):

We appreciate the reviewer's comments. The following are our point-by-point responses:

This manuscript describes a novel function of the transcription factor OVOL2 in postnatal maintenance of the thymus function, based on immune cell screening of a large-scale ENU mutagenesis in mice. Results show that the ENU-induced C120Y mutation in OVOL2 causes a defect in immune cell development, including T cells and B cells. Interestingly, the severe defect in T cell development is ascribed to defective function of thymic stromal cells. Foxn1-Cre-mediated OVOL2 deletion phenocopies the ENU mutants in causing defective T cell development. Results are also provided to show the interaction of OVOL2 protein with BRAF-HDAC complex and OVOL2-mediated negative regulation of BRAF-HDAC activity. It is further suggested that OVOL2 regulates epithelial-mesenchymal transition (EMT) in postnatal thymic epithelial cells. Postnatal maintenance of the thymus function is an important topic in basic and clinical immunology. The discovery that OVOL2 is a key regulator of postnatal thymus involution is interesting. However, the manuscript suffers from the following uncertainties to draw the major conclusion that OVOL2 sustains TEC identity by preventing EMT.

1. The manuscript describes that the medullary region and mTECs are severely affected in the OVOL2 mutant mice and in Foxn1-Cre-mediated OVOL2-deleted mice. On the contrary, the effects to the thymic cortex or cTECs are not clearly presented. cTECs rather than mTECs play a major role in thymic T cell production by promoting T cell specification in lymphoid progenitors and inducing positive selection of newly generated T cells, so that the defective T cell production is most likely due to some problems in cTECs rather than mTECs. The manuscript describes that cTECs are not reduced in number but should provide more in details how OVOL2 deficiency may affect cTECs. Are cTECs reduced in T cell specifying molecules, such as DLL4, IL-7, SCF, and CXCL12, and/or positive selection inducing molecules, such as MHC-I/II, Psmb11, cathepsin L, Prss16, and CD83? Does OVOL2 deficiency postnatally reduce Foxn1 in cTECs?

We analyzed thymocyte positive selection defined by CD5 or CD69 and TCR- β cell surface expression in 12-week-old or 4-week old *Ovol2*^{C120Y/-} mice and their WT littermates, and in 20-week-old *Foxn1-Cre;Ovol2*^{fl/fl} mice and *Ovol2*^{fl/fl} littermates. **We present the data here and in Supplementary Fig. 9.** We discovered decreased percentages of T cells undergoing positive selection in 12-week-old *Ovol2*^{C120Y/-} mice, with milder impairments observed in 4-week-old *Ovol2*^{C120Y/-} mice (Supplementary Fig. 9A-F). 20-week-old *Foxn1-cre;Ovol2*^{fl/fl} mice phenocopied *Ovol2*^{C120Y/-} mice (Supplementary Fig. 9G-I). The data demonstrate defective positive selection in *Ovol2*^{C120Y/-} mice and in mice with TEC-targeted OVOL2 deletion.

Next, we analyzed the expression of genes encoding T cell specifying molecules, including *Dll4*, *Il-7*, *Scf*, and *Cxcl12*, and genes necessary for positive selection, including *Psbm11*, *Cathepsin L*, and *Prss16*, in cTECs from 12-week-old and 4-week-old *Ovo2*^{C120Y/-} mice. We discovered diminished expression of each of these genes (with the exception of *Psbm11* in 4-week-olds) (Fig. 4O and 4P). The expression of *Foxn1* was normal in *Ovo2*^{C120Y/-} mice (Fig. 4O and 4P). MHC class II protein expression on the surface of *Ovo2*^{C120Y/-} cTECs was reduced, although MHC class I protein expression was normal (Fig. 4Q and 4R). Together with data on positive selection, these findings suggest that cTECs in *Ovo2*^{C120Y/-} mice are functionally impaired in their ability to promote T cell specification and positive selection, resulting in defective differentiation of early T cell progenitors to DP T cells, despite the fact that numbers of cTECs in *Ovo2*^{C120Y/-} mice were normal. **We present the new data below and in Fig. 4O-R.**

2. The lack in the thymic medulla (Fig. 3B and 3C) is not convincing and should be supported by keratin 5 and/or keratin 14 immunohistological analysis.

We used immunohistochemistry to analyze cytokeratins in *Ovol2*^{C120Y^{-/-} and *Foxn1-cre;Ovol2*^{fl/fl} thymi. We present the data below and have added them in Supplementary Fig. 5A-C. Cytokeratin (CK) 8 clearly labeled the cortical regions and was absent from the medullary regions of WT thymus, while the ubiquitous CK8 staining in the *Ovol2*^{C120Y^{-/-} and *Foxn1-cre;Ovol2*^{fl/fl} thymi suggested that most of the medullary epithelium was lost (Supplementary Fig. 5A). CK5 and CK14 labeled the medullary regions and were absent from the cortical regions of WT thymus, while in *Ovol2*^{C120Y^{-/-} and *Foxn1-cre;Ovol2*^{fl/fl} thymi the weak and diffuse ubiquitous staining suggested low levels of these epitopes (Supplementary Fig. 5B and 5C). Note that for WT samples, sections of a portion of the thymus are shown; for mutant samples, sections of the whole thymus are shown. These data support the conclusions in our original manuscript. However, we revised our statement in the Results to say, “*Ovol2*^{C120Y^{-/-} thymi ... had a greatly reduced medullary region (Fig. 3B).”}}}}

3. Despite the lack in the medulla and the reduction in *Aire* and *Fzf2*, the lack of autoimmunity described in the manuscript is a surprise, because T cell self-tolerance is the best-known function of the thymic medulla. The description of “no lymphocytic infiltration of tissues” should be supported by showing the results.

We conducted further analyses for signs of autoimmunity in *Ovo12*^{C120Y/C120Y} mice. The new data are presented below and in Supplementary Fig. 11. *Ovo12*^{C120Y/C120Y} mice displayed normal antibody titers (Supplementary Fig. 11A). *Ovo12*^{C120Y/C120Y} mice had severe lymph node and spleen hypoplasia (Supplementary Fig. 11B and 1B). *Ovo12*^{C120Y/C120Y} mice had normal percentages of CD4+FoxP3+ Treg cells in the lymph node and spleen, although they had decreased numbers of these cells because of lymph node and spleen hypoplasia (Supplementary Fig. 11C-D). Also, there were no signs of lymphocytic infiltration into the kidney, lung, liver, intestine, and salivary gland (Supplementary Fig. 11E-I). In contrast, there were decreased proportions of lymphocytes (CD45+), especially CD19+ and CD3+ cells in several organs, and *Ovo12*^{C120Y/-} mice had reduced B cell and T cell numbers. Finally, as previously reported (PMID:

36228616), H&E staining showed no lymphocytic infiltration of the liver and skin in *Ovol2*^{C120Y/-} mice. Therefore, we concluded that *Ovol2*^{C120Y/-} mice do not have autoimmunity.

4. Foxn1-Cre is not a TEC-specific deleter but deletes flox genes in TECs and skin cells, although a TEC-specific deleter is available (PMID: 35042581). The thymus involution late in the life could be secondary to systemic stress due to the problems in skin cells. Is the skin and hair cells normal without signs of lesions in those animals?

We appreciate this discerning question. Unfortunately, $\beta 5t$ -Cre mice, which are TEC-specific, are not commercially available. In support of a skin/hair-independent effect on the thymus of *Foxn1-Cre*-driven *OVOL2* deletion, we observed that 20-week-old *Foxn1-Cre;Ovol2*^{fl/fl} mice showed normal skin and hair (pictured below for the reviewer only) but nonetheless had hypoplastic thymi that were surrounded by adipose tissue (black arrows). We present these data below and in Fig. 3K and Supplementary Fig. 6C.

These data demonstrate that the thymus phenotypes in *Foxn1-Cre;Ovol2^{fl/fl}* mice are not secondary to systemic stress due to problems in skin cells.

5. Purity and cTEC/mTEC profiles of isolated TECs for biochemical analysis should be shown (Fig 4, Fig. 6). Do those data from isolated TECs primarily reflect the biology of mTECs rather than cTECs?

In Fig. 6I (original Fig. 4I) and Fig. 8A-D (original Fig. 6A-D), samples were from *Ovol2^{C120Y/C120Y}* mice 4.5 weeks of age, when they have normal proportions of cTECs and mTECs. We measured the frequencies of mTECs and cTECs at 4 weeks of age and found that approximately 75-85% and 10-24% of TECs are medullary or cortical, respectively (presented below and in Fig. 3H and Supplementary Fig. 6D). Thus, the biochemical experiments in original Figs. 4 and 6 primarily represent the biology of mTECs.

Regarding cell purity, we sorted mTECs, cTECs, and total TECs with purity rather than yield as a priority. The purity of these cells was >80%.

6. Regulation of EMT is an important conclusion of the study and should be more directly evaluated by immunohistological visualization of EMT in the thymic sections (e.g. PMID: 19648267).

We agree and thank the reviewer for this suggestion. To look more directly at EMT of TECs in *Ovol2*^{C120Y/C120Y} thymi, we generated mice in which developing TECs were marked with the fluorescent reporter tdTomato. These mice (*Ovol2*^{C120Y/C120Y}; *Foxn1-Cre*; *Ai9*) express *Ai9*, a Cre reporter knocked into the *Rosa26* locus, which contains the *tdTomato* gene separated from a CAG promoter by a loxP-flanked STOP cassette. CAG promoter-driven expression of tdTomato is activated by Cre-mediated deletion of the STOP cassette. Driven by the *Foxn1* promoter, Cre expression activates tdTomato fluorescence in TEC progenitors and keratinocytes. Note the pink color of the skin in *Ai9*⁺ mice.

We analyzed the *tdtomato*^{high} TEC population and present these new data below and in new Fig. 5C-G. We observed that ~80% of *tdtomato*^{high} cells in WT thymi were epithelial cells, whereas only ~26% were epithelial cells in *Ovol2*^{C120Y/C120Y} thymi from 20-week-old mice (Fig. 5C-F). Moreover, ~50% of *Ovol2*^{C120Y/C120Y} *tdtomato*^{high} cells were CD31-EpCAM-PDGFR- α/β ⁺ cells; i.e. mesenchymal cells, in comparison to only ~4% of WT *tdtomato*^{high} cells (Fig. 5C-F). Endothelial cell frequencies among *tdtomato*^{high} cells were slightly elevated in *Ovol2*^{C120Y/C120Y} thymi compared to WT thymi (Fig. 5C-F). These data definitively demonstrate that in *Ovol2*^{C120Y/C120Y} thymi, TEC were converted to mesenchymal cells, which was also corroborated by analysis of epithelial and mesenchymal gene expression (Fig. 5G).

We note that *Foxn1-Cre;Ai9* mice on a WT background contain a population of cells with intermediate *tdTomato*^{int} that is not detected in *Ai9* mice lacking *Cre* expression (figure below, and Supplementary Fig. 12). We found that the population of *tdTomato*^{int} cells in *Foxn1-Cre;Ai9* thymi

was >99.8% Lin+CD45+ (marking cells of hematopoietic origin), while the remaining 0.17% tdTomato^{int} Lin-CD45- cell population was not thymic stromal cells, in that they did not express surface markers of TECs, endothelial, or mesenchymal cells. The specific expression of Foxn1-Cre in TECs and keratinocytes has been reported dozens of times. Our bone marrow transplantation and adoptive transfer experiments (Fig. 2), and the epithelial-specific phenotypes observed in *Foxn1-Cre;Ovol2^{fl/fl}* mice strongly support a hematopoietic- and T cell-extrinsic defect impairs T cell development in *Ovol2^{C120Y/-}* mice. Thus, our analyses of marked TECs focused on the clearly separated population of tdTomato^{high} cells that had appropriate epithelial characteristics.

Further support that mesenchymal cells originated from TEC via EMT in *Foxn1-cre;Ovol2^{fl/fl}* mice included the finding that thymi from *Foxn1-cre;Ovol2^{fl/fl}* mice were surrounded by adipose tissue (black arrows) similar to thymi in *Ovol2^{C120Y/-}* mice (figure below, and now in Fig. 3K and Supplementary Fig. 6C). In *Foxn1-cre;Ovol2^{fl/fl}* mice this adipose tissue was only present around the thymus, whereas in *Ovol2^{C120Y/-}* mice the adipose tissue was present throughout the body. This strongly suggests that deletion of OVOL2 in TECs results in the conversion of TECs to mesenchymal cells, specifically adipocytes, which accumulated around the thymus.

Reviewer #3 (expert in T cell development and transcriptional regulation of T cell fate):

We appreciate the reviewer's comments. The following are our point-by-point responses:

The authors study the effects of loss of *Ovol2*, previously implicated in maintaining expression of epithelial lineage genes. Mice with gremlin defects in *Ovol2* were shown to have profound defects in T and B lymphopoiesis, as well as NK cell effector function, and reciprocal bone marrow chimeras indicated the requirement for *Ovol2* is extrinsic to hematopoietic lineage cells. The authors evaluate the requirement for *Ovol2* specifically in thymic epithelial cells (TEC) using a conditional allele crossed to *Foxn1-Cre*. Interestingly, these mice have normal thymus size initially, and normal frequencies of TEC populations, but the thymus progressively declines in size from 4 to 20 weeks, with an apparent loss of mTEC populations, and loss of *Aire* and *Fezf2* transcription factor expression (genes associated with establishment of self-tolerance) in mTECs. The authors subsequently perform biochemical analyses in EL4 and HeK293T cells, and their studies support a model where *Ovol2* is required to displace *Lsd1* from interactions with a complex containing *Rco1*, *Lsd1* and HDAC. In the absence of *Ovol2*, there is increased association of *Lsd1* in this complex, and increased histone H3K4 demethylation. This model is consistent with ATAC-seq data showing increased open chromatin in thymic epithelial cells of *Ovol2* deficient mice. Because genes altered in expression in *Ovol2* deficient thymus include genes implicated in epithelial-mesenchymal transition, the authors conclude that *Ovol2* is implicated in thymic epithelial cell identity.

The work that is performed here is certainly of interest. However, there are areas where significant improvement is needed. The analysis of thymic epithelial cell diversity and T cell development in *Foxn1-Cre*, *Ovol2^{f/f}* mice needs to be significantly enhanced. Autoimmunity is expected in mice with cortical TEC but lacking medullary TEC, and the authors pay almost no attention to this point except to remark that the mice show no signs of autoimmunity. The authors do not directly show that TEC lacking *Ovol2* undertake EMT, as opposed to dying of apoptosis.

1. In more detail - levels of *Kit* and *CD25* on DN subsets should be assessed to determine whether reduced frequencies of ETP are evident, indicating defects in thymic settling contribute to the observed phenotype, perhaps due to reduced expression of chemokines by TEC (for example, see Zlotoff et al., *Blood*, 2010).

We thank the reviewer for this excellent suggestion. We analyzed early T-cell progenitors (ETP) in thymi from 12-week-old *Ovol2^{C120Y/-}* mice and 20-week-old *Foxn1-cre;Ovol2^{f/f}* mice and littermate control mice (based on markers in PMID: 20543111). We present the data below and in Fig. 4F-H and Supplementary Fig. 8A-F. We discovered numerical reductions of ETP in both mutant groups (Fig. 4G and H). The frequencies of these ETP were elevated, reflecting their impaired development to the DN stage (Supplementary Fig. 8A-D). 4-week-old *Ovol2^{C120Y/-}* mice had only slightly reduced numbers of ETP (Fig. 4F), and their frequencies were similar to those of WT mice at this age (Supplementary Fig. 8E and 8F).

We also examined chemokine expression by TEC, which is necessary for ETP recruitment and settling. According to published data (PMID: 19965655), CCR7 and CCR9 expressed in CLP play an important role in recruiting hematopoietic progenitors to the adult thymus. CCR7 has 2 known ligands, CCL19 and CCL21. Both ligands are expressed in the adult thymus by mTEC, and CCL19 is additionally associated with the endothelium. Also, P-selectin and CCL25 are expressed by the thymic endothelium and are essential signals for thymocyte settling (PMID: 19289576). We measured *Ccl19* and *Ccl21* transcript expression in mTECs and *P-selectin* and *Ccl25* transcript expression in thymic endothelial cells from *Ovo12^{C120Y/-}* mice (4 weeks and 12 weeks of age) and *Foxn1-Cre;Ovo12^{fl/fl}* mice (20 weeks of age). **We present the data below and in Fig. 4I-N.**

We discovered that transcript levels of *Ccl19* and *Ccl21* were reduced in mTEC from 12-week-old *Ovol2^{C120Y/-}* mice (Fig. 4J) and 20-week-old *Foxn1-cre;Ovol2^{fl/fl}* mice (Fig. 4K). An endothelial cell deficiency in the transcript levels of *P-selectin* and *Ccl25* was observed in thymic endothelial cells from 12-week-old *Ovol2^{C120Y/-}* mice (Fig. 4M), but not 20-week-old *Foxn1-cre;Ovol2^{fl/fl}* mice (Fig. 4N). *Ccl19* displayed decreased expression in 4-week-old *Ovol2^{C120Y/-}* mTECs (Fig. 4I), although *Ccl21*, *P-selectin*, and *Ccl25* expression were normal (Fig. 4I and 4L). These data suggest a mechanistic basis for the reduction of ETP in thymi from OVOL2 mutant mice, namely the failure of thymic stromal cells to express critical recruiting molecules. In addition, TEC-specific OVOL2 deletion altered recruiting molecule expression in TECs but not in endothelial cells.

2. Defects in negative selection should be assessed, minimally using markers for stages of T cell development (for example, see Baran-Gale, eLife, 2020).

We assessed negative selection in 4-week-old or 12-week-old *Ovol2^{C120Y/-}* and 20-week-old *Foxn1-Cre;Ovol2^{fl/fl}* mice (stages defined as in PMID: 32840480). We present the data below and in

Supplementary Fig. 10. We found that T cell negative selection was altered in *OVOL2* mutant mice, in that greater percentages of T cells were present at each stage of negative selection in 12-week-old *Ovo12*^{C120Y/-} thymi (Supplementary Fig. 10A). Smaller increases in the frequencies of T cells at various stages of negative selection were detected in 4-week-old *Ovo12*^{C120Y/-} mice and in 20-week-old *Foxn1-cre;Ovo12*^{fl/fl} mice (Supplementary Fig. 10B-C).

3. Defects in immune tolerance should be assessed (Korai et al., JEM, 2017), as expected in these mice; or an explanation why autoimmunity is not evident is needed.

We conducted analyses for signs of autoimmunity in *Ovo12*^{C120Y/C120Y} mice. **The new data are presented below and in Supplementary Fig. 11.** *Ovo12*^{C120Y/C120Y} mice displayed normal antibody titers (Supplementary Fig. 11A). *Ovo12*^{C120Y/C120Y} mice had severe lymph node and spleen hypoplasia (Supplementary Fig. 11B and 1B). *Ovo12*^{C120Y/C120Y} mice had normal percentages of CD4+FoxP3+ Treg cells in the lymph node and spleen, although they had decreased numbers of these cells because of lymph node and spleen hypoplasia (Supplementary Fig. 11C-D). Also, there were no signs of lymphocytic infiltration into the kidney, lung, liver, intestine, and salivary gland (Supplementary Fig. 11E-I). In contrast, there were decreased proportions of lymphocytes (CD45+), especially CD19+ and CD3+ cells in several organs, and *Ovo12*^{C120Y/-} mice had reduced B cell and T cell numbers. Finally, as previously reported (PMID: 36228616), H&E staining showed no lymphocytic infiltration of the liver and skin in *Ovo12*^{C120Y/-} mice. Therefore, we concluded that *Ovo12*^{C120Y/-} mice do not have autoimmunity.

We hypothesize that $Ovol2^{C120Y/-}$ mice do not have autoimmunity in part because, for unknown reasons, they have greater percentages of T cells at all stages of negative selection compared to WT mice (Supplementary Fig. 10A). Moreover, $Ovol2^{C120Y/C120Y}$ mice displayed normal AIRE expression in mature mTECs (UEA1+ MHC-II^{high}) (Fig. 4T). These data suggest that preserved AIRE expression in mature mTECs may support efficient negative selection in $Ovol2^{C120Y/-}$ thymi.

4. Finally, it is difficult to assess apoptosis in TEC, but Foxn1-Cre based lineage-tracing can establish whether TEC adopt other mesenchymal cell fates in the absence of *Ovol2*. This needs to be performed, and given the answers, the proposed mechanism should be discussed considering the possibilities. With these improvements, I think this will be a fine contribution.

We agree and thank the reviewer for this suggestion. To understand whether mesenchymal cells originated from TECs in $Ovol2^{C120Y/C120Y}$ thymi, we generated mice in which developing TECs were marked

with the fluorescent reporter tdTomato. These mice (*Ovol2*^{C120Y/C120Y};*Foxn1-Cre*;*Ai9*) express Ai9, a Cre reporter knocked into the *Rosa26* locus, which contains the *tdTomato* gene separated from a CAG promoter by a loxP-flanked STOP cassette. CAG promoter-driven expression of tdTomato is activated by Cre-mediated deletion of the STOP cassette. Driven by the *Foxn1* promoter, Cre expression activates tdTomato fluorescence in TEC progenitors and keratinocytes. Note the pink color of the skin in Ai9+ mice.

We analyzed the *tdtomato*^{high} TEC population and present these new data below and in new Fig. 5C-G. We observed that ~80% of *tdtomato*^{high} cells in WT thymi were epithelial cells, whereas only ~26% were epithelial cells in *Ovol2*^{C120Y/C120Y} thymi from 20-week-old mice (Fig. 5C-F). Moreover, ~50% of *Ovol2*^{C120Y/C120Y} *tdtomato*^{high} cells were CD31-EpCAM-PDGFR- α/β + cells; i.e. mesenchymal cells, in comparison to only ~4% of WT *tdtomato*^{high} cells (Fig. 5C-F). Endothelial cell frequencies among *tdtomato*^{high} cells were slightly elevated in *Ovol2*^{C120Y/C120Y} thymi compared to WT thymi (Fig. 5C-F). These data definitively demonstrate that in *Ovol2*^{C120Y/C120Y} thymi, TEC were converted to mesenchymal cells, which was also corroborated by analysis of epithelial and mesenchymal gene expression (Fig. 5G).

We note that *Foxn1-Cre;Ai9* mice on a WT background contain a population of cells with intermediate $\text{tdTomato}^{\text{int}}$ that is not detected in *Ai9* mice lacking Cre expression (figure below, and Supplementary Fig. 12). We found that the population of $\text{tdTomato}^{\text{int}}$ cells in *Foxn1-Cre;Ai9* thymi was >99.8% $\text{Lin}^+\text{CD45}^+$ (marking cells of hematopoietic origin), while the remaining 0.17% $\text{tdTomato}^{\text{int}}$ $\text{Lin}^-\text{CD45}^-$ cell population was not thymic stromal cells, in that they did not express surface markers of TECs, endothelial, or mesenchymal cells. The specific expression of *Foxn1-Cre* in TECs and keratinocytes has been reported dozens of times. Our bone marrow transplantation and adoptive transfer experiments (Fig. 2), and the epithelial-specific phenotypes observed in *Foxn1-Cre;Ovol2^{fl/fl}* mice strongly support a hematopoietic- and T cell-extrinsic defect impairs T cell development in *Ovol2^{C120Y/-}* mice. Thus, our analyses of marked TECs focused on the clearly separated population of $\text{tdTomato}^{\text{high}}$ cells that had appropriate epithelial characteristics.

Further support that mesenchymal cells originated from TEC via EMT in *Foxn1-cre;Ovol2^{fl/fl}* mice included the finding that thymi from *Foxn1-cre;Ovol2^{fl/fl}* mice were surrounded by adipose tissue (black arrows) similar to thymi in *Ovol2^{C120Y/-}* mice (figure below, and now in Fig. 3K and Supplementary Fig. 6C). In *Foxn1-cre;Ovol2^{fl/fl}* mice this adipose tissue was only present around the thymus, whereas in *Ovol2^{C120Y/-}* mice the adipose tissue was present throughout the body. This strongly suggests that deletion of OVOL2 in TECs results in the conversion of TECs to mesenchymal cells, specifically adipocytes, which accumulated around the thymus.

REVIEWER COMMENTS

Reviewer #1 (expert in TEC development and epigenetics):

The rebuttal answered the reviewer's comments satisfactorily with the notable exception of the last point (i.e. query #16) which remains in fact unaddressed.

The original statement "OVOL2 might also contribute to TEC development before the period of Foxn1 expression' is further discussed in the response to the first reviewer's query #3. There are a few factual errors in the author's rebuttal (e.g. FOXP1 expression does not already occur at E8, as claimed, as the thymus anlage is not formed before E10.5; vascularisation of the thymus anlage during embryogenesis only happens well after FOXP1 expression has been initiated) and hence their explanations remain do not serve as a clarification.

The revised Suppl. Fig 5A-C is difficult to interpret given the low quality of the IHC analysis.

Revised Manuscript: Additional comments

Text, Line 105: The comment referring to "a block" would need to be edited as only a partial block is observed as documented by the presence of more mature phenotypic stages in the mutant mice.

Text, Line 106: "The proportion of mature T cells (CD8+TCR- β + cells among CD8+ SP 107 thymocytes) was elevated in the *Ovol2*C120Y⁻ mice, reflecting the reduced frequency of newly produced T cells (Supplementary Fig. 1D). The conclusion of this sentence does not make sense in light of the data presented.

Text, Line 228: reference is made to mTEC isolation at E13.5. As medullary lineage TEC are extremely rare - if present at all - at this early stage, it would be informative to know, their frequency, phenotype and how many cells were collected at this developmental stage for the analysis.

Text, Line 234: "These data also imply an important role for OVOL2 in the maintenance of cTEC function despite a lack of effect on cTEC numbers." The conclusion of this sentence requires more explanation as it is in its present statement unsubstantiated by data.

Text, Line 291: The determination of "Protein levels of AIRE (intracellular)" is not informative (and possibly misleading) as the relative frequency of mature TEC, the only TEC subpopulation that expresses AIRE at substantial levels, is reduced (Figure 3G).

Data related to Figure 5C-F: The figure legend to panel 5C does not detail which cells are displayed. Consequently, it is unclear and also not explained why the dTomato signal identifies roughly two different populations of labelled cells, namely those that express intermediate and those that are marked with very high levels of the fluorochrome, respectively. The total percentage of positively labelled cells (independent of the level of label) differs between wild type and mutant mice, which needs further explanation. Moreover, the cell types that label either highly or only intermediately will need to be defined so as to explain the difference in label between wild type and mutant mice. Finally, it is not clearly explained whether only cells with the highest dTomato label are analysed in panels E-F.

Panel 5D demonstrates positivity for transmembrane homophilic receptor CD31 on cells simultaneously labelled with dTomato. CD31 is used here to identify endothelial cells but the gating of cells displayed in that panel is not explained and CD31 expression can also be observed in some subsets of haematopoietic cells, including granulocytes, macrophages, dendritic cells (DCs), and B-cells. Hence, how are the authors sure that the CD31+ cells in their analysis are indeed endothelial cell and if so how is this unexpected labelling explained given that the experiment is designed to express the label initially exclusively in TEC and only subsequently in cells that transition from them. However, an epithelial to endothelial transition has only been noted in the malignant tumour environment. The finding of labelled endothelial cells thus requires a further explanation.

Reviewer #2 (expert in thymic function and TEC development):

I appreciate that the authors constructively responded to all my comments (I was reviewer #2).

Reviewer #3 (expert in T cell development and transcriptional regulation of T cell fate):

This is well revised, and the authors have addressed essentially all of my comments. I am supportive.

Reviewer #1 (expert in TEC development and epigenetics):

We appreciate the reviewer's comments. The following are our point-by-point responses:

The rebuttal answered the reviewer's comments satisfactorily with the notable exception of the last point (i.e. query #16) which remains in fact unaddressed.

We tested the expression of several tissue restricted antigens (TRA) in mature mTEC of WT and *Ovol2*^{C120Y/C120Y} mice. We chose four AIRE-dependent TRAs: *Tff3*, *Ins2*, *Mup1*, and *Spt1*, which were downregulated in the Aire deficient mouse according to publicly available microarray data (PMID: 15983066). *Insulin2* (*Ins2*) knockout non-obese diabetic (NOD) CD4+ and CD8+ T cells respond to insulin and enhance development of autoimmune diabetes (PMID: 19874548 and PMID: 19966211). We chose four FEZF2-dependent TRAs: *Ttr*, *Amy2a*, *Afp*, and *Muc1* (PMID: 26544942). **We present these new data below and in Supplementary Fig. 11A-B.** Compared to WT littermates, the expression of *Tff3*, *Ins2*, *Mup1*, *Amy2a*, and *Afp* were preserved or elevated in mature mTECs from 12-week-old *Ovol2*^{C120Y/-} mice. Only *Ttr* expression was reduced in *Ovol2*^{C120Y/-} mice.

We hypothesize that *Ovol2*^{C120Y/-} mice do not have autoimmunity in part because, for unknown reasons, they have greater percentages of T cells at all stages of negative selection compared to WT mice (Supplementary Fig. 10A). Moreover, *Ovol2*^{C120Y/C120Y} mice displayed normal AIRE expression in mature mTECs (UEA1+ MHC-II^{high}) (Fig. 4T). Together with the intact transcript expression of selected TRAs, these data suggest that preserved AIRE expression in mature mTECs may support efficient negative selection in *Ovol2*^{C120Y/-} thymi.

The original statement "OVOL2 might also contribute to TEC development before the period of Foxn1 expression' is further discussed in the response to the first reviewer's query #3. There are a few factual errors in the author's rebuttal (e.g. FOXN12 expression does not already occur at E8, as claimed, as the thymus anlage is not formed before E10.5; vascularisation of the thymus anlage during embryogenesis only happens well after FOXN1 expression has been initiated) and hence their explanations remain do not serve as a clarification.

We apologize for the confusion caused by our response to query #3. Please allow us to explain. The response had two parts. The first part provided an "another explanation ... for a role of OVOL2 in thymus development that is TEC independent" for which we have molecular evidence. We explained that endothelial cells were increased in thymi when OVOL2 was mutated globally (*Ovol2*^{C102Y/-}) but not when it was deleted only in TECs (*Foxn1-cre;Ovol2*^{fl/fl}).

The second part of our response described why we think that OVOL2 does not contribute to TEC development before the period of *Foxn1* expression; this part of our response was not related to the increase in endothelial cells in *Ovol2*^{C102Y/-} thymi. We did make an error in stating that "*Foxn1* expression is initiated on E8;" as the reviewer knows, it is detected in the thymus primordium from E11.25 (PMID: 11335122). Nonetheless, our reasoning for why it is unlikely that OVOL2 contributes to TEC development

before the period of *Foxn1* expression (i.e. around midgestation) still holds, in that *Ovol2* expression is relatively lowest in the embryo and ramps up postnatally. This suggests that *OVOL2* function becomes more important postnatally, and we therefore deleted from the revised manuscript the statement quoted above by the reviewer.

The revised Suppl. Fig 5A-C is difficult to interpret given the low quality of the IHC analysis.

We used higher magnification to re-analyze cytokeratin (CK) 8, CK5, and CK14 IHC staining of thymus sections. We found CK8 clearly labeled cortical epithelium and was absent from the medullary regions of WT thymus. However, the ubiquitous CK8 staining in the *Ovol2*^{C120Y/-} and *Foxn1-cre;Ovol2*^{fl/fl} thymi suggests that most of the medullary epithelium has been lost. CK5 clearly labeled the medullary regions and was absent from the cortical regions of WT thymus. The reduced medullary region in *Foxn1-cre;Ovol2*^{fl/fl} thymi was also visible by CK5 staining. **We present these data below and in Fig. Supplementary Fig. 5A-B.** We agree with the reviewer that the CK14 staining was not ideal, and likely represented background staining. This may be due to non-specific signal from the CK14 antibody, so we deleted the CK14 staining.

Supplementary Fig. 5

Revised Manuscript: Additional comments

Text, Line 105: The comment referring to “a block” would need to be edited as only a partial block is observed as documented by the presence of more mature phenotypic stages in the mutant mice.

The reviewer is correct and we edited the sentence to state that a partial block was caused.

Text, Line 106: “The proportion of mature T cells (CD8+TCR-β+ cells among CD8+ SP thymocytes) was elevated in the *Ovol2*^{C120Y/-} mice, reflecting the reduced frequency of newly produced T cells (Supplementary Fig. 1D). The conclusion of this sentence does not make sense in light of the data presented.

We edited the sentence to state, “The proportion of CD8+TCR-β+ cells among CD8+ SP thymocytes was elevated in the *Ovol2*^{C120Y/-} mice.”

Text, Line 228: reference is made to mTEC isolation at E13.5. As medullary lineage TEC are extremely rare - if present at all - at this early stage, it would be informative to know, their frequency, phenotype and how many cells were collected at this developmental stage for the analysis.

This was shown in Supplementary Fig. 6D. We collected more than 70,000 cells at each developmental stage.

Text, Line 234: “These data also imply an important role for OVOL2 in the maintenance of cTEC function despite a lack of effect on cTEC numbers.” The conclusion of this sentence requires more explanation as it is in its present statement unsubstantiated by data.

We interpreted the facts that *Ovol2* expression by cTECs increased to its highest levels at 12 weeks of age, and that cTECs had greater *Ovol2* expression than mTECs at all postnatal ages tested, to suggest that OVOL2 has some function in cTECs postnatally. In support of this conclusion, cTECs displayed impaired expression of several molecules necessary for positive selection, and T cell positive selection was reduced in 12-week-old *Ovol2*^{C120Y/-} mice and 20-week-old *Foxn1-cre;Ovol2*^{fl/fl} mice. Those were shown in Fig. 4O-R.

Text, Line 291: The determination of “Protein levels of AIRE (intracellular)” is not informative (and possibly misleading) as the relative frequency of mature TEC, the only TEC subpopulation that expresses AIRE at substantial levels, is reduced (Figure 3G).

At the reviewer’s suggestion, we measured the expression of eight tissue restricted antigens (TRA) as an indication of the functionality of mTECs from *Ovol2*^{C120Y/-} mice. Compared to WT littermates, the expression of AIRE or FEZF2 dependent TRAs, including *Tff3*, *Ins2*, *Mup1*, *Amy2a*, and *Afp* was preserved in mature mTECs from 12-week-old *Ovol2*^{C120Y/-} mice (with the exception of *Ttr*). We present these new data below and in Fig. Supplementary Fig. 11A-B.

The fact is that we do not detect signs of autoimmunity despite the reduction in $mTEC^{MHC-II^{(high)}}$ and $mTEC^{AIRE^{(high)}}$ cells, and we agree with the reviewer that this is quite interesting. *Ovo12^{C120Y/-}* mice had greater percentages of T cells at all stages of negative selection compared to WT mice, suggesting that the residual $mTEC^{AIRE^{(high)}}$ cells were sufficient to support the negative selection of T cells in *Ovo12^{C120Y/-}* thymi. Another possibility is that defects of T cell function preclude autoimmunity in mutant mice.

Data related to Figure 5C-F: The figure legend to panel 5C does not detail which cells are displayed.

The scatter plot in Fig. 5C shows flow cytometric analysis of cells from 20-wk-old *Ovo12^{C120Y/C120Y}* and WT thymi, visualized according to side scatter profile and tdTomato^{high} fluorescent signal. We apologize for not explicitly describing this information in the legend and have now included it.

Consequently, it is unclear and also not explained why the dTomato signal identifies roughly two different populations of labelled cells, namely those that express intermediate and those that are marked with very high levels of the fluorochrome, respectively.

The presence of both a tdTomato^{high} population and a tdTomato^{int} population was unexpected to us as well. To our knowledge, this effect has not previously been reported and therefore we examined the identity of these populations based on flow cytometric analysis of cell surface markers distinguishing hematopoietic vs. non-hematopoietic cells, epithelial vs. non-epithelial cells, and mesenchymal vs. non-mesenchymal cells. In WT mice, 99.8% of tdTomato^{int} cells are Lin⁺CD45⁺, indicating that they are T cells or early T cell progenitors. The remaining cells (0.2% of tdTomato^{int} cells) are a CD31^{neg}EpCAM^{neg}PDGFR- α ^{neg} cell type, and are not epithelial cells, endothelial cells, or mesenchymal cells. The majority of tdTomato^{high} cells are Lin⁻CD45⁻, and among those 80.2% are epithelial cells, 10.5% are endothelial cells, and 4.4% are mesenchymal cells. Those data were shown in Supplementary Fig. 12.

The total percentage of positively labelled cells (independent of the level of label) differs between wild type and mutant mice, which needs further explanation. Moreover, the cell types that label either highly or only intermediately will need to be defined so as to explain the difference in label between wild type and mutant mice.

As described in our response just above, the tdTomato^{high} and tdTomato^{int} populations were defined in Supplementary Fig. 12. The frequency of tdTomato^{pos} cells is lower in *Ovo12^{C120Y/C120Y}* mice relative to WT mice. Based on the analysis in Supplementary Fig. 12, the reduced frequency of tdTomato^{pos} cells is likely due to the reduction in the number of T cells (i.e. tdTomato^{int} cells) caused by *Ovo12* mutation, as shown in Fig. 1F.

Finally, it is not clearly explained whether only cells with the highest dTomato label are analysed in panels E-F.

We apologize for this omission. The tdTomato^{high} cells were analyzed in Fig. 5D-G and we have added this to the legend.

Panel 5D demonstrates positivity for transmembrane homophilic receptor CD31 on cells simultaneously labelled with dTomato. CD31 is used here to identify endothelial cells but the gating of cells displayed in that panel is not explained and CD31 expression can also be observed in some subsets of haematopoietic cells, including granulocytes, macrophages, dendritic cells (DCs), and B-cells. Hence, how are the authors sure that the CD31⁺ cells in their analysis are indeed endothelial cell and if so how is this unexpected labelling explained given that the experiment is designed to express the label initially exclusively in TEC and only subsequently in cells that transition from them. However, an epithelial to endothelial transition has only been noted in the malignant tumour environment. The finding of labelled endothelial cells thus requires a further explanation.

Fig. 5D shows that 80.2% of tdTomato^{high} cells are epithelial cells, but indeed 10.5% are endothelial based upon CD31 positivity in WT mice. Only the tdTomato^{high} (Lin-CD45-) cells were analyzed in Fig. 5D, thus excluding any hematopoietic cell types. By contrast, the frequency of epithelial cells among tdTomato^{high} cells was reduced to 25.7% in *Ovol2*^{C120Y/C120Y} thymi, while the frequency of endothelial cells was increased to 18.8% in *Ovol2*^{C120Y/C120Y} thymi. We did not investigate the mechanism responsible for tdTomato expression in endothelial cells.

REVIEWERS' COMMENTS

Reviewer #1 (expert in TEC development and epigenetics):

The authors have provided a detailed response to the reviewer's queries and the additional explanations and data provided provide convincing responses to points raised.

I wish to congratulate the authors on a very nice piece of work.

Response to Reviewers

Reviewer #1 (expert in TEC development and epigenetics):

The authors have provided a detailed response to the reviewer's queries and the additional explanations and data provided provide convincing responses to points raised.

I wish to congratulate the authors on a very nice piece of work.

Thank you.